# Population-Free Pareto Tracking for Sample-Efficient Multi-Policy MORL

Zeyu Zhao [1]   Yueling Che [1]   Kaichen Liu [1]   Jian Li [1]   Junmei Yao [1]

## Abstract

Multi-objective reinforcement learning (MORL) is a fundamental framework for real-world decision-making problems involving multiple conflicting criteria. Existing multi-policy (MP) methods typically rely on online evolutionary frameworks that maintain large policy populations, leading to high sample complexity and excessive agent–environment interactions. To mitigate these limitations, we present Multi-policy Pareto Front Tracking (MPFT), a framework without a self-evolving population. It leverages an efficient Pareto-tracking mechanism initialized with single-objective extreme policies to trace the Pareto front, and further densifies sparse regions to achieve an accurate approximation of the full Pareto front. MPFT can be seamlessly integrated with advanced offline MORL algorithms, thereby substantially improving sample efficiency. We evaluate MPFT on six robotic control tasks with up to three objectives and three high-dimensional tasks with more than three objectives. Experimental results show that MPFT outperforms state-of-the-art baselines in terms of hypervolume and expected utility. It also significantly reduces agent–environment interactions. These results further demonstrate that MPFT serves as a general-purpose framework that can seamlessly integrate both online and offline MORL algorithms.

## 1. Introduction

Deep Reinforcement Learning (RL) is a promising approach for solving decision-making problems. It has been widely used to a variety of fields such as robot control (Duan et al., 2016), autonomous driving (Feng et al., 2023), and wireless communications (Zhao et al., 2024). In deep RL, the behavior of the agent is guided by a reward function that defines the optimization objective. However, decision-making problems may involve multiple conflicting objectives, which pose a severe challenge for RL applications in practice (Dulac-Arnold et al., 2021). For example, bipedal robot motion control requires balancing running speed and energy efficiency, which are inherently conflicting objectives: accelerating the robot increases energy consumption, while optimizing energy efficiency generally constrains its speed. In such multi-objective decision-making problems, a single optimal policy does not exist in general, because conflicting objectives cannot be simultaneously optimized. Instead, there exists a Pareto policy set. Each policy in this set corresponds to a distinct objective trade-off, and the final policy selection depends on the user preferences.

In this context, multi-objective reinforcement learning (MORL) has attracted the increasing attention. The single-policy (SP) based MORL was proposed to use one consistent policy to address the user's needs under a given user preference (Yang et al., 2019). However, since the user preferences are very difficult to quantify (Lin et al., 2024a), the SP-MORL policy generally cannot align with varied user preferences in practice. Unlike the SP-MORL, the multi-policy (MP) based MORL generates a high-quality Pareto-approximation policy set, where the user can freely select the policies that align with his/her preferences even in the real-time tasks (Hu & Luo, 2024).

It is noted that almost all of the existing MP-MORL approaches rely on the evolutionary frameworks that require maintaining a self-evolving population of policies to approximate the Pareto front under diverse objective preferences[1]. To let that population evolve in real time through massively parallel interactions and updates, the online RL algorithms like PPO (Schulman et al., 2017) are usually employed. However, the evolutionary framework requires many agent–environment interactions, and online RL has low sample efficiency. As a result, MP-MORL methods built on evolutionary frameworks are hard to deploy in real-world settings where interactions and sampling are expensive.

This paper focuses on designing a general MP-MORL framework. In contrast to evolutionary frameworks, we propose a

---

[1]College of Computer Science and Software Engineering, Shenzhen University, China. Correspondence to: Yueling Che <yuelingche@szu.edu.cn>.

*Proceedings of the 43rd International Conference on Machine Learning*, Seoul, South Korea. PMLR 306, 2026. Copyright 2026 by the author(s).

---

[1]A self-evolving population of policies is one in which, at each iteration, policies are selected from the population, evolved through updates, and then used to update the population itself.

novel multi-policy Pareto front tracking (MPFT) framework that does not require maintaining a self-evolving policy population. MPFT maintains Pareto policy sets that are updated solely based on tracked policies without policy selection, and thus supports advanced offline RL algorithms, achieving a tight approximation of the Pareto front with substantially reduced agent–environment interactions. Specifically, MPFT first identifies approximate Pareto-vertex policies and then tracks the Pareto front using a newly designed Pareto-tracking mechanism. Each tracked policy originates from the same Pareto-vertex policy, eliminating the need to select policies from the Pareto policy set during tracking. The main contributions of this work are summarized as follows:

- We propose the MPFT framework to obtain a high-quality Pareto-approximation policy set in four stages: 1) approximate Pareto-vertex policies via single-objective optimization, 2) track the Pareto front from each Pareto-vertex policy, 3) fill the top-$K$ sparse regions by tracking from $K$ Pareto-interior policies, and 4) combine all tracked policies to complete the Pareto-approximation policy set.

- We propose a novel Pareto-tracking mechanism for Stages 2 and 3, which updates policies along a Pareto-reverse followed by a Pareto-ascent direction, and is theoretically proven to converge exponentially to a neighborhood of the Pareto-stationary set. An objective weight adjustment method is particularly designed in Stage 3 to identify $K$ Pareto-interior policies located in $K$ sparse regions, and then to complete the Pareto front starting from the identified policies.

- We propose multi-objective TD7 algorithm (Fujimoto et al., 2023) that is originally designed for a single objective, and integrate it within the MPFT framework, to further improve sample efficiency and reduce agent–environment interactions.

- We evaluate MPFT on six robotic control tasks with up to three objectives. Results show that MPFT achieves higher hypervolume and expected utility than SOTA benchmarks, while reducing agent–environment interactions by up to 78.22%. Then high-dimensional tasks are also considered, where the MPFT achieves superior hypervolume and expected utility, even when tracking only from a single Pareto-vertex policy.

## 2. Related Works

Existing MORL approaches are broadly categorized into the SP-based and the MP-based approaches.

**SP-MORL**: Meta-RL is widely used in SP-MORL to enable a single policy to adapt to all objective preferences, facilitating rapid online adaptation in scenarios with costly sampling. Similarly, recent offline MORL algorithms, such as those in (Yang et al., 2019; Zhu et al., 2023; Lin et al., 2024b), leverage superior sample efficiency to swiftly derive policies aligned with user preferences. The framework proposed by (Lin et al., 2024a) further extends these benefits by generating preference-aligned policies during deployment and by incorporating safety considerations.

Despite these strengths, several notable limitations persist. First, meta-RL approaches often exhibit suboptimal behaviors under a universal meta policy (Chen et al., 2019). Second, the continual training of meta-RL agents may be unstable due to the issue of catastrophic forgetting (Dohare et al., 2024). Third, the offline MORL algorithms generally depend heavily on users' prior preference inputs, restricting their applicability in domains where such inputs are unavailable (e.g., autonomous driving). It is noted that the paradigm in (Lin et al., 2024a) obviates the need for pre-specified preferences during deployment. However, it is still fundamentally meta-RL–based, and cannot fully circumvent the issue of catastrophic forgetting. Another branch of SP-MORL is Pareto set learning, which indirectly realizes the effect of multi-policy by learning a "generalized policy generator" at once through conditional generation mechanisms (e.g., hypernetworks) (Liu et al., 2025a), but it is still a SP-based approach, which is difficult to cover the entire Pareto front (Lin et al., 2022). In addition the size of the hypernetwork will grow geometrically as the parameters of the policy network increase. Moreover, SP-based approaches suffer from gradient interference across objectives, which leads to performance imbalance over the preference space.

**MP-MORL**: Unlike the SP-MORL, the MP-MORL maintains a set of non-dominated policies (i.e., the Pareto-approximation policy set) to support flexible decision making (Xu et al., 2020; Hayes et al., 2022; Alegre et al., 2022; Hu & Luo, 2024). Hence, the MP-MORL allows users to select solutions that align with their preferences from the learned Pareto-approximation policy set (Horie et al., 2019; Wen et al., 2020).

However, almost all the existing MP-MORL approaches rely on the evolutionary framework with online RL, such as the SOTA algorithms PGMORL (Xu et al., 2020), PA2D-MORL (Hu & Luo, 2024), and C-MORL (Liu et al., 2025b). Hence, the existing MP-MORL approaches generally suffer from the low sample efficiency and large agent-environment interactions. Moreover, under the evolutionary framework, since the policy corresponding to the solution farthest from the reference point is selected in each generation, the PGMORL and PA2D-MORL also suffers from the over-optimization of frequently selected policies and stagnation of others. To address this limitation, C-MORL replaces population-based policy selection during the evolution phase with a crowding-distance-based selection over the non-dominated

set, achieving strong scalability with high-dimensional objectives. However, as other evolutionary methods, its applicability is still constrained in scenarios with expensive interactions. To alleviate the sample inefficiency, (Tran et al., 2023) introduced offline RL in the warm-up phase, but it still uses the online RL for the policy training during the evolutionary phase, and it does not fundamentally solve this problem. In addition, (Alegre et al., 2022) proposed a non-evolutionary approach that learns a convex coverage set (CCS) of successor features for optimal zero-shot transfer to arbitrary linear tasks, though its worst-case cost grows exponentially with the number of objectives. (Alegre et al., 2023) further introduced Dyna-style MORL method (GPI-LS) to alleviate this problem. However, GPI-LS still requires iterative training of multiple scalarized policies, and its computational cost increases with the number of policies needed to approximate the CCS, particularly in high-dimensional objective spaces.

To address the above issues, we propose the novel MPFT framework compatible with both online and offline RL algorithms in this work, to tightly and efficiently approximate the Pareto front, where the time complexity increases linearly with the number of objectives.

## 3. Preliminaries

### 3.1. Multi-Objective Markov Decision Process

The MORL problem is usually modeled as a multi-objective Markov decision process (MOMDP). An MOMDP is defined as the tuple $(\mathcal{S}, \mathcal{A}, \mathcal{P}, \boldsymbol{R}, \gamma)$. As in the standard MDP, the agent under the state $s \in \mathcal{S}$, takes an action $a \in \mathcal{A}$, and transits into a new state $s' \in \mathcal{S}$ with the state transition probability $\mathcal{P}(s'|s, a)$. The agent then obtains a reward vector $\boldsymbol{R} = [R_1, ..., R_m]^\top$ for $m > 1$ objectives with the discount factor $\gamma \in [0, 1]$, where $R_i$ is the reward of the $i$-th objective, $i \in \{1, ..., m\}$, and $\top$ is the transposed operation. Denote the policy parameterized by $\boldsymbol{\theta} \in \mathbb{R}^d$ as $\pi_{\boldsymbol{\theta}} : \mathcal{S} \to \mathcal{A}$, where $d$ is the dimension of $\boldsymbol{\theta}$. Since $\pi_{\boldsymbol{\theta}}$ can be completely represented by $\boldsymbol{\theta}$, we will use $\pi_{\boldsymbol{\theta}}$ and $\boldsymbol{\theta}$ interchangeably in the rest of this paper. Appendix A lists the key notations.

Denote the vector of expected return of the policy $\pi_{\boldsymbol{\theta}}$ as $\boldsymbol{J}(\boldsymbol{\theta}) = [J_1(\boldsymbol{\theta}), J_2(\boldsymbol{\theta}), ..., J_m(\boldsymbol{\theta})]^\top$, where $J_i(\boldsymbol{\theta})$ is the expected return of the $i$-th objective. As in (Xu et al., 2020; Hu & Luo, 2024), $J_i(\boldsymbol{\theta})$ is defined as:

$$J_i(\boldsymbol{\theta}) = \mathbb{E}\left[\sum_{t=0}^{T} \gamma^t R_i(s_t, a_t)\middle| a_t \sim \pi_{\boldsymbol{\theta}}(s_t), s_0 = s\right], \quad (1)$$

where $\sim$ represents the sampling operation, $s_0 = s$ denotes the initial state, and $t \in \{0, \ldots, T\}$ is the timestep. The goal of the MOMDP problem is to find the optimal policy, that maximizes the expected return $\boldsymbol{J}(\boldsymbol{\theta})$ of all the objectives. This can be achieved by using the policy gra-

dient descent method. The policy gradient of the objective $i \in \{1, \ldots, m\}$ is given by:

$$\nabla_{\boldsymbol{\theta}} J_i(\boldsymbol{\theta}) = -\nabla \mathbb{E}\left[\mathbf{Loss}_i(s, a)\right], \quad (2)$$

where $\mathbf{Loss}_i(s, a)$ is the loss function with respect to state $s$ and action $a$. Let $\nabla_{\boldsymbol{\theta}} \boldsymbol{J}(\boldsymbol{\theta}) = [\nabla_{\boldsymbol{\theta}} J_1(\boldsymbol{\theta}), \ldots, \nabla_{\boldsymbol{\theta}} J_m(\boldsymbol{\theta})]^\top \in \mathbb{R}^{m \times d}$ represent the policy gradient matrix of the $m$ objectives.

### 3.2. Pareto-Optimal Policy

For two policies $\pi_{\boldsymbol{\theta}^1}$ and $\pi_{\boldsymbol{\theta}^2}$, where $\boldsymbol{J}(\boldsymbol{\theta}^1) \neq \boldsymbol{J}(\boldsymbol{\theta}^2)$, we say that the policy $\pi_{\boldsymbol{\theta}^1}$ dominates the policy $\pi_{\boldsymbol{\theta}^2}$, if $J_i(\boldsymbol{\theta}^1) \geq J_i(\boldsymbol{\theta}^2), \forall i \in \{1, \ldots, m\}$. In the Pareto policy set, there is no policy dominated by any other policy, i.e., each policy is Pareto-optimal. In other words, in the Pareto policy set, no objective can be further improved without sacrificing another objective. The mapping from the Pareto policy set to the objective space is known as the Pareto front (Xu et al., 2020; Hu & Luo, 2024; Liu et al., 2025b). However, it is usually very difficult to obtain the Pareto policy set in practice. The goal becomes to find the Pareto-approximation policy set that best approximates the Pareto policy set.

### 3.3. Pareto Stationarity

To find the Pareto-approximation policy set, the MOMDP problem is usually transformed into a series of single-objective Markov decision process (SOMDP) problems, where the $m$ objectives are scalarized into a single objective under different objective weight vectors $\boldsymbol{\omega} = [\omega_1, \ldots, \omega_m]^\top$ in different SOMDP problems. For a given $\boldsymbol{\omega}$, the SOMDP problem is formulated as follows:

$$(\text{P1}): \quad \max_{\boldsymbol{\theta}} \ \boldsymbol{J}(\boldsymbol{\theta})^\top \boldsymbol{\omega},$$
$$\text{s.t. } \boldsymbol{\omega} \in \Delta_m, \Delta_m := \{\boldsymbol{\omega} \in \mathbb{R}^m | \omega_i \geq 0, \|\boldsymbol{\omega}\|_1 = 1\},$$

where $\|\cdot\|_1$ represents the L1-norm operation. The solutions to all the SOMDP problems form the Pareto-approximation policy set.

However, the above method requires an exhaustive search of the objective weight vectors, which is impractical. As in (Sener & Koltun, 2018; Désidéri, 2012), a policy $\pi_{\boldsymbol{\theta}}$ is Pareto-stationary, if $\min_{\boldsymbol{\omega} \in \Delta_m} \left\|\nabla_{\boldsymbol{\theta}} \boldsymbol{J}(\boldsymbol{\theta})^\top \boldsymbol{\omega}\right\|_2^2 = 0$ holds, where $\|\cdot\|_2$ represents the L2-norm operation. Pareto stationarity is a necessary condition for a policy to be Pareto-optimal (Désidéri, 2012). As will be introduced in the next subsection, the Pareto-ascent direction is exploited to find the Pareto-stationary policy.

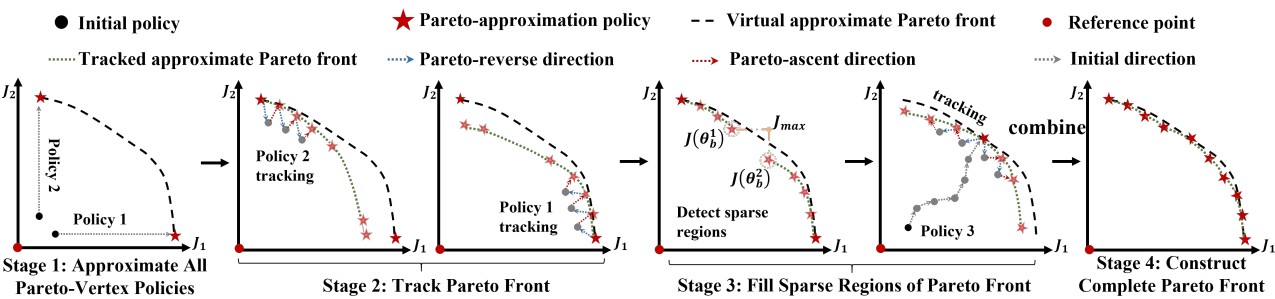

*Figure 1.* MPFT framework overview ($m = 2$, $K = 1$, $u = 1$, $v = 1$).

## 3.4. Pareto-Ascent Direction

The Pareto-ascent direction can be derived by solving the following problem:

$$(P2): \quad \min_{\boldsymbol{\alpha} \in \mathbb{R}^m} \quad \left\| \nabla_{\boldsymbol{\theta}} \boldsymbol{J}(\boldsymbol{\theta})^\top \boldsymbol{\alpha} \right\|_2^2,$$
$$\text{s.t.} \quad \alpha_i \geq 0, \|\boldsymbol{\alpha}\|_1 = 1, \forall i \in \{1, \ldots, m\},$$

where $\boldsymbol{\alpha} = [\alpha_1, \ldots, \alpha_m]^\top$. If $\left\| \nabla_{\boldsymbol{\theta}} \boldsymbol{J}(\boldsymbol{\theta})^\top \boldsymbol{\alpha}^* \right\|_2^2 > 0$, where $\boldsymbol{\alpha}^*$ is the optimal solution to problem (P2), the Pareto-ascent direction is obtained as $\nabla_{\boldsymbol{\theta}} \boldsymbol{J}(\boldsymbol{\theta})^\top \boldsymbol{\alpha}^*$. When the policy $\pi_{\boldsymbol{\theta}}$ is updated along the Pareto-ascent direction, all the objectives obtain the same amount of improvement, since the gradient vectors of different objectives project equally in the Pareto-ascent direction (Hu & Luo, 2024). If $\left\| \nabla_{\boldsymbol{\theta}} \boldsymbol{J}(\boldsymbol{\theta})^\top \boldsymbol{\alpha}^* \right\|_2^2 = 0$, the corresponding policy $\pi_{\boldsymbol{\theta}}$ is Pareto-stationary at $\boldsymbol{\omega} = \boldsymbol{\alpha}^*$. Further, based on (Zhou et al., 2024), if problem (P1) is convex, the Pareto-stationary $\pi_{\boldsymbol{\theta}}$ is also a Pareto-optimal policy; and if problem (P1) is non-convex, we say that the Pareto-stationary $\pi_{\boldsymbol{\theta}}$ is a Pareto-approximation policy. Please refer to Appendix B.1 for solving problem (P2).

## 4. MPFT Framework

In this section, we propose the Multi-policy Pareto Front Tracking (MPFT) framework based on the designed Pareto-tracking mechanism.

### 4.1. Pareto-Vertex and Pareto-Interior Policies

We divide the Pareto-optimal policies into two types, namely Pareto-vertex policies and Pareto-interior policies, according to their roles on the Pareto front.

A Pareto-vertex policy corresponds to an extreme single-objective solution. Specifically, for objective $i \in \{1, \ldots, m\}$, when the weight vector is set to $\omega_i = 1$ and $\omega_j = 0$ for all $j \neq i$, the optimal solution to problem (P1) is referred to as the Pareto-vertex policy associated with objective $i$. These policies represent boundary solutions obtained by exclusively optimizing individual objectives,

and they serve as anchor points of the Pareto front rather than geometric vertices.

In contrast, a Pareto-interior policy refers to any Pareto-optimal policy that is not a Pareto-vertex policy. Such a policy reflects a trade-off among multiple objectives and can be obtained by solving problem (P1) with a weight vector $\boldsymbol{\omega}$ in which at least two elements are strictly positive.

In complex environments, obtaining exact Pareto-vertex and Pareto-interior policies is often intractable. Therefore, in the following subsections, we focus on learning approximate Pareto-vertex policies and approximate Pareto-interior policies.

### 4.2. MPFT Framework Overview

The goal of the proposed MPFT framework is to find the Pareto-approximation policy set, denoted by $\mathcal{F}$, together with its mapping $\mathcal{N}$ to the objective space, which lies tightly close to the Pareto front.

As illustrated in Figure 1, the MPFT framework consists of four stages, and Figure 1 further visualizes the population-free training process in the bi-objective case, where only three policies are kept self-updating independently to track an approximate Pareto front:

1) *Stage 1: Approximate Pareto-vertex policies.* For each objective $i$, $\forall i \in \{1, \ldots, m\}$, by continuously training in the direction of the gradient $\nabla_{\boldsymbol{\theta}} \boldsymbol{J}(\boldsymbol{\theta})^\top \boldsymbol{\omega}$, where $\omega_i = 1$ and $\omega_j = 0$, $\forall j \in \{1, \ldots, m\}$ and $j \neq i$, we obtain the approximate Pareto-vertex policy $\pi_{\boldsymbol{\theta}^{i,*}}$.

2) *Stage 2: Track the Pareto front.* Starting from each $\pi_{\boldsymbol{\theta}^{i,*}}$, we design the Pareto-tracking mechanism to track the Pareto front, where the tracked policies form the Pareto-edge policy set[2], denoted by $\mathcal{F}_{edge}^i$, $i \in \{1, \ldots, m\}$. By deleting the dominated policies from the sets $\{\mathcal{F}_{edge}^i\}_{i \in \{1, \ldots, m\}}$, we obtain the initial $\mathcal{F} = \cup_{i=1}^{+m} \{\mathcal{F}_{edge}^i\}$ and its corresponding

---

[2]The term "Pareto-edge" denotes the $m$-dimensional segment induced by any two adjacent preference vectors, not a geometric edge.

tracked Pareto front $\mathcal{N}$, where $\cup^+$ is the combined operation of the set union and the deletion of the dominated policies.

3) *Stage 3: Fill sparse regions of the tracked Pareto front $\mathcal{N}$.* We focus on the top-$K$ sparse regions in $\mathcal{N}$, and find the $K$ approximate Pareto-interior policies $\{\pi_{\boldsymbol{\theta}^{k,\star}}\}_{k\in\{1,\dots,K\}}$. Then starting from each $\pi_{\boldsymbol{\theta}^{k,\star}}$, we continuously updating the policies to track the Pareto front. All the tracked policies form the $k$-th Pareto-interior policy tracking set, denoted by $\mathcal{F}_{inter}^k$, and are used to fill the $k$-th sparse region.

4) *Stage 4: Construct the complete $\mathcal{F}$.* The complete $\mathcal{F}$ is obtained as $\mathcal{F}\cup^+\left\{\cup_{k=1}^{+K}\{\mathcal{F}_{inter}^k\}\right\}$.

Since the Pareto tracking mechanism performs local updates, one trajectory cannot move across disconnected regions of the Pareto front. Thus, MPFT launches multiple tracks, each starting from different policies to explore distinct local components. The four stages design inherently accommodates non-connected or discrete Pareto fronts. The Stage 1 and Stage 4 are straightforward, we detail Stages 2 and 3 in the following subsections, respectively.

### 4.3. Stage 2: Track Pareto front

Following (Li et al., 2025), the Pareto front is continuous under the topology of the policy parameter space for MOMDP with a continuous parameterized policy class, in the sense that sufficiently small perturbations in the policy parameters induce arbitrarily small changes in the objective vector. Therefore, starting from a high-quality approximate Pareto policy, only a small number of updates are typically required to reach a nearby Pareto solution. Enlightened by this, Stage 2 constructs $m$ parallel tracks of the Pareto front, by sequentially generating the Pareto-edge policies $\{\pi_{\boldsymbol{\theta}^{i,(l)}}\}$, where $\pi_{\boldsymbol{\theta}^{i,(l+1)}}$ is updated from $\pi_{\boldsymbol{\theta}^{i,(l)}}$ in each track. For each $\pi_{\boldsymbol{\theta}^{i,\star}}$, a total of $\Psi_i$ training episodes are conducted. We also initialize $\pi_{\boldsymbol{\theta}^{i,(0)}}=\pi_{\boldsymbol{\theta}^{i,\star}}$ and $\mathcal{F}_{edge}^i=\{\pi_{\boldsymbol{\theta}^{i,\star}}\}$.

We propose the *Pareto-tracking mechanism* to guide the policy updating. Specifically, we newly define the Pareto-reverse direction $\nabla_{\boldsymbol{\theta}}\boldsymbol{J}(\boldsymbol{\theta})^{\top}\boldsymbol{\omega}$ of objective $i$ as that leads to value increase of all $\{J_j(\boldsymbol{\theta})\}_{j\in\{1,\dots,m\},i\neq j}$ except $J_i(\boldsymbol{\theta})$. The Pareto-reverse direction of objective $i$, $\forall i\in\{1,\dots,m\}$, can be derived by solving problem (P2) with an extra constraint with $\alpha_i=0$. By intentionally sacrificing one objective, the Pareto-reverse update provides a controllable policy updating direction that drives the policy away from the Pareto-vertex region and along the Pareto front, thereby enabling traversal of the Pareto front. To obtain $\pi_{\boldsymbol{\theta}^{i,(l+1)}}$, $\pi_{\boldsymbol{\theta}^{i,(l)}}$ is first trained along the Pareto-reverse direction for $u\geq 0$ consecutive episodes, and then along the Pareto-ascent direction for $v\geq 0$ consecutive episodes[3]. After obtaining $\pi_{\boldsymbol{\theta}^{i,(l+1)}}$, $\mathcal{F}_{edge}^i$ is updated as

$\mathcal{F}_{edge}^i\cup^+\{\pi_{\boldsymbol{\theta}^{i,(l+1)}}\}$. We also initialize $\mathcal{F}$ as $\cup_{i=1}^{+m}\{\mathcal{F}_{edge}^i\}$ by combining $m$ Pareto front tracks.

In Appendix C, we give a proof of exponential convergence of the Pareto-tracking mechanism. As shown in our convergence analysis, the Pareto-tracking mechanism converges exponentially toward a neighborhood of the Pareto-stationary set in any arbitrary objective dimension. Full coverage of the Pareto front is achieved empirically via the multi-track procedure in Stage 2 and Stage 3.

### 4.4. Stage 3: Fill Sparse Regions of $\mathcal{N}$

Sparse regions in the approximated Pareto front naturally arise due to the potentially complex geometric structure of the Pareto front, and are difficult to track by solely using Stage 2. Stage 3 is designed to address this challenge.

We focus on the top-$K$ sparse regions, and propose the objective weight adjustment method to find $K$ approximate Pareto-interior policies $\{\pi_{\boldsymbol{\theta}^{k,\star}}\}_{k\in\{1,\dots,K\}}$, whose mapping to the objective space are located in the top-$K$ sparse regions of $\mathcal{N}$, respectively. Unlike existing heuristic methods such as (Hu & Luo, 2024) or (Liu et al., 2025b), which directly select policies from the archive $\mathcal{F}$, our approach retrains $K$ new approximate Pareto-interior policies to avoid inheriting the coverage gaps produced in Stage 2. This ensures that Stage 3 can effectively fill sparse or disconnected regions of the Pareto front.

Specifically, we represent a sparse region by its $m$ boundary policies, denoted as $\{\pi_{\boldsymbol{\theta}_b^i}\}_{i=1}^m$, whose mappings in the objective space form the bounding simplex that characterizes the location and extent of the sparse region. For the bi-objective case shown in Figure 1, the sparse region is represented by two boundary policies. Please refer to Appendix B.2 for more details on how to find the boundary policies. We denote by $\max^{\mathrm{v}}(\boldsymbol{x}_1,\dots,\boldsymbol{x}_m)$ the element-wise maximum among $\boldsymbol{x}_i$, $i\in\{1,\dots,m\}$, and by $\mathrm{D}[\boldsymbol{x},\boldsymbol{y}]$ the element-wise division of $\boldsymbol{x}$ by $\boldsymbol{y}$. The following introduces the case with $K=1$ to highlight the key steps of the objective weight adjustment method:

1) Find $\{\pi_{\boldsymbol{\theta}_b^i}\}_{i=1}^m$, and calculate $\boldsymbol{J}_{max}=\max^{\mathrm{v}}\left(\boldsymbol{J}(\boldsymbol{\theta}_b^1),\dots,\boldsymbol{J}(\boldsymbol{\theta}_b^m)\right)$. Initialize a random policy $\pi_{\boldsymbol{\theta}}$;

2) Find $\boldsymbol{\beta}=\mathrm{D}[\boldsymbol{J}_{\max},\boldsymbol{J}(\boldsymbol{\theta})]$. Let weight $\boldsymbol{\omega}=\frac{\boldsymbol{\beta}}{\|\boldsymbol{\beta}\|_1}$[4];

---

[3]The term $u+v$ represents the number of episodes used in one policy-tracking process and is generally orders of magnitude

smaller than the episodes required for one policy-evolution process in evolutionary framework. As supported by the theoretical analysis in Appendix C and the empirical validation in Appendix G.4, $v$ must be greater than $u$. Ensuring a higher proportion of Pareto-ascent episodes allow the policy to self-correct from suboptimal initializations and stably climb toward the true frontier before tracking.

[4]$J_i(\boldsymbol{\theta})\geq 0$ can be easily guaranteed in reward design for each objective $i\in\{1,\dots,m\}$ to assure non-negative weights.

---

**Algorithm 1** MPFT algorithm

---

**Input**: episodes $\{\Xi_i\}_{i=1}^m$, $\{\Psi_i\}_{i=1}^m$, $\{\Xi_k\}_{k=1}^K$, $\{\Psi_k\}_{k=1}^K$, $u$, $v$, and timestep steps.

**Initialize**: $\mathcal{F}_{edge}^i = \emptyset$, $\mathcal{F}_{inter}^k = \emptyset$, $\pi_{\boldsymbol{\theta}^i}$, and $\pi_{\boldsymbol{\theta}^k}$, $\forall i \in \{1, \ldots, m\}, \forall k \in \{1, \ldots, K\}$.

1: Stage 1: Follow Section 4.2 to continuously train $\pi_{\boldsymbol{\theta}^i}$ for $\Xi_i$ episodes to get $\pi_{\boldsymbol{\theta}^i,*}$. Initialize $\pi_{\boldsymbol{\theta}^i,(l=0)} = \pi_{\boldsymbol{\theta}^i,*}$ and $\mathcal{F}_{edge}^i = \{\pi_{\boldsymbol{\theta}^i,*}\}$;

2: Stage 2: Apply the Pareto-tracking mechanism in Section 4.3 to parallely train each $\pi_{\boldsymbol{\theta}^i,(l)}$ for $\Psi_i$ episodes. Sequentially update the Pareto-edge policy set $\mathcal{F}_{edge}^i = \mathcal{F}_{edge}^i \cup^+ \{\pi_{\boldsymbol{\theta}^i,(l+1)}\}$ for $\frac{\Psi_i}{u+v}$ times. Obtain $\mathcal{F} = \mathcal{F} \cup^+ \left\{ \cup_{i=1}^{+m} \{\mathcal{F}_{edge}^i\} \right\}$;

3: Stage 3: Follow Section 4.4 to continuously train $\pi_{\boldsymbol{\theta}^k}$ for $\Xi_k$ episodes based on the objective weight adjustment method to get $\pi_{\boldsymbol{\theta}^k,\star}$. Initialize $\pi_{\boldsymbol{\theta}^k,(l=0)} = \pi_{\boldsymbol{\theta}^k,\star}$ and $\mathcal{F}_{inter}^k = \{\pi_{\boldsymbol{\theta}^k,\star}\}$. Use the Pareto-tracking mechanism to continuously update $\mathcal{F}_{inter}^k$ for $\frac{\Psi_k}{u+v}$ times;

4: Stage 4: Complete $\mathcal{F} = \mathcal{F} \cup^+ \left\{ \cup_{k=1}^{+K} \{\mathcal{F}_{inter}^k\} \right\}$;

**Output**: $\mathcal{F}$.

---

3) Update $\pi_{\boldsymbol{\theta}}$ along $\nabla_{\boldsymbol{\theta}} \boldsymbol{J}(\boldsymbol{\theta})^\top \frac{\boldsymbol{\beta}}{\|\boldsymbol{\beta}\|_1}$ for one episode, resulting in a new policy $\pi_{\boldsymbol{\theta}'}$;

4) Let $\pi_{\boldsymbol{\theta}} = \pi_{\boldsymbol{\theta}'}$, and repeat steps 2) and 3) until $\|\boldsymbol{J}(\boldsymbol{\theta}) - \boldsymbol{J}_{max}\|_2 \le \epsilon$ or reaching the maximum episode.

Figure 1 shows $\boldsymbol{J}_{max}$ when $m = 2$ and $K = 1$. By the above method, we can find the approximate Pareto-interior policy $\pi_{\boldsymbol{\theta}^1,\star}$ for $K = 1$, where the weight $\boldsymbol{\omega} = \frac{\boldsymbol{\beta}}{\|\boldsymbol{\beta}\|_1}$ in step 2) ensures that the update direction of the policy $\pi_{\boldsymbol{\theta}}$ is between the objectives $\{\boldsymbol{J}(\boldsymbol{\theta}^i)\}_{i=1}^m$. For the more complicated case with $K > 1$ as well as the methods to find the boundary policies and the sparse regions, please refer to Appendix B.2. Next, as illustrated in Figure 1, by using the Pareto-tracking mechanism in Stage 2 to track toward each of the $m$ objectives, where each $\pi_{\boldsymbol{\theta}^k,\star}$ is the start point of the tracking, we can obtain the $K$ approximate Pareto-interior policy tracking sets $\mathcal{F}_{inter}^k$, $k \in \{1, ..., K\}$, parallely. The top-$K$ sparse regions from $\mathcal{N}$ are filled by $\{\mathcal{F}_{inter}^k\}_{k \in \{1,...,K\}}$.

Finally, by combining all the tracked policies in Stages 2 and 3, we complete $\mathcal{F} = \mathcal{F} \cup^+ \left\{ \cup_{k=1}^{+K} \{\mathcal{F}_{inter}^k\} \right\}$ in Stage 4, where the corresponding $\mathcal{N}$ approximates the Pareto front tightly. The MPFT algorithm is presented in Algorithm 1, where each episode includes steps timesteps of policy training. Let $\Upsilon$ represent a fixed upper bound on the cost of training a single policy, independent of $m$. As analyzed in Appendix D.4, the time complexity of Algorithm 1 is $T_{\text{MPFT}} = \mathcal{O}\left( \Upsilon \times \text{steps} \times \left( \sum_{i=1}^m (\Xi_i + \Psi_i) + \sum_{k=1}^K (\Xi_k + \Psi_k) \right) \right)$, which scales linearly with the number of objectives $m$.

## 5. Multi-objective Online and Offline RL

This section takes the offline TD7 (Fujimoto et al., 2023) as an example and incorporates it into our proposed MPFT framework. Other typical RL algorithms, such as the online PPO algorithm (Schulman et al., 2017) and the offline SAC algorithm (Haarnoja et al., 2018), can also be incorporated similarly, as given in Appendices B.3 and B.4, respectively. All these online and offline RL algorithms can be applied for the policy training in each episode of the MPFT algorithm.

The TD7 algorithm, as an improved version of the TD3 algorithm (Fujimoto et al., 2018), is one of the SOTA offline RL algorithms. In TD7 algorithm, the high-dimensional state embedding vector $z_s$ and state-action embedding vector $z_{sa}$, are used to improve the sample efficiency. To the best our knowledge, there is no existing multi-objective TD7 algorithms. In the following, we propose multi-objective TD7 (MOTD7) for the MPFT framework. Specifically, for the MOTD7, $\nabla_{\boldsymbol{\theta}} J_i(\boldsymbol{\theta})$ in (2) is obtained as $\nabla_{\boldsymbol{\theta}} J_i(\boldsymbol{\theta}) = -\nabla_{\boldsymbol{\theta}} \mathbb{E}_{\mathcal{B} \sim \mathcal{D}, s_t \leftarrow \mathcal{B}, a_t' = \pi_{\boldsymbol{\theta}}^{(z_s)}} [\mathbf{Loss}_i(s_t, a_t')] = \nabla_{\boldsymbol{\theta}} \mathbb{E} \left[ \frac{1}{|\mathcal{B}|} \sum_{t=1}^{|\mathcal{B}|} Q_i^{(z_s, z_{sa})}(s_t, a_t') \right]$, where $\leftarrow$ indicates an extract operation, $\mathcal{D}$ is the replay buffer, $\mathcal{B}$ is a mini-batch, $\pi_{\boldsymbol{\theta}}^{(z_s)}$ is the policy with $z_s$ embedded, $Q_i^{(z_s, z_{sa})}(s_j, a_j)$ is the Q-value of objective $i$ with $z_s$ and $z_{sa}$ embedded. For notation simplicity, we omit $z_s$ and $z_{sa}$, and use $\pi_{\boldsymbol{\theta}}$ and $Q_i$ to represent $\pi_{\boldsymbol{\theta}}^{(z_s)}$ and $Q_i^{(z_s, z_{sa})}$, respectively, the gradient direction of the policy $\pi_{\boldsymbol{\theta}}$ under a given weight $\boldsymbol{\omega}$ is: $\nabla_{\boldsymbol{\theta}} \boldsymbol{J}(\boldsymbol{\theta})^\top \boldsymbol{\omega} = \mathbb{E} \left[ \frac{1}{|\mathcal{B}|} \sum_{t=1}^{|\mathcal{B}|} \nabla_{\boldsymbol{\theta}} \boldsymbol{Q}^{\boldsymbol{\omega}}(s_t, a_t') \right]$, where $\boldsymbol{Q}^{\boldsymbol{\omega}}(s_t, a_t') = \boldsymbol{\omega}^\top \boldsymbol{Q}(s_t, a_t')$ is the weighted Q-value scalar, with $\boldsymbol{Q} = [Q_1, ..., Q_m]^\top$. Details of MOTD7 are provided in Appendix B.5. It is straightforward to incorporate MOTD7 into the proposed MPFT framework: one can apply Algorithm 5 in Appendix B.5 to policy training of each episode in Algorithm 1.

## 6. Experiments

In our experiments, we adopt three widely used metrics to evaluate the quality of the Pareto-approximation front: hypervolume (HV), sparsity (SP), and expected utility (EU). HV reflects the convergence and distribution of the Pareto-approximation front (Falcón-Cardona et al., 2022), SP measures its density (Hu & Luo, 2024), and EU evaluates expected utility under different sampling preferences (Liu et al., 2025b). Let **env_steps** denote the total number of agent–environment interactions, equivalent to the total training steps across all policies. As shown in Table 1, each method is allocated an **env_steps** budget sufficient for convergence, and comparisons are based on their *converged performance*. Convergence under the respective budgets is verified in Figures 7 and 8 in Appendix G.3. Notably, MPFT requires fewer **env_steps** than all benchmarks across all

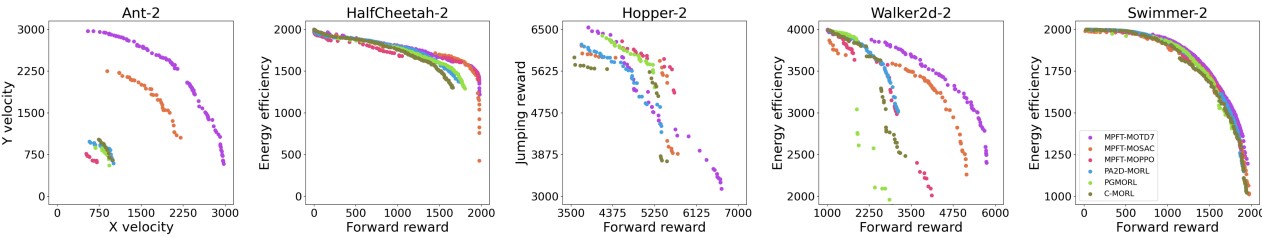

*Figure 2.* Comparison of Pareto front approximations. For each environment, we show the best random seed for each algorithm.

environments. Detailed calculations of HV, SP, EU, and **env_steps** are provided in Appendix D.

### 6.1. Simulation Environment

The existing offline MORL datasets (Zhu et al., 2023; Xue et al., 2024) are constructed using PGMORL (Xu et al., 2020). Because the PGMORL is prone to local optima, these datasets often consist of suboptimal trajectories, making them inadequate for evaluating the MPFT framework. To demonstrate the compatibility of our framework in offline settings, we provide additional D4MORL evaluations in Appendix G.6. Moreover, since MOPPO is also not applicable to the offline dataset, we build our own robot control environment, which is more complicated than those in (Xu et al., 2020; Hu & Luo, 2024; Liu et al., 2025b).

Our main simulation environment is built by Mujuco-3.3.0 (Todorov et al., 2012) and the v5 version of the game in Gymnasium-1.1.1. According to the official documentation (Farama Foundation, 2025), the v5 version fixes the bug in the previous version that the robot can still obtain rewards when it is in an unhealthy state, and increases the difficulty of robot control. Based on the v5 version, we set up six robot control environments with continuous action spaces, which are HalfCheetah-2, Hopper-2, Swimmer-2, Ant-2, Walker2d-2, and Hopper-3, respectively. Except the three-objective environment Hopper-3, the others are all the two-objective environments. Additionally, to validate MPFT performance in discrete and higher-dimensional tasks, we constructed three environments identical to C-MORL (Liu et al., 2025b): the discrete 4-objectives environment Lunar-Lander-4 (LL-4), the discrete 6-objectives environment Fruit-Tree-6 (FT-6), and the continuous 9-objectives environment Building-9 (B-9). For detailed settings, please refer to Appendix F.1.

### 6.2. Benchmarks

To verify that the proposed MPFT framework can support to both online RL and offline RL, we implement the MOPPO, the MOSAC, and the MOTD7 in Section 5 under the MPFT framework, which give MPFT-MOPPO, MPFT-MOSAC, and MPFT-MOTD7, respectively. We also con-

sider three evolutionary framework based MP-MORL algorithms as main benchmarks **PGMORL** (Xu et al., 2020), **PA2D-MORL** (Hu & Luo, 2024), and **C-MORL** (Liu et al., 2025b). For a detailed comparison between MPFT and this methods, please refer to Appendix E.

To ensure fairness, all evaluations and comparisons are implemented in the same environment. The PGMORL, the PA2D-MORL, the C-MORL and the MPFT-MOPPO use the online MOPPO algorithm and have the same network structure. We conduct the PGMORL based on the code repository (https://github.com/mit-gfx/PGMORL) provided by (Xu et al., 2020) and the C-MORL based on the code repository (https://github.com/RuohLiuq/C-MORL) provided by (Liu et al., 2025b). Since (Hu & Luo, 2024) did not open the codes, we implement PA2D-MORL exactly as described in (Hu & Luo, 2024) and put it with our codes online available[5]. Additionally, to show the actual performance of all the algorithms, we do not use the policy interpolation to fill in the sparse regions of the Pareto front. All baselines and MPFT are set with the same policy-buffer size, i.e., 200 for 2-objective environments and 300 for $\geq$3-objective environments. For details on the benchmark and MPFT parameter settings, please refer to Appendix F.2.

### 6.3. Results

In experiments, all the MPFT framework based MP-MORL algorithms, we consider the top-1 sparse region with $K = 1$ if not otherwise specified.

Table 1 gives the evaluation results for all $\leq$3-objective environments, which shows that our proposed MPFT-MOTD7 framework can achieve better HV and EU performance with less **env_steps**. Specifically, **env_steps** of the MPFT-MOTD7, the MPFT-MOSAC, and the MPFT-PPO average decrease by 78.22%, 72.04%, and 66.27%, respectively, as compared to the evolutionary framework based algorithms PGMORL, PA2D-MORL and C-MORL. It is also observed that in some environments our proposed MPFT

---

[5] https://github.com/zzy2048/MPFT-Framework

*Table 1.* Evaluation of HV, SP, EU, and **env_steps** for $\leq$3-objective environments. All data are based on five independent runs.

| Env. | Metrics | MPFT-MOTD7 | MPFT-MOSAC | MPFT-MOPPO | PGMORL | PA2D-MORL | C-MORL |
|---|---|---|---|---|---|---|---|
| Walker2d-2 | HV $\uparrow (\times 10^7)$ | **2.04 $\pm$ 0.10** | 1.72 $\pm$ 0.22 | 1.37 $\pm$ 0.03 | 1.11 $\pm$ 0.25 | 1.31 $\pm$ 0.09 | 1.21 $\pm$ 0.16 |
| | SP $\downarrow (\times 10^4)$ | 1.13 $\pm$ 0.53 | 1.52 $\pm$ 0.24 | 2.75 $\pm$ 0.56 | 2.02 $\pm$ 1.05 | **0.32 $\pm$ 0.13** | 0.57 $\pm$ 0.29 |
| | EU $\uparrow (\times 10^3)$ | **4.44 $\pm$ 0.06** | 4.01 $\pm$ 0.33 | 3.51 $\pm$ 0.02 | 3.25 $\pm$ 0.32 | 3.47 $\pm$ 0.15 | 3.29 $\pm$ 0.24 |
| | env_steps $\downarrow$ | **11,200,000** | 16,400,000 | 17,612,800 | 41,943,040 | 41,943,040 | 41,943,040 |
| HalfCheetah-2 | HV $\uparrow (\times 10^6)$ | **3.51 $\pm$ 0.01** | 3.25 $\pm$ 0.18 | 3.10 $\pm$ 0.26 | 3.04 $\pm$ 0.21 | 3.09 $\pm$ 0.17 | 2.93 $\pm$ 0.34 |
| | SP $\downarrow (\times 10^4)$ | 0.24 $\pm$ 0.13 | 1.48 $\pm$ 0.64 | 0.25 $\pm$ 0.17 | 0.13 $\pm$ 0.11 | 0.03 $\pm$ 0.01 | **0.02 $\pm$ 0.01** |
| | EU $\uparrow (\times 10^3)$ | **1.78 $\pm$ 0.01** | 1.72 $\pm$ 0.02 | 1.68 $\pm$ 0.17 | 1.66 $\pm$ 0.01 | 1.67 $\pm$ 0.16 | 1.61 $\pm$ 0.12 |
| | env_steps $\downarrow$ | **5,000,000** | 7,800,000 | 13,516,800 | 41,943,040 | 41,943,040 | 41,943,040 |
| Hopper-2 | HV $\uparrow (\times 10^7)$ | **3.68 $\pm$ 0.27** | 3.47 $\pm$ 0.44 | 3.37 $\pm$ 0.24 | 3.17 $\pm$ 0.11 | 3.18 $\pm$ 0.05 | 3.16 $\pm$ 0.24 |
| | SP $\downarrow (\times 10^4)$ | **1.55 $\pm$ 0.55** | 12.0 $\pm$ 5.96 | 2.66 $\pm$ 0.37 | 2.36 $\pm$ 1.68 | 2.28 $\pm$ 0.67 | 18.9 $\pm$ 4.17 |
| | EU $\uparrow (\times 10^3)$ | **5.85 $\pm$ 0.15** | 5.66 $\pm$ 0.49 | 5.44 $\pm$ 0.23 | 5.34 $\pm$ 0.21 | 5.39 $\pm$ 0.05 | 5.36 $\pm$ 0.37 |
| | env_steps $\downarrow$ | **19,200,000** | 22,800,000 | 27,033,600 | 85,196,800 | 85,196,800 | 85,196,800 |
| Ant-2 | HV $\uparrow (\times 10^6)$ | **7.04 $\pm$ 0.19** | 3.75 $\pm$ 0.82 | 0.43 $\pm$ 0.11 | 0.68 $\pm$ 0.17 | 0.87 $\pm$ 0.06 | 0.94 $\pm$ 0.35 |
| | SP $\downarrow (\times 10^3)$ | **1.18 $\pm$ 0.11** | 5.63 $\pm$ 2.21 | 1.32 $\pm$ 0.09 | 2.32 $\pm$ 1.00 | 1.36 $\pm$ 0.35 | 1.86 $\pm$ 0.38 |
| | EU $\uparrow (\times 10^3)$ | **2.47 $\pm$ 0.04** | 1.77 $\pm$ 0.25 | 0.54 $\pm$ 0.10 | 0.79 $\pm$ 0.10 | 0.88 $\pm$ 0.11 | 0.92 $\pm$ 0.14 |
| | env_steps $\downarrow$ | **19,200,000** | 22,800,000 | 24,576,000 | 85,196,800 | 85,196,800 | 85,196,800 |
| Swimmer-2 | HV $\uparrow (\times 10^6)$ | **3.54 $\pm$ 0.07** | 3.52 $\pm$ 0.07 | 3.49 $\pm$ 0.01 | 3.46 $\pm$ 0.08 | 3.51 $\pm$ 0.00 | 3.52 $\pm$ 0.04 |
| | SP $\downarrow (\times 10^3)$ | 0.81 $\pm$ 0.36 | 0.81 $\pm$ 0.15 | 0.33 $\pm$ 0.13 | 0.64 $\pm$ 0.30 | **0.19 $\pm$ 0.03** | 0.36 $\pm$ 0.09 |
| | EU $\uparrow (\times 10^3)$ | **1.75 $\pm$ 0.02** | 1.74 $\pm$ 0.01 | 1.74 $\pm$ 0.00 | 1.74 $\pm$ 0.01 | 1.74 $\pm$ 0.00 | 1.74 $\pm$ 0.01 |
| | env_steps $\downarrow$ | **3,800,000** | 5,400,000 | 5,529,600 | 20,971,520 | 20,971,520 | 20,971,520 |
| Hopper-3 | HV $\uparrow (\times 10^{10})$ | **9.61 $\pm$ 0.09** | 8.54 $\pm$ 0.34 | 7.58 $\pm$ 0.50 | 7.62 $\pm$ 0.80 | 6.73 $\pm$ 0.35 | 7.19 $\pm$ 0.82 |
| | SP $\downarrow (\times 10^4)$ | **0.32 $\pm$ 0.11** | 2.03 $\pm$ 0.44 | 1.62 $\pm$ 0.40 | 4.85 $\pm$ 0.58 | 3.00 $\pm$ 0.58 | 3.28 $\pm$ 0.60 |
| | EU $\uparrow (\times 10^3)$ | **4.44 $\pm$ 0.05** | 4.37 $\pm$ 0.08 | 4.28 $\pm$ 0.04 | 4.30 $\pm$ 0.08 | 4.21 $\pm$ 0.06 | 4.27 $\pm$ 0.07 |
| | env_steps $\downarrow$ | **39,000,000** | 41,600,000 | 55,705,600 | 135,168,000 | 135,168,000 | 127,795,200 |

*Table 2.* MPFT-MOTD7 with different $K$.

| $K$ | Walker2d-2 | | HalfCheetah-2 | |
|---|---|---|---|---|
| | HV $\uparrow (\times 10^7)$ | SP $\downarrow (\times 10^4)$ | HV $\uparrow (\times 10^6)$ | SP $\downarrow (\times 10^3)$ |
| 0 | 1.95 $\pm$ 0.09 | 2.09 $\pm$ 0.99 | 3.47 $\pm$ 0.04 | 5.95 $\pm$ 5.42 |
| 1 | 2.04 $\pm$ 0.10 | 1.13 $\pm$ 0.53 | 3.51 $\pm$ 0.01 | 2.38 $\pm$ 1.30 |
| 3 | 2.15 $\pm$ 0.06 | 1.09 $\pm$ 0.28 | 3.52 $\pm$ 0.01 | 1.10 $\pm$ 0.25 |
| 6 | 2.20 $\pm$ 0.01 | 0.89 $\pm$ 0.13 | 3.54 $\pm$ 0.00 | 0.63 $\pm$ 0.04 |

framework does not achieve optimal SP performance, but this is a problem that cannot be avoided due to the reduced **env_steps** or achieving the same level of policy diversity as the benchmarks. This problem can be alleviated, however, by applying more advanced RL algorithms, such as the TD7 algorithm.

Figure 2 shows the Pareto front of all the algorithms in all 2-objective environments. It shows that our proposed MPFT framework can better explore policies whose mappings to the objective space locate almost at the edge of the Pareto front; moreover, our MPFT-MOTD7 obtains the best Pareto-approximating policy set among all the algorithms.

Table 2 further shows the ablation results of the MPFT-MOTD7 in the two-objective environments Walker2d-2 and HalfCheetah-2. Specifically, for different values of $K$, both the mean and standard deviation of the HV and SP metrics are reported. All data are based on five independent runs. From Table 2, we generally observe an increased HV value and a reduced SP value as $K$ increases, and the performance is lowest when $K = 0$, i.e., without Stage 3. This validates that the proposed Stage 3 can effectively fill the sparse re-

gions of the tracked Pareto front in Stage 2. However, larger $K$ values require more agent-environment interactions, and therefore trade-offs usually need to be made in practice.

*Table 3.* Evaluation of HV, SP, EU, and **env_steps** for $>3$-objective environments. All data are based on five independent runs. T/O indicates that the trainingtime exceeded the maximum limit of 100 hours. N/A indicates not applicable.

| Env. | Metrics | Envelope | GPI-LS | C-MORL | Our (SPFT) |
|---|---|---|---|---|---|
| FT-6 | HV$\uparrow (\times 10^4)$ | 3.66$\pm$0.23 | 3.57$\pm$0.05 | 3.67$\pm$0.14 | **5.02$\pm$0.09** |
| | SP$\downarrow (\times 10^{-1})$ | 5.46$\pm$0.15 | 5.29$\pm$0.21 | **0.42$\pm$0.03** | 0.50$\pm$0.18 |
| | EU$\uparrow (\times 10^0)$ | 6.15$\pm$0.00 | 6.15$\pm$0.00 | 6.53$\pm$0.00 | **6.59$\pm$0.08** |
| | env_steps $\downarrow$ | 500,000 | 500,000 | 500,000 | **67,200** |
| LL-4 | HV$\uparrow (\times 10^9)$ | 0.43$\pm$0.18 | 1.06$\pm$0.16 | 1.12$\pm$0.03 | **1.24$\pm$0.11** |
| | SP$\downarrow (\times 10^3)$ | 0.19$\pm$0.16 | **0.13$\pm$0.01** | 1.04$\pm$0.24 | 1.75$\pm$0.56 |
| | EU$\uparrow (\times 10^1)$ | -2.84$\pm$4.06 | 1.69$\pm$0.34 | **2.35$\pm$0.18** | 2.36$\pm$0.29 |
| | env_steps $\downarrow$ | 500,000 | 500,000 | 500,000 | **480,000** |
| B-9 | HV$\uparrow (\times 10^{31})$ | N/A | T/O | 7.93$\pm$0.07 | **8.22$\pm$0.10** |
| | SP$\downarrow (\times 10^3)$ | N/A | T/O | 2.79$\pm$0.40 | **1.92$\pm$0.63** |
| | EU$\uparrow (\times 10^3)$ | N/A | T/O | 3.50$\pm$0.00 | **3.53$\pm$0.00** |
| | env_steps $\downarrow$ | N/A | 2,500,000 | 2,500,000 | **630,000** |

To further evaluate MPFT in environments with more than three objectives and in discrete-action settings, we compare MPFT-MOTD7 against **GPI-LS** (Alegre et al., 2023), **Envelope** (Yang et al., 2019) and C-MORL across the LL-4, FT-6, and B-9 benchmarks[6]. To highlight the efficiency of the proposed Pareto-tracking mechanism in tracking high-dimensional Pareto front, we intentionally disable *multiple policy tracking* and sparse region filling, retaining only a *single policy tracking* trajectory (e.g., tracking solely from

---

[6]Note that the original MOTD7 does not support discrete action settings. We adapted it for discrete versions using one-hot encoding.

one policy). This variant is called Single-policy Pareto Front Tracking (SPFT). The hyperparameter configuration is reported in Appendix F.2. Envelope, GPI-LS, and C-MORL share the same **env_steps** budget, which is larger than the budget allocated to SPFT.

Table 3 gives the evaluation results for all $>3$-objectives environments, which demonstrate that SPFT achieves best performance on both HV and EU metrics with fewer agent-environment interactions, indicating the efficiency of the Pareto-tracking mechanism in tracing high-dimensional Pareto front. Additional results are reported in Appendix G, and further discussions and limitations of the proposed MPFT framework are presented in Appendix H.

## 7. Conclusion

We propose the MPFT framework to efficiently learn a high-quality Pareto-approximation policy set. MPFT achieves linear-time complexity and enables controllable traversal along the Pareto front via a Pareto-tracking mechanism, without population evolution and policy selection. Extensive experiments demonstrate that MPFT attains SOTA hypervolume and expected utility, and significantly reduced agent–environment interactions. The framework is readily compatible with advanced offline RL algorithms, making it scalable to sampling-challenging multi-objective tasks.

## Acknowledgements

The authors want to thank the anonymous reviewers for their helpful comments and suggestions. This work was supported in part by the Key Research and Development Program in Xinjiang Uygur Autonomous Region under Grant 2025B04019-001, in part by the National Natural Science Foundation of China under Grant 62572329, in part by the Guangdong Basic and Applied Basic Research Foundation under Grant 2025A1515011557, Grant 2026A1515050003, and Grant 2025A1515010125, and in part by the Shenzhen Science and Technology Program under Grant JCYJ20240813142301003 and Grant JCYJ20230808105906014.

## Impact Statement

Real-world applications of our Multi-policy Pareto Front Tracking framework are broad among various fields. One typical example is robotic control, where our framework provides a sample-efficient architecture with theoretical convergence guarantees. More applications include autonomous driving, recommendation systems, dynamic pricing, etc. In industry, related methods on multi-policy MORL have been proposed and applied with different evolutionary focuses in the last few years, while our work focuses more on an efficient, population-free tracking mechanism. There can be potential societal consequences of our work, but none we feel must be specifically highlighted here.

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

# Appendixes

## A. Key Notations

| Notation | Description |
|---|---|
| $\pi_{\boldsymbol{\theta}}$ | Policy parameterized by $\boldsymbol{\theta}$ |
| $\boldsymbol{J}(\boldsymbol{\theta})$ | Multi-objective expected-return vector of $\pi_{\boldsymbol{\theta}}$ |
| $J_i(\boldsymbol{\theta})$ | Expected return for the $i$-th objective |
| $\nabla_{\boldsymbol{\theta}} J_i(\boldsymbol{\theta})$ | Policy-gradient vector for the $i$-th objective |
| $\boldsymbol{\omega}, \boldsymbol{\alpha}$ | Weight vector over objectives, element of the simplex $\Delta_m$ |
| $\pi_{\boldsymbol{\theta}^{i,*}}$ | Pareto-vertex policy of objective $i$ |
| $\pi_{\boldsymbol{\theta}^{k,\star}}$ | Pareto-interior policy in $k$-th sparse region |
| $\mathcal{F}$ | Pareto-approximating policy set |
| $\mathcal{F}_{edge}^i$ | Pareto-edge policy set tracked from $\pi_{\boldsymbol{\theta}^{i,*}}$ |
| $\mathcal{F}_{inter}^k$ | Pareto-interior policy tracking set tracked from $\pi_{\boldsymbol{\theta}^{k,\star}}$ |
| $\mathcal{N}$ | The mapping of $\mathcal{F}$ in the objective space (approximated Pareto front) |
| $K$ | Number of top-$K$ sparse regions |
| $u, v$ | Training episode number for Pareto-reverse and Pareto-ascent phase |
| $\Xi_i, \Psi_i$ | Training episodes for $\pi_{\boldsymbol{\theta}^{i,*}}$ and $\mathcal{F}_{edge}^i$ |
| $\Xi_k, \Psi_k$ | Training episodes for $\pi_{\boldsymbol{\theta}^{k,\star}}$ and $\mathcal{F}_{inter}^k$ |

## B. Algorithm Detail

### B.1. Pareto-Ascent Direction

For problem (P2), the stochastic gradient descent can be used to find the $\boldsymbol{\alpha}^*$. Specifically, for the constraints, the projection function $\mathcal{P}_{ro}(\cdot)$ can be used with $\boldsymbol{\alpha}^{t+1} = \mathcal{P}_{ro}(\boldsymbol{\alpha}^t - \eta \nabla_{\boldsymbol{\alpha}} f(\boldsymbol{\alpha}^t))$, where $f(\boldsymbol{\alpha}) = \left\| \nabla_{\boldsymbol{\theta}} \boldsymbol{J}(\boldsymbol{\theta})^\top \boldsymbol{\alpha} \right\|_2^2$ and $\eta$ is the step size. For the case with two objectives, i.e., $m = 2$, we can find the analytical solution of $\boldsymbol{\alpha}^* = [\alpha_1^*, \alpha_2^*]^\top$, which is either orthogonal to the difference of the two gradient vectors $\nabla_{\boldsymbol{\theta}} J_1(\boldsymbol{\theta})$ and $\nabla_{\boldsymbol{\theta}} J_2(\boldsymbol{\theta})$ or coincides with one of the gradient vectors, i.e.,

$$\alpha_1^* = \max\left( \min\left( \frac{(\nabla_{\boldsymbol{\theta}} J_2(\boldsymbol{\theta}) - \nabla_{\boldsymbol{\theta}} J_1(\boldsymbol{\theta}))^\top \nabla_{\boldsymbol{\theta}} J_2(\boldsymbol{\theta})}{\|\nabla_{\boldsymbol{\theta}} J_2(\boldsymbol{\theta}) - \nabla_{\boldsymbol{\theta}} J_1(\boldsymbol{\theta})\|_2^2}, 1 \right), 0 \right),$$

other is $\alpha_2^* = 1 - \alpha_1^*$. The Pareto-ascent direction is therefore found as $\nabla_{\boldsymbol{\theta}} \boldsymbol{J}(\boldsymbol{\theta})^\top \boldsymbol{\alpha}^*$ for the policy $\pi_{\boldsymbol{\theta}}$.

### B.2. Finding Boundary Policies for Top-$K$ Sparse Regions

To find the top-$K$ sparse regions from $\mathcal{N}$, we can first view the solutions in $\mathcal{N}$ as points in an $m$-dimensional ($m$-D) space. We focus on the case with $m \leq 3$, as shown in Algorithm 2.

When $m = 2$, the algorithm proceeds as follows: 1) Based on the "non-dominated" property of the solutions in $\mathcal{N}$, we can sort each point in ascending order according to the value of objective 1; 2) Calculate the Euclidean distance between each adjacent pair of points; 3) Select the top-$K$ pairs of points with the largest distances; 4) The $K$ point pairs represent the top-$K$ sparse regions, and the boundaries of these $K$ sparse regions are returned as $\{\boldsymbol{J}_{max}^1, \ldots, \boldsymbol{J}_{max}^K\}$.

When $m = 3$, the points in $\mathcal{N}$ form a convex surface in 3-D space (Li et al., 2025). To identify sparse regions on the surface, we develop the following algorithm: 1) Use principal component analysis (PCA) to project the 3-D point set $\mathcal{N}$ onto the best-fitting 2-D plane; 2) Use Delaunay triangulation to construct a triangular network on the 2-D plane; 3) Calculate the area of each triangle in the triangular network and select the top-$K$ triangles with the largest areas; 4) Map the selected top-$K$ triangles back to the 3-D space; 5) Use the vertices of these $K$ triangles to represent the sparse regions and return the boundaries of these $K$ sparse regions $\{\boldsymbol{J}_{max}^1, ..., \boldsymbol{J}_{max}^K\}$.

Moreover, although not the focus of this paper, for higher dimensions with $m > 3$, we can use PCA dimension reduction to reduce it to 3-D, and then use the above algorithm to obtain the boundaries of the top $K$ sparse regions. In addition, we can

---

**Algorithm 2** Boundary Detection of Top-$K$ Sparse Regions

---

1: **Input** $\texttt{arr} = \mathcal{N} \in \mathbb{R}^{n \times m}$: A set of $n$ solutions in $m$-dimensional objective space (Pareto Front);
   $K$: Number of sparse regions to detect;
2: **Ensure** $\texttt{SparseRegions}$: Set of detected sparse regions (line segments for $m = 2$, triangles for $m = 3$);
3: Initialize $\texttt{SparseRegions}$ as empty set;
4: **if** $m == 2$ **then**
5:     Sort $\texttt{arr}$ by the first objective value;
6:     Compute pairwise Euclidean distances between adjacent points:
       $d_i = \|\texttt{arr}[i+1] - \texttt{arr}[i]\|_2$ for $i = 0, \cdots, n-2$;
7:     Select the indices of top-$K$ largest distances;
8:     **for** each selected index $i$ **do**
9:         Append segment $(\texttt{arr}[i], \texttt{arr}[i+1])$ to $\texttt{SparseRegions}$;
10:     **end for**
11: **else if** $m == 3$ **then**
12:     Project 3D points onto a best-fit 2-D plane using PCA:
13:     $\texttt{arr\_proj}, \texttt{pca} \leftarrow \text{ProjectToPlane}(\texttt{arr})$
14:     Perform Delaunay triangulation on projected 2-D points;
15:     For each triangle, compute its area;
16:     Select top-$K$ triangles with largest areas;
17:     Map selected triangles back to original 3-D space using $\texttt{pca}$;
18:     Add them to $\texttt{SparseRegions}$;
19: **end if**
20: Caculate the boundaries $\{\boldsymbol{J}_{max}^1, \ldots, \boldsymbol{J}_{max}^K\}$ of the $\texttt{SparseRegions} \in \mathbb{R}^{K \times m \times m}$
21: **Output** $\{\boldsymbol{J}_{max}^1, \ldots, \boldsymbol{J}_{max}^K\}$

---

also apply the crowding distance defined in (Liu et al., 2025b) to find the top-$K$ solutions that lie in the sparse region of the case with $m > 3$. However, both methods are heuristic, and neither has a formal proof.

### B.3. MOPPO Algorithm

Most existing MP-MORL research are based on the PPO algorithm (Schulman et al., 2017), primarily because PPO supports large-scale parallel training and aligns well with the evolutionary framework used in existing MP-MORL algorithms. Although the evolutionary framework can fully explore more policies on the Pareto front to obtain a denser Pareto policy set, this is feasible only in simulation, while in real applications, each policy needs to interact with the environment and adopt its own trajectory for training, which is obviously impractical. We design an MOPPO algorithm that can be applied within our proposed MPFT framework.

Unlike the traditional single-objective PPO algorithm, the expected return in MOPPO is a weighted sum of multiple objectives. To avoid relearning the state value function when $\boldsymbol{\omega}$ changes, we need to define a vectorized state value function $\boldsymbol{V}(s; \boldsymbol{\phi}) = [V_1(s; \boldsymbol{\phi}), ..., V_m(s; \boldsymbol{\phi})]^\top \in \mathbb{R}^m$ to evaluate each objective $i \in \{1, ..., m\}$'s state value, where $\boldsymbol{\phi}$ is parameter of $\boldsymbol{V}$. $\boldsymbol{V}(s; \boldsymbol{\phi})$ is defined as

$$\boldsymbol{V}(s; \boldsymbol{\phi}) = \mathbb{E}_{a_t \sim \pi_{\boldsymbol{\theta}}} \left[ \sum_{t=0}^{T} \gamma^t \boldsymbol{R}(s_t, a_t) \mathbf{Done}_t \big| s_0 = s \right],$$

which is updated following Bellman equation. $\mathbf{Done}_t \in \{0, 1\}$ is the termination condition, with $\mathbf{Done}_t = 0$ if the task terminates at timestep $t$, or $\mathbf{Done}_t = 1$, otherwise. Note that, although the $\mathbf{Done}_t = 1$ variable already denotes task completion, a finite maximal horizon $T$ is retained in practice to avoid pathological non-termination cases and to align with standard RL environments. The theoretical analysis uses $T = \infty$, whereas the finite $T$ in implementation serves only as a practical safeguard

$$\mathcal{T}^\pi \boldsymbol{V}(s_t; \boldsymbol{\phi}) = \hat{\boldsymbol{V}}(s_t) = \sum_{a_t} \pi_{\boldsymbol{\theta}}(a_t | s_t) \left[ \boldsymbol{R}(s_t, a_t) + \gamma \boldsymbol{V}(s_{t+1}; \boldsymbol{\phi}) \right],$$

where $\mathcal{T}^\pi$ is the Bellman backup operator. For each objective $i$, the policy gradient with advantage estimation (Schulman et al., 2015) is defined as $\nabla_{\boldsymbol{\theta}} J_i(\boldsymbol{\theta}) = -\nabla \mathbb{E}_{s \sim e_{\boldsymbol{\theta}}, a \sim \pi_{\boldsymbol{\theta}}}[\mathbf{Loss}_i(s_t, a_t)] = \mathbb{E}\left[\sum_{t=0}^{T} A_i(s_t, a_t) \nabla_{\boldsymbol{\theta}} \log \pi_{\boldsymbol{\theta}}(a_t | s_t) | s_0 = s\right],$

where $e_{\boldsymbol{\theta}}$ is the state distribution based on policy $\pi_{\boldsymbol{\theta}}$, $A_i(s_t, a_t) \in \boldsymbol{A}(s_t, a_t)$ is the advantage function for objective $i$, and $\boldsymbol{A}(s_t, a_t) = [A_1(s_t, a_t), ..., A_m(s_t, a_t)]^\top \in \mathbb{R}^m$. The gradient direction of the policy $\pi_{\boldsymbol{\theta}}$ under a given weight $\boldsymbol{\omega}$ is

$$
\begin{aligned}
\boldsymbol{\omega}^\top \nabla_{\boldsymbol{\theta}} \boldsymbol{J}(\boldsymbol{\theta}) &= \sum_{i=1}^m w_i \nabla_{\boldsymbol{\theta}} J_i(\boldsymbol{\theta}) \\
&= \mathbb{E}\left[\sum_{t=0}^T \boldsymbol{\omega}^\top \boldsymbol{A}(s_t, a_t) \nabla_{\boldsymbol{\theta}} \log \pi_{\boldsymbol{\theta}}(a_t|s_t)\right] \\
&= \mathbb{E}\left[\boldsymbol{A}^{\boldsymbol{\omega}}(s_t, a_t) \nabla_{\boldsymbol{\theta}} \log \pi_{\boldsymbol{\theta}}(a_t|s_t)\right],
\end{aligned}
$$

where $\boldsymbol{A}^{\boldsymbol{\omega}}(a_t, s_t) = \boldsymbol{\omega}^\top \boldsymbol{A}(a_t, s_t)$ is the weighted advantage scalar. Integrating the above into PPO algorithm, we obtain the MOPPO policy training process:

1) Collect trajectories (i.e., rollout) using the current policy $\pi_{\boldsymbol{\theta}}$.

2) Calculate the advantage by the GAE approach in (Schulman et al., 2015), $\boldsymbol{A}^{\boldsymbol{\omega}}(s_t, a_t) = \boldsymbol{\omega}^\top \sum_{l=0}^\infty (\gamma\lambda)^l \boldsymbol{\delta}_{t+l+1}$ with $\boldsymbol{\delta}_t = \boldsymbol{R}(s_t, a_t) + \gamma \boldsymbol{V}(s_{t+1}; \boldsymbol{\phi}) - \boldsymbol{V}(s_t; \boldsymbol{\phi})$, and $t \in \{1, ..., T\}$ is the timestep.

3) Update the policy $\pi_{\boldsymbol{\theta}}$ by minimizing the clipped surrogate loss $L^\pi(\boldsymbol{\theta}) = -\mathbb{E}[\min(r_t(\boldsymbol{\theta})\boldsymbol{A}^{\boldsymbol{\omega}}(s_t, a_t), \texttt{clip}(r_t(\boldsymbol{\theta}), 1-\epsilon, 1+\epsilon)\boldsymbol{A}^{\boldsymbol{\omega}}(s_t, a_t))]$, where $r_t(\boldsymbol{\theta}) = \frac{\pi_{\boldsymbol{\theta}}(a_t|s_t)}{\pi_{old}(a_t|s_t)}$, $\epsilon \in [0, 1)$, $\texttt{clip}(x, 1-\epsilon, 1+\epsilon)$ is a clip function make the value of $x$ lies in the range of $[1-\epsilon, 1+\epsilon]$, and $\pi_{old}$ is the old policy.

4) Update the state value function $\boldsymbol{V}(s_t; \boldsymbol{\phi})$ by the MSE Loss $L^V(\boldsymbol{\phi}) = \mathbb{E}\left[\boldsymbol{V}(s_t; \boldsymbol{\phi}) - \hat{\boldsymbol{V}}(s_t)\right]$.

5) Repeat steps 3) and 4) until reaching the maximum epoch.

In each MPFT-MOPPO episode, the policy training process is shown in Algorithm 3.

---

**Algorithm 3** MOPPO Algorithm

---

1: **Input** $\pi_{\boldsymbol{\theta}}$, and $\boldsymbol{V}(s; \boldsymbol{\phi})$;
2: **Obtain** $\boldsymbol{\omega}$ based on the Pareto-tracking mechanism;
3: **Fixed** initialize $\epsilon$, $T$, and epoch;
4: Rollout: Run policy $\pi_{old}$ in environment for steps timesteps to collect trajectories. (Agent interacts with the environment steps times);
5: Compute advantage: $\{\boldsymbol{A}^{\boldsymbol{\omega}}(s_t, a_t)\}_{t=1}^T$;
6: **for** $e \leftarrow 1, \cdots, \text{epoch}$ **do**
7:   Update the policy: Use loss function $L^\pi$ to update $\boldsymbol{\theta}$;
8:   Update the state value network: Use loss function $L^V(\boldsymbol{\phi})$ to update $\boldsymbol{\phi}$;
9: **end for**
10: **Output**: $\pi_{\boldsymbol{\theta}}$ and $\boldsymbol{V}(s; \boldsymbol{\phi})$;

---

### B.4. MOSAC Algorithm

The SAC algorithm is an offline RL algorithm based on entropy maximization (Haarnoja et al., 2018). Its output, similar to PPO, is a probability distribution over actions.

Unlike MOPPO, MOSAC has a state-action value (Q-value) network $\boldsymbol{Q}(s, a; \boldsymbol{\varphi}) = [Q_1(s, a; \boldsymbol{\varphi}), \ldots, Q_m(s, a; \boldsymbol{\varphi})]^\top \in \mathbb{R}^m$ in addition to $\boldsymbol{V}(s; \boldsymbol{\phi}) = [V_1(s; \boldsymbol{\phi}), \ldots, V_m(s; \boldsymbol{\phi})]^\top \in \mathbb{R}^m$, where $\boldsymbol{\varphi}$ are the parameters of $\boldsymbol{Q}(s, a; \boldsymbol{\varphi})$. The update of $\boldsymbol{Q}(s, a; \boldsymbol{\varphi})$ is given by the following Bellman equation:

$$
\mathcal{T}^\pi \boldsymbol{Q}(s_t, a_t; \boldsymbol{\varphi}) = \hat{\boldsymbol{Q}}(s_t, a_t) = \mathbb{E}_{\mathcal{B} \sim \mathcal{D}, (\boldsymbol{R}(s_t, a_t), s_{t+1}, \textbf{Done}_t) \leftarrow \mathcal{B}}\left[\boldsymbol{R}(s_t, a_t) + \gamma \boldsymbol{V}(s_{t+1}; \boldsymbol{\phi})\textbf{Done}_t\right],
$$

where $\leftarrow$ indicates extract operation, $\mathcal{D}$ is the replay buffer, $\mathcal{B}$ is a mini-batch, and $\boldsymbol{V}(s_t; \boldsymbol{\phi}) = \mathbb{E}_{s_t \leftarrow \mathcal{B}, a_t' \sim \pi_{\boldsymbol{\theta}}}\left[\boldsymbol{Q}(s_t, a_t'; \boldsymbol{\varphi}) - \log \pi_{\boldsymbol{\theta}}(a_t'|s_t)\right]$ is the soft state value function.

To prevent overestimation of the Q-value, we used two state-action value networks, $\mathbf{Q}_1(s, a; \boldsymbol{\varphi}_1)$ and $\mathbf{Q}_2(s, a; \boldsymbol{\varphi}_2)$ to estimate the minimum Q-value $\mathbf{Q}_{est}(s, a) = \min\left(\{\mathbf{Q}_j(s, a; \boldsymbol{\varphi}_j)\}_{j=1}^2\right) \in \mathbb{R}^m$. The policy gradient of objective $i \in \{1, ..., m\}$ is defined as: $\nabla_{\boldsymbol{\theta}} J_i(\boldsymbol{\theta}) = -\nabla_{\boldsymbol{\theta}} \mathbb{E}_{\mathcal{B} \sim \mathcal{D}, s_t \leftarrow \mathcal{B}, a'_t \sim \pi_{\boldsymbol{\theta}}} [\mathbf{Loss}_i(s_t, a'_t)] = \mathbb{E}_{s_t \leftarrow \mathcal{B}} \left[ \frac{1}{|\mathcal{B}|} \sum_{t=1}^{|\mathcal{B}|} \kappa_i \nabla_{\boldsymbol{\theta}} \log \pi_{\boldsymbol{\theta}}(a'_t|s_t) - \nabla_{\boldsymbol{\theta}} Q_i(s_t, a'_t) \right]$, where $\kappa_i$ is the temperature parameter for the entropy regularizer of objective $i$, and $Q_i(s_t, a'_t)$ is the $i$-th element of $\mathbf{Q}_{est}(s_t, a'_t)$. The gradient direction of the policy $\pi_{\boldsymbol{\theta}}$ under a given weight $\boldsymbol{\omega}$ is:

$$\nabla_{\boldsymbol{\theta}} \mathbf{J}(\boldsymbol{\theta})^\top \boldsymbol{\omega} = -\nabla_{\boldsymbol{\theta}} \boldsymbol{\omega}^\top \mathbb{E}_{s_t \leftarrow \mathcal{B}} \left[ \frac{1}{|\mathcal{B}|} \sum_{t=1}^{|\mathcal{B}|} \log \pi_{\boldsymbol{\theta}}(\pi_{\boldsymbol{\theta}}(s_t)|s_t) \boldsymbol{\kappa} - \mathbf{Q}(s_t, \pi_{\boldsymbol{\theta}}(s_t)) \right]$$

$$= -\mathbb{E} \left[ \frac{1}{|\mathcal{B}|} \sum_{t=1}^{|\mathcal{B}|} \boldsymbol{\kappa}^{\boldsymbol{\omega}} \nabla_{\boldsymbol{\theta}} \log \pi_{\boldsymbol{\theta}}(\pi_{\boldsymbol{\theta}}(s_t)|s_t) - \nabla_{\boldsymbol{\theta}} \mathbf{Q}^{\boldsymbol{\omega}}(s_t, \pi_{\boldsymbol{\theta}}(s_t)) \right]$$

$$= -\mathbb{E} \left[ \frac{1}{|\mathcal{B}|} \sum_{t=1}^{|\mathcal{B}|} \boldsymbol{\kappa}^{\boldsymbol{\omega}} \nabla_{\boldsymbol{\theta}} \log \pi_{\boldsymbol{\theta}}(a'_t|s_t) - \nabla_{\boldsymbol{\theta}} \mathbf{Q}^{\boldsymbol{\omega}}(s_t, a'_t) \right]$$

where $\boldsymbol{\kappa}^{\boldsymbol{\omega}} = \boldsymbol{\omega}^\top \boldsymbol{\kappa}$ and $\mathbf{Q}^{\boldsymbol{\omega}}(s_t, a') = \boldsymbol{\omega}^\top \mathbf{Q}_{est}(s_t, a')$ are weighted entropy temperature scalar and weighted Q-value scalar, respectively, with $\boldsymbol{\kappa} = [\kappa_1, ..., \kappa_m]^\top$.

---

**Algorithm 4** MOSAC Algorithm

---

1: **Input** $\pi_{\boldsymbol{\theta}}$, $\{\mathbf{Q}_j(s, a; \boldsymbol{\varphi}_j)\}_{j=1}^2$, $\mathbf{V}(s; \phi)$, and $\mathcal{D}$;
2: **Obtain** $\boldsymbol{\omega}$ based on the Pareto-tracking mechanism;
3: **Fixed** initialize steps and $\{\hat{\mathcal{H}}_i\}_{i=1}^m$;
4: Data Storage: Run policy $\pi_{\boldsymbol{\theta}}$ for steps timesteps in the environment to collect experiences, and store these experiences into $\mathcal{D}$. (Agent interacts with the environment steps times);
5: **for** $t \leftarrow 1, \cdots, $ steps **do**
6:    Update the policy: Use loss function $L^\pi$ to update $\boldsymbol{\theta}$;
7:    Update the state value network: Use loss function $L^V(\phi)$ to update $\phi$;
8:    Update the state-action value network: Use loss function $L_j^Q(\boldsymbol{\varphi}_j)$ to update $\boldsymbol{\varphi}_j, j \in \{1, 2\}$;
9:    Update the temperature: Use loss function $L^\kappa(\boldsymbol{\kappa})$ to update $\boldsymbol{\kappa}$;
10: **end for**
11: **Output**: $\pi_{\boldsymbol{\theta}}$, $\{\mathbf{Q}_j(s, a; \boldsymbol{\varphi}_j)\}_{j=1}^2$, $\mathbf{V}(s; \phi)$, and $\mathcal{D}$;

---

The following is the multi-objective loss function for all networks:

1) State value network loss:

$$L^V(\phi) = \mathbb{E}_{s_t \leftarrow \mathcal{B}, a'_t \sim \pi_{\boldsymbol{\theta}}} \left[ \frac{1}{|\mathcal{B}|} \sum_{t=1}^{|\mathcal{B}|} \left\| \mathbf{V}(s_t; \phi) - \left( \mathbf{Q}_{est}(s_t, a'_t) - \nabla_{\boldsymbol{\theta}} \log \pi_{\boldsymbol{\theta}}(a'_t|s_t) \cdot \boldsymbol{\kappa} \right) \right\|_2^2 \right].$$

2) State-action value network loss:

$$L_j^Q(\boldsymbol{\varphi}_j) = \mathbb{E}_{(s_t, a_t, \mathbf{R}(s_t, a_t), s_{t+1}) \leftarrow \mathcal{B}} \left[ \frac{1}{|\mathcal{B}|} \sum_{t=1}^{|\mathcal{B}|} \left\| \mathbf{Q}_j(s_t, a_t; \boldsymbol{\varphi}_j) - \hat{\mathbf{Q}}(s_t, a_t) \right\|_2^2 \right], j \in \{1, 2\}.$$

3) Policy loss:

$$L^\pi(\boldsymbol{\theta}) = \mathbb{E}_{s_t \leftarrow \mathcal{B}, a'_t \sim \pi_{\boldsymbol{\theta}}} \left[ \frac{1}{|\mathcal{B}|} \sum_{t=1}^{|\mathcal{B}|} \boldsymbol{\kappa}^{\boldsymbol{\omega}} \log \pi_{\boldsymbol{\theta}}(a'_t|s_t) - \boldsymbol{\omega}^\top \mathbf{Q}_{est}(s_t, a'_t) \right].$$

4) Entropy temperature loss:

$$L^{\kappa}(\boldsymbol{\kappa}) = \mathbb{E}_{s_t \leftarrow \mathcal{B}, a'_t \sim \pi_{\boldsymbol{\theta}}} \left[ -\frac{1}{|\mathcal{B}| \cdot m} \sum_{t=1}^{|\mathcal{B}|} \sum_{i=1}^{m} \left( \log \kappa_i \cdot \left( \log \pi_{\boldsymbol{\theta}}(a'_t|s_t) + \hat{\mathcal{H}}_i \right) \right) \right],$$

where $\hat{\mathcal{H}}_i$ is the target entropy of the objective $i$, it is usually set to $-|\mathcal{A}|$.

In each MPFT-MOSAC episode, the policy training process is shown in Algorithm 4, where the `Data Storage` operation can be omitted if the offline dataset $\mathcal{D}$ is given and the amount of data is sufficient.

### B.5. MOTD7 Algorithm

The TD7 algorithm (Fujimoto et al., 2023) introduces four enhancements to the TD3 algorithm (Fujimoto et al., 2018), which are the state-action learned embeddings (SALE), the Loss-Adjusted Prioritized (LAP) replay (Fujimoto et al., 2020), the checkpoint, and the behavior cloning. The SALE maps the low-dimensional state and state-action into high-dimensional embedding vectors $z_s$ and $z_{sa}$, which enhance the inputs to the value function and policy to improve sample efficiency and performance. The LAP is a prioritized experience replay technique that prioritizes samples in the replay buffer based on their importance (e.g. TD error). During training, higher-priority samples are selected first, which helps RL algorithms learn important experiences faster and improve learning efficiency. Checkpoint maintains training stability. Sample efficiency, learning efficiency, and training stability are critical for applying MORL algorithm to real-world tasks, thus SALE, LAP, and checkpoint are retained in our MOTD7 setting. The checkpoint is the same as the setting in (Fujimoto et al., 2023), and the SALE and LAP settings are as follows.

SALE uses two independent encoders $f(s)$ and $g(f(s), a)$ to encode the state $s$ and the state-action $(s, a)$ into $z_s$ and $z_{sa}$, respectively. Since $f(\cdot)$ and $g(\cdot)$ are decoupled, the encoders $f(\cdot)$ and $g(\cdot)$ can be jointly trained using the mean squared error (MSE) between the state-action embedding $z_{sa}^t = g(f(s_t), a_t)$ and the next state embedding $z_s^{t+1} = f(s_{t+1})$ with $(s_t, a_t, s_{t+1}) \leftarrow \mathcal{B}$ and $\mathcal{B} \sim \mathcal{D}$. The encoder loss function is defined as:

$$L^{enc}(f, g) = \frac{1}{|\mathcal{B}|} \sum_{t=1}^{|\mathcal{B}|} \|g(f(s_t), a_t) - |f(s_{t+1})|_{\times}\|_2^2,$$

where $| \cdot |_{\times}$ is the stop-gradient operation. Therefore, the original state-action value function $\boldsymbol{Q}(s, a; \boldsymbol{\varphi})$ becomes $\boldsymbol{Q}^{(z_s, z_{sa})}(s, a; \boldsymbol{\varphi}) \triangleq \boldsymbol{Q}(z_s, z_{sa}, s, a; \boldsymbol{\varphi})$ with $z_s$ and $z_{sa}$ embedded. The original policy $\pi_{\boldsymbol{\theta}}(s)$ becomes $\pi_{\boldsymbol{\theta}}^{(z_s)}(s) \triangleq \pi_{\boldsymbol{\theta}}(z_s, s)$ with $z_s$ and $z_{sa}$ embedded. For notation simplicity, we omit the symbols $z_s$ and $z_{sa}$, i.e., $\pi_{\boldsymbol{\theta}} = \pi_{\boldsymbol{\theta}}^{(z_s)}$, $\boldsymbol{Q}(s, a; \boldsymbol{\varphi}) = \boldsymbol{Q}^{(z_s, z_{sa})}(s, a; \boldsymbol{\varphi})$. In implementation, MOTD7 also uses two state-action value networks $\boldsymbol{Q}_1(s, a; \boldsymbol{\varphi}_1)$ and $\boldsymbol{Q}_2(s, a; \boldsymbol{\varphi}_2)$ to alleviate the impact of overestimating the Q-value. The Q-value is represented by the minimum value estimated by the two networks, i.e., $\boldsymbol{Q}_{est}(s, a) = \min \left( \{\boldsymbol{Q}_j(s, a; \boldsymbol{\varphi}_j)\}_{j=1}^{2} \right) \in \mathbb{R}^m$.

For the LAP, the probability of sampling a transition tuple $t := (s_t, a_t, \boldsymbol{R}(s_t, a_t), s_{t+1}, \mathbf{Done}_t)$ from the replay buffer $\mathcal{D}$ is defined as:

$$p(t) = \frac{\max \left( \mathtt{mean}(\boldsymbol{\delta}(t))^{\xi}, 1 \right)}{\sum_{t'=1}^{|\mathcal{D}|} \max \left( \mathtt{mean}(\boldsymbol{\delta}(t'))^{\xi}, 1 \right)},$$

where $\mathtt{mean}(\boldsymbol{x})$ is an operation to obtain the mean of all elements in $\boldsymbol{x}$, $\boldsymbol{\delta}(t) = \frac{\boldsymbol{\delta}_1(t) + \boldsymbol{\delta}_2(t)}{2}$, $\boldsymbol{\delta}_j(t) = \mathtt{abs} \left( \boldsymbol{Q}_j(s_t, a_t; \boldsymbol{\varphi}_j) - \hat{\boldsymbol{Q}}(s_t, a_t) \right) = [\delta_{j,1}(t), ..., \delta_{j,m}(t)]^{\top} \in \mathbb{R}_+^m$ is TD error of $\boldsymbol{Q}_j(\cdot)$ with $j \in \{1, 2\}$. $\mathtt{abs}(\boldsymbol{x})$ is an operation to take the element-wise absolute value in $\boldsymbol{x}$, and $\hat{\boldsymbol{Q}}(s_t, a_t) = \boldsymbol{R}(s_t, a_t) + \gamma \mathtt{clip} \left( \boldsymbol{Q}_{est}(s_{t+1}, \pi_{\boldsymbol{\theta}}(s_{t+1})), \boldsymbol{Q}_{min}, \boldsymbol{Q}_{max} \right)$ is target Q-value. $\boldsymbol{Q}_{min} = \min_{(s,a) \in \mathcal{D}} \boldsymbol{Q}_{est}(s, a) \in \mathbb{R}^m$ and $\boldsymbol{Q}_{max} = \max_{(s,a) \in \mathcal{D}} \boldsymbol{Q}_{est}(s, a) \in \mathbb{R}^m$ are the minimum and maximum Q-values that have been recorded during training. In order to reduce the effect of overestimation or underestimation of $\boldsymbol{Q}_{est}$ by a small number of samples in replay buffer $\mathcal{D}$, the value of $\boldsymbol{Q}_{est}$ will be restricted between $\boldsymbol{Q}_{min}$ and $\boldsymbol{Q}_{max}$.

The policy gradient of objective $i \in \{1, \dots, m\}$ is defined as: $\nabla_{\boldsymbol{\theta}} J_i(\boldsymbol{\theta}) = -\nabla_{\boldsymbol{\theta}} \mathbb{E}_{\mathcal{B}: \sim \mathcal{D}, s_t \leftarrow \mathcal{B}, a'_t = \pi_{\boldsymbol{\theta}}(s_t)} [\mathbf{Loss}_i(s_t, a'_t)] = -\mathbb{E} \left[ -\frac{1}{|\mathcal{B}|} \sum_{t=1}^{|\mathcal{B}|} Q_i(s_t, a'_t) \right]$, where $:\sim$ is the LAP sample operation, and $Q_i(s_t, a'_t)$ is the $i$-th element of the $\boldsymbol{Q}_{est}(s_t, a'_t)$.

The gradient direction of the policy $\pi_{\boldsymbol{\theta}}$ under a given weight $\boldsymbol{\omega}$ is:

$$\nabla_{\boldsymbol{\theta}} \boldsymbol{J}(\boldsymbol{\theta})^{\top} \boldsymbol{\omega} = -\nabla_{\boldsymbol{\theta}} \mathbb{E}_{\mathcal{B}:\sim\mathcal{D},s_t\leftarrow\mathcal{B}} \left[ -\frac{1}{|\mathcal{B}|} \sum_{t=1}^{|\mathcal{B}|} \boldsymbol{\omega}^{\top} \boldsymbol{Q}_{est}\left(s_t, \pi_{\boldsymbol{\theta}}(s_t)\right) \right]$$

$$= \mathbb{E} \left[ \frac{1}{|\mathcal{B}|} \sum_{t=1}^{|\mathcal{B}|} \nabla_{\boldsymbol{\theta}} \boldsymbol{Q}^{\boldsymbol{\omega}}\left(s_t, a'_t\right) \right]$$

where $\boldsymbol{Q}^{\boldsymbol{\omega}}\left(s_t, a'_t\right) = \boldsymbol{\omega}^{\top} \boldsymbol{Q}_{est}(s_t, a'_t)$.

The following are the multi-objective loss functions for all networks:

1) State-action value network loss: Instead of the MSE loss, we use the LAP Huber loss (Huber, 1992), which is defined as

$$L_j^Q(\boldsymbol{\varphi}_j) = \mathbb{E}_{\mathcal{B}:\sim\mathcal{D}} \left[ \frac{1}{|\mathcal{B}|} \sum_{t=1}^{|\mathcal{B}|} \texttt{mean}\left(\boldsymbol{\delta}_j^{LAP}(t)\right) \right], j \in \{1, 2\},$$

where $\boldsymbol{\delta}_j^{LAP}(t) = \left[\delta_{j,1}^{LAP}(t), ..., \delta_{j,m}^{LAP}(t)\right]^{\top}$, and $\delta_{j,i}^{LAP}(t)$ equals to $\frac{1}{2}\left(\delta_{j,i}(t)\right)^2$ if $\delta_{j,i}(t) < 1$ else $\delta_{j,i}(t)$, for all $i \in \{1, \ldots, m\}$.

2) Policy loss:

$$L^{\pi}(\boldsymbol{\theta}) = -\mathbb{E}_{\mathcal{B}:\sim\mathcal{D},s_t\leftarrow\mathcal{B}} \left[ \frac{1}{|\mathcal{B}|} \sum_{t=1}^{|\mathcal{B}|} \boldsymbol{\omega}^{\top} \boldsymbol{Q}_{est}\left(s_t, \pi_{\boldsymbol{\theta}}(s_t)\right) \right].$$

In each MPFT-MOTD7 episode, the policy training process is shown in Algorithm 5, where the `Data Storage` operation can be omitted if the offline dataset $\mathcal{D}$ is given and the amount of data is sufficient.

---

**Algorithm 5** MOTD7 Algorithm

---

1: **Input** $\pi_{\boldsymbol{\theta}}$, $\{\boldsymbol{Q}_j(s, a; \boldsymbol{\varphi}_j)\}_{j=1}^2$, $(f(s), g(z_s, a))$, and $\mathcal{D}$;
2: **Obtain** $\boldsymbol{\omega}$ based on the Pareto-tracking mechanism;
3: **Fixed** initialize steps;
4: `Data Storage`: Run policy $\pi_{\boldsymbol{\theta}}$ for steps timesteps in the environment to collect the experiences, and store these experiences into the LAP $\mathcal{D}$. (Agent interacts with the environment steps times);
5: **for** t $\leftarrow 1, \cdots,$ steps **do**
6:     `Update the policy`: Use loss function $L^{\pi}$ to updates $\boldsymbol{\theta}$;
7:     `Update the state-action value network`: Use loss function $L_j^Q(\boldsymbol{\varphi}_j)$ to updates $\phi_j, j \in \{1, 2\}$;
8:     `Update the encoders`: Use loss function $L^{enc}(f, g)$ to updates $(f(s), g(z_s, a))$;
9: **end for**
10: **Output**: $\pi_{\boldsymbol{\theta}}$, $\{\boldsymbol{Q}_j(s, a; \boldsymbol{\varphi}_j)\}_{j=1}^2$, $(f(s), g(z_s, a))$, and $\mathcal{D}$;

---

## C. Convergence Proof of the Pareto-Tracking Mechanism

To isolate the Pareto-tracking mechanism from specific algorithmic details, we adopt a continuous-time ordinary differential equation (ODE) formulation and interpret discrete updates as Euler discretization. The alternating reverse/ascent dynamics are modeled as a switched system, which is approximated by the averaged vector field with duty cycle $\lambda$ (Sanders et al., 2007). To guarantee smoothness and uniqueness of the inner minimizer, we introduce a $\mu$-regularization based on the Moreau–Yosida envelope, ensuring differentiability and bounding the perturbation error by $\mathcal{O}(\mu)$. This enables us to define the Lyapunov function $V_{\mu}(\boldsymbol{\theta}) = \frac{1}{2}\|\nabla_{\boldsymbol{\theta}} \boldsymbol{J}(\boldsymbol{\theta})^{\top} \boldsymbol{\alpha}_{\mu}^*\|^2$, whose derivative can be analyzed via Danskin-type results. By combining projection inequalities, the directional curvature assumption, and Young's inequality, we derive a linear decay bound of the form $\dot{V}_{\mu}(\boldsymbol{\theta}) \leq -cV_{\mu}(\boldsymbol{\theta}) + C\mu$. Grönwall's inequality and Lyapunov theory then establish exponential convergence to an $\mathcal{O}(\mu)$ neighborhood. A first-order Euler discretization with bounded step size preserves this stability, while robustness results from averaging theory (Llibre et al., 2014) ensure that the switched system tracks the averaged

dynamics up to an $\mathcal{O}(\tau)$ error. Collectively, these results yield exponential convergence to an $\mathcal{O}(\mu + \tau)$ neighborhood of the Pareto-stationary set.

Note that, to avoid symbolic ambiguity, all symbol definitions introduced in this section are local and valid only within its context.

## C.1. Notations and Basic Assumptions

For the multi-objective optimization problem with $m$ objectives, the objective function vector $\boldsymbol{J}(\boldsymbol{\theta}) = [J_1(\boldsymbol{\theta}), \ldots, J_m(\boldsymbol{\theta})]^\top$ is determined by the parameters $\boldsymbol{\theta} \in \mathbb{R}^d$, where $d$ is the dimension of $\boldsymbol{\theta}$. We assume the parameter space $\Theta$ is convex and compact, and all objective functions $\{J_i(\boldsymbol{\theta})\}$ are second-order continuous differentiable, $i \in \{1, \ldots, m\}$. Since the gradient descent algorithm (GD) is used to update $\boldsymbol{\theta}$, our goal is to minimize the weighted sum $\boldsymbol{J}^\top \boldsymbol{\omega}$, where $\boldsymbol{\omega} = [\omega_1, \ldots, \omega_m]^\top \in \Delta_m$, and $\Delta_m := \{\boldsymbol{\omega} \in \mathbb{R}^m | \omega_i \geq 0, \|\boldsymbol{\omega}\|_1 = 1\}$ denotes the simplex. Note that $J_i(\boldsymbol{\theta})$ denotes any second-order continuous differentiable function, not solely the cumulative discounted reward. For maximization problems, it suffices to prefix it with a negative sign, which does not alter our conclusions.

Denote the gradient of each objective by $\nabla_{\boldsymbol{\theta}} J_i(\boldsymbol{\theta}) \in \mathbb{R}^d$, and the gradient matrix as

$$\boldsymbol{g}(\boldsymbol{\theta}) := \nabla_{\boldsymbol{\theta}} \boldsymbol{J}(\boldsymbol{\theta}) = \begin{bmatrix} \nabla_{\boldsymbol{\theta}} J_1(\boldsymbol{\theta})^\top \\ \vdots \\ \nabla_{\boldsymbol{\theta}} J_m(\boldsymbol{\theta})^\top \end{bmatrix} \in \mathbb{R}^{m \times d}, \quad g_i(\boldsymbol{\theta}) = \nabla_{\boldsymbol{\theta}} J_i(\boldsymbol{\theta}).$$

By compactness of $\Theta$ we obtain boundedness and smoothness constants:

$$G_{\max} := \sup_{\boldsymbol{\theta} \in \Theta} \sup_{i \in \{1, \ldots, m\}} \|\nabla J_i(\boldsymbol{\theta})\| < \infty,$$

and

$$H_{\max} := \sup_{\boldsymbol{\theta} \in \Theta} \sup_{i \in \{1, \ldots, m\}} \left\| \nabla^2 J_i(\boldsymbol{\theta}) \right\|_{\mathrm{op}} < \infty,$$

where $\|\cdot\|_{\mathrm{op}}$ is the operator norm of a matrix.

We aim to show that the Pareto-tracking mechanism (alternating Pareto-reverse and Pareto-ascent phases) converges exponentially to a neighborhood of the Pareto-stationary set, under appropriate duty-cycle and step-size conditions.

## C.2. Regularization and Differentiability

Since Pareto-ascent and Pareto-reverse directions rely on the solution of a minimum-norm combination

$$\boldsymbol{\alpha}_0^* = \arg \min_{\boldsymbol{\alpha} \in \Delta_m} \|\nabla_{\boldsymbol{\theta}} \boldsymbol{J}(\boldsymbol{\theta})^\top \boldsymbol{\alpha}\|^2,$$

the minimizer may not be unique everywhere, leading to non-differentiability. To obtain a smooth Lyapunov function, we introduce a quadratic regularization (Moreau-type):

$$\boldsymbol{\alpha}_\mu^* := \arg \min_{\boldsymbol{\alpha} \in \Delta_m} \|\nabla_{\boldsymbol{\theta}} \boldsymbol{J}(\boldsymbol{\theta})^\top \boldsymbol{\alpha}\|^2 + \frac{\mu}{2} \|\boldsymbol{\alpha}\|^2, \tag{3}$$

where $\mu > 0$ is a constant. This ensures a unique solution $\alpha_\mu$, and by Danskin's theorem the smoothed energy function

$$V_\mu(\boldsymbol{\theta}) = \tfrac{1}{2} \|\boldsymbol{v}_\mu(\boldsymbol{\theta})\|^2,$$

is differentiable, where $\boldsymbol{v}_\mu(\boldsymbol{\theta}) := \boldsymbol{g}(\boldsymbol{\theta})^\top \boldsymbol{\alpha}_\mu^* \in \mathbb{R}^d$. The gradient of $V_\mu(\boldsymbol{\theta})$ is defined as

$$\nabla_{\boldsymbol{\theta}} V_\mu(\boldsymbol{\theta}) = \nabla_{\boldsymbol{\theta}} \left( \tfrac{1}{2} \|\boldsymbol{g}(\boldsymbol{\theta})^\top \boldsymbol{\alpha}_\mu^*\|^2 \right) = \boldsymbol{H}_\mu(\boldsymbol{\theta}) \boldsymbol{v}_\mu(\boldsymbol{\theta}),$$

where $\boldsymbol{H}_\mu(\boldsymbol{\theta}) := \sum_{i=1}^m \boldsymbol{\alpha}_{\mu,i}^* \nabla^2 J_i(\boldsymbol{\theta})$ is the weighted Hessian matrix of $J_i(\boldsymbol{\theta})$.

*Remark* C.1. When $\mu \to 0$, the minimizer $\alpha_\mu(\boldsymbol{\theta})$ converges to the minimizer of the original problem. If the original problem is unique over all $\boldsymbol{\theta}$, no regularization is required.

In Appendix D.C.4, we quantify the projection error when $\mu > 0$, and show that when using $\mu$-regularization, the regularization introduces only a small perturbation, and the projection property is approximately preserved.

### C.3. System Model (Averaged System)

For each Pareto-vertex policy $\pi_{\boldsymbol{\theta}^{i,*}}$, the Pareto-tracking mechanism alternates between two phases:

**Phase 1 (Pareto-reverse):** move along the Pareto-reverse direction of the objective $i$, the objective weight $\boldsymbol{\omega}$ equals $\boldsymbol{\alpha}_\mu^{*,(i)} := \arg\min_{\boldsymbol{\alpha} \in \Delta_m, \boldsymbol{\alpha}_i = 0} \|\nabla_{\boldsymbol{\theta}} \boldsymbol{J}(\boldsymbol{\theta})^\top \boldsymbol{\alpha}\|^2 + \frac{\mu}{2}\|\boldsymbol{\alpha}\|$. The Pareto-reverse direction of objective $i$ is defined as

$$d_{\text{rev}}^{(i)}(\boldsymbol{\theta}) := -\boldsymbol{u}_\mu^{(i)}(\boldsymbol{\theta}) := -\boldsymbol{g}(\boldsymbol{\theta})^\top \boldsymbol{\alpha}_\mu^{*,(i)}.$$

**Phase 2 (Pareto-ascent):** move along the Pareto-ascent direction, which is defined as

$$d_{\text{asc}}(\boldsymbol{\theta}) := -\boldsymbol{v}_\mu(\boldsymbol{\theta}) := -\boldsymbol{g}(\boldsymbol{\theta})^\top \boldsymbol{\alpha}_\mu^*.$$

In practice these two phases alternate periodically. For analysis we first study the averaged system with duty-cycle $\lambda \in [0,1]$:

$$\dot{\boldsymbol{\theta}} = F_i(\boldsymbol{\theta}) := \lambda d_{\text{rev}}^{(i)}(\boldsymbol{\theta}) + (1-\lambda)d_{\text{asc}}(\boldsymbol{\theta}).$$

For any $\boldsymbol{\theta} \in \boldsymbol{\Theta}$ and $\mu > 0$, The derivative of $V_\mu(\boldsymbol{\theta})$ along the vector field $F_i(\boldsymbol{\theta})$ is

$$\dot{V}_\mu(\boldsymbol{\theta}) = \nabla_{\boldsymbol{\theta}} V_\mu(\boldsymbol{\theta})^\top \cdot F_i(\boldsymbol{\theta}) = -\boldsymbol{v}_\mu(\boldsymbol{\theta})^\top \boldsymbol{H}_\mu(\boldsymbol{\theta}) \left( \lambda \boldsymbol{u}_\mu^{(i)}(\boldsymbol{\theta}) + (1-\lambda)\boldsymbol{v}_\mu(\boldsymbol{\theta}) \right).$$

Let $\boldsymbol{v} := \boldsymbol{v}_\mu(\boldsymbol{\theta})$, $\boldsymbol{u} := \boldsymbol{u}_\mu^{(i)}(\boldsymbol{\theta})$, and $\boldsymbol{H} := \boldsymbol{H}_\mu(\boldsymbol{\theta})$. Then we have

$$\dot{V}_\mu(\boldsymbol{\theta}) = -\lambda \boldsymbol{v}^\top \boldsymbol{H} \boldsymbol{u} - (1-\lambda)\boldsymbol{v}^\top \boldsymbol{H} \boldsymbol{v} \tag{4}$$

### C.4. $\mu$-Perturbed Projection Approximation

Define the convex set for a fixed $\boldsymbol{\theta} \in \boldsymbol{\Theta}$ as

$$\mathcal{U}(\boldsymbol{\theta}) := \{\boldsymbol{g}(\boldsymbol{\theta})^\top \boldsymbol{\omega} : \boldsymbol{\omega} \in \Delta_m\} \subset \mathbb{R}^d.$$

For the original problem where $\mu = 0$, let $\boldsymbol{v}_0(\boldsymbol{\theta}) = \boldsymbol{g}(\boldsymbol{\theta})^\top \boldsymbol{\alpha}_0^*$ denote the minimally normed point from the set $\mathcal{U}(\boldsymbol{\theta})$ to the origin. Then the projection inequality (projection theorem) holds:

$$\langle \boldsymbol{v}_0(\boldsymbol{\theta}), \boldsymbol{u} \rangle \geq \|\boldsymbol{v}_0(\boldsymbol{\theta})\|^2, \forall \boldsymbol{u} \in \mathcal{U}(\boldsymbol{\theta}), \quad \text{(Proj)}$$

Where $\langle \boldsymbol{a}, \boldsymbol{b} \rangle = \boldsymbol{a}^\top \boldsymbol{b}$ denotes the inner product of vectors $\boldsymbol{a}$ and $\boldsymbol{b}$. When employing $\mu$-regularisation, the set undergoes a slight perturbation while its projection properties are approximately preserved. The following quantifies the projection error to demonstrate that for $\mu > 0$ and sufficiently small values, $\mu$-regularisation does not significantly impact the original problem, i.e., $\|\boldsymbol{\alpha}_\mu^* - \boldsymbol{\alpha}_0^*\| = \mathcal{O}(\mu)$.

**Lemma C.2** ($\mu$-Perturbed Projection Approximation). *There exists a constant $C_1 = \mathcal{O}(G_{\max}\sqrt{m})$ such that for all $\boldsymbol{\theta} \in \boldsymbol{\Theta}$ and $\mu > 0$, we have*

$$\|\boldsymbol{v}_\mu(\boldsymbol{\theta}) - \boldsymbol{v}_0(\boldsymbol{\theta})\| \leq C_1 \mu. \tag{5}$$

*Proof.* The proof process follows the following two facts:

1) From the compactness of $\boldsymbol{\Theta}$ and the boundedness of $\|\boldsymbol{g}(\boldsymbol{\theta})\|$, it follows that the set $\mathcal{U}(\boldsymbol{\theta})$ is uniformly up-bounded on $\boldsymbol{\theta}$.

2) The regularization term $\frac{1}{2}\|\boldsymbol{\alpha}\|$ unambiguously defines $\boldsymbol{\alpha}_\mu^*$ from the original $\boldsymbol{\alpha}_0^*$. Since the equation in (3) has a lower bound on the minimum eigenvalue of the Hessian matrix with respect to $\boldsymbol{\alpha}$, i.e., $\boldsymbol{g}(\boldsymbol{\theta})\boldsymbol{g}(\boldsymbol{\theta})^\top + \mu \boldsymbol{I}$, which is greater than or equal to $\mu$, the sensitivity of the inner-layer solution to $\boldsymbol{\theta}$ is controlled.

According to Moreau smoothing theory, we have $\|\boldsymbol{\alpha}_\mu^* - \boldsymbol{\alpha}_0^*\| = \mathcal{O}(\mu)$. Recall $\boldsymbol{v}_\mu(\boldsymbol{\theta}) = \boldsymbol{g}(\boldsymbol{\theta})^\top \boldsymbol{\alpha}_\mu^*$ and $\boldsymbol{v}_0(\boldsymbol{\theta}) = \boldsymbol{g}(\boldsymbol{\theta})^\top \boldsymbol{\alpha}_0^*$, we have

$$\|\boldsymbol{v}_\mu(\boldsymbol{\theta}) - \boldsymbol{v}_0(\boldsymbol{\theta})\| = \|\boldsymbol{g}(\boldsymbol{\theta})^\top (\boldsymbol{\alpha}_\mu^* - \boldsymbol{\alpha}_0^*)\| \leq \|\boldsymbol{g}(\boldsymbol{\theta})^\top\|_{\text{op}} \cdot \|\boldsymbol{\alpha}_\mu^* - \boldsymbol{\alpha}_0^*\|,$$

where $\|\boldsymbol{g}(\boldsymbol{\theta})^\top\|_{\mathrm{op}} = \sup_{\|\mathbf{x}\|=1}\|\boldsymbol{g}(\boldsymbol{\theta})^\top\mathbf{x}\|$. For any $\mathbf{x} \in \mathbb{R}^m$, we have

$$\|\boldsymbol{g}(\boldsymbol{\theta})^\top\mathbf{x}\| = \left\|\sum_{i=1}^m x_i \nabla J_i(\boldsymbol{\theta})\right\| \leq \sum_{i=1}^m |x_i| \cdot \|\nabla J_i(\boldsymbol{\theta})\| \leq G_{\max} \sum_{i=1}^m |x_i| = G_{\max}\|\mathbf{x}\|_1 \leq G_{\max}\sqrt{m}\|\mathbf{x}\|.$$

This means that

$$\|\boldsymbol{g}(\boldsymbol{\theta})^\top\|_{\mathrm{op}} = \sup_{\|\mathbf{x}\|=1}\|\boldsymbol{g}(\boldsymbol{\theta})^\top\mathbf{x} \leq G_{\max}\sqrt{m}$$

and

$$\|\boldsymbol{v}_\mu(\boldsymbol{\theta}) - \boldsymbol{v}_0(\boldsymbol{\theta})\| \leq G_{\max}\sqrt{m} \cdot \|\boldsymbol{\alpha}_\mu^* - \boldsymbol{\alpha}_0^*\| = G_{\max}\sqrt{m} \cdot \mathcal{O}(\mu) = \mathcal{O}(G_{\max}\sqrt{m})\mu = C_1\mu,$$

and we complete our proof. $\square$

*Remark* C.3. Lemma C.2 serves to relate quantities involving $\mu$ to the original ($\mu = 0$) projection properties, and provides an upper bound on the error introduced by perturbation; if the original problem can be guaranteed to be globally unique and continuous, then we can directly set $\mu = 0$.

### C.5. Exponential Convergence

To obtain a uniform negative convergence rate, we make a key assumption: there exists a constant $h_{\min} \in (0, 1)$ such that for all $\boldsymbol{\theta} \in \boldsymbol{\Theta}_s$ and $\mu > 0$, the following holds:

$$\boldsymbol{v}^\top \boldsymbol{H} \boldsymbol{v} \geq h_{\min}\|\boldsymbol{v}\|^2, \quad \text{(A1)}$$

where $\boldsymbol{\Theta}_s := \left\{\boldsymbol{\theta} | \boldsymbol{\theta} \in \boldsymbol{\Theta}, \min_{\boldsymbol{\alpha} \in \Delta_m}\|\nabla_{\boldsymbol{\theta}}\boldsymbol{J}(\boldsymbol{\theta})^\top\boldsymbol{\alpha}\|^2 \leq \epsilon\right\}$ is defined as the set consisting of policies located near the Pareto-stationary point, and $\epsilon > 0$ is a constant. An intuitive interpretation of this assumption: if the weighted Hessian $\boldsymbol{H}$ is a positive definite matrix then the assumption (A1) holds, i.e., the objective function $\boldsymbol{J}(\boldsymbol{\theta})$ is locally convex (i.e. directional positive curvature), which is common in the vicinity of Pareto-stationary set (Zhou et al., 2024).

**Lemma C.4.** *There exist constants $C_2 = 2C_1$ and $C_3 = C_1 G_{max}$ such that for all $\boldsymbol{\theta} \in \boldsymbol{\Theta}_s$, we have*

$$\langle \boldsymbol{v}, \boldsymbol{u} \rangle \geq \|\boldsymbol{v}\|^2 - C_2\mu\|\boldsymbol{v}\| - C_3\mu.$$

*Proof.*

$$\langle \boldsymbol{v}, \boldsymbol{u} \rangle = \langle \boldsymbol{v}_0 + (\boldsymbol{v} - \boldsymbol{v}_0), \boldsymbol{u} \rangle = \langle \boldsymbol{v}_0, \boldsymbol{u} \rangle + \langle \boldsymbol{v} - \boldsymbol{v}_0, \boldsymbol{u} \rangle$$

According to the Cauchy-Schwarz inequality, we have

$$\langle \boldsymbol{v} - \boldsymbol{v}_0, \boldsymbol{u} \rangle \geq -|\langle \boldsymbol{v} - \boldsymbol{v}_0, \boldsymbol{u} \rangle| \geq -\|\boldsymbol{v} - \boldsymbol{v}_0\| \cdot \|\boldsymbol{u}\|,$$

and according to the projection inequality (Proj) $\langle \boldsymbol{v}_0, \boldsymbol{u} \rangle \geq \|\boldsymbol{v}_0\|^2$, we have

$$\langle \boldsymbol{v}, \boldsymbol{u} \rangle \geq \|\boldsymbol{v}_0\|^2 - \|\boldsymbol{v} - \boldsymbol{v}_0\| \cdot \|\boldsymbol{u}\|.$$

According to the triangle inequality $\|\boldsymbol{v}_0\| \geq \|\boldsymbol{v}\| - \|\boldsymbol{v} - \boldsymbol{v}_0\|$, we have

$$\|\boldsymbol{v}_0\|^2 \geq (\|\boldsymbol{v}\| - \|\boldsymbol{v} - \boldsymbol{v}_0\|)^2 = \|\boldsymbol{v}\|^2 - 2\|\boldsymbol{v}\|\|\boldsymbol{v} - \boldsymbol{v}_0\| + \|\boldsymbol{v} - \boldsymbol{v}_0\|^2 \geq \|\boldsymbol{v}\|^2 - 2\|\boldsymbol{v}\|\|\boldsymbol{v} - \boldsymbol{v}_0\|$$

$$\Rightarrow \langle \boldsymbol{v}, \boldsymbol{u} \rangle \geq \|\boldsymbol{v}\|^2 - 2\|\boldsymbol{v}\|\|\boldsymbol{v} - \boldsymbol{v}_0\| - \|\boldsymbol{v} - \boldsymbol{v}_0\| \cdot \|\boldsymbol{u}\|$$

Finally, substituting $\|\boldsymbol{v}_\mu(\boldsymbol{\theta}) - \boldsymbol{v}_0(\boldsymbol{\theta})\| \leq C_1\mu$ (Lemma C.2) and $\|\boldsymbol{u}\| \leq G_{\max}$ into above yields

$$\langle \boldsymbol{v}, \boldsymbol{u} \rangle \geq \|\boldsymbol{v}\|^2 - 2C_1\mu\|\boldsymbol{v}\| - C_1 G_{\max}\mu,$$

and we complete our proof. $\square$

**Lemma C.5** (The Upper Bound of $\dot{V}_\mu(\boldsymbol{\theta})$)**.** *There exist constants $C = \frac{(\lambda C_4)^2}{2c} + \lambda C_3$, $C_4 = C_2\mu + (H_{\max} + 1)G_{\max}$, $c = \lambda + (1 - \lambda)h_{\min}$, $\lambda \in [0, \frac{h_{\min}}{1+h_{\min}})$, and $\mu > 0$ such that for all $\boldsymbol{\theta} \in \boldsymbol{\Theta}_s$, we have*

$$\dot{V}_\mu(\boldsymbol{\theta}) \leq -cV_\mu(\boldsymbol{\theta}) + C\mu. \tag{6}$$

*Proof.*

$$\boldsymbol{v}^\top \boldsymbol{H} \boldsymbol{u} = \langle \boldsymbol{v}, \boldsymbol{u} \rangle + \boldsymbol{v}^\top (\boldsymbol{H} - \boldsymbol{I}) \boldsymbol{u}$$

According to the Cauchy-Schwarz inequality, we have

$$\boldsymbol{v}^\top \boldsymbol{H} \boldsymbol{u} \geq \langle \boldsymbol{v}, \boldsymbol{u} \rangle - \|\boldsymbol{H} - \boldsymbol{I}\|_{\mathrm{op}} \|\boldsymbol{v}\| \|\boldsymbol{u}\|.$$

Since $\|\boldsymbol{H}\|_{\mathrm{op}} \leq H_{\max}$ and $\|\boldsymbol{I}\|_{\mathrm{op}} = 1$, it follows that $\|\boldsymbol{H} - \boldsymbol{I}\|_{\mathrm{op}} \leq H_{\max} + 1$. Further, with $\|\boldsymbol{u}\| \leq G_{\max}$, we obtain $\|\boldsymbol{H} - \boldsymbol{I}\|_{\mathrm{op}} \|\boldsymbol{v}\| \|\boldsymbol{u}\| \leq (H_{\max} + 1) G_{\max} \|\boldsymbol{v}\|$. Combining $\langle \boldsymbol{v}, \boldsymbol{u} \rangle \geq \|\boldsymbol{v}\|^2 - C_2 \mu \|\boldsymbol{v}\| - C_3 \mu$ (Lemma C.4) yields

$$\boldsymbol{v}^\top \boldsymbol{H} \boldsymbol{u} \geq \|\boldsymbol{v}\|^2 - C_2 \mu \|\boldsymbol{v}\| - C_3 \mu - (H_{\max} + 1) G_{\max} \|\boldsymbol{v}\| = \|\boldsymbol{v}\|^2 - C_4 \|\boldsymbol{v}\| - C_3 \mu$$

$$\Rightarrow -\dot{V}_\mu(\boldsymbol{\theta}) = \lambda \boldsymbol{v}^\top \boldsymbol{H} \boldsymbol{u} + (1 - \lambda) \boldsymbol{v}^\top \boldsymbol{H} \boldsymbol{v} \geq \lambda(\|\boldsymbol{v}\|^2 - C_4 \|\boldsymbol{v}\| - C_3 \mu) + (1 - \lambda) h_{\min} \|\boldsymbol{v}\|^2$$

$$\Rightarrow \dot{V}_\mu(\boldsymbol{\theta}) \leq -(\lambda + (1 - \lambda) h_{\min}) \|\boldsymbol{v}\|^2 + \lambda C_4 \|\boldsymbol{v}\| + \lambda C_3 \mu. \tag{7}$$

Observing the above inequality, we find that for $\dot{V}_\mu(\boldsymbol{\theta})$ to be negative, we must have $\lambda + (1 - \lambda) h_{\min} > 0$, thus $\lambda > -\frac{h_{\min}}{1 - h_{\min}}$. Moreover, since $-\frac{h_{\min}}{1 - h_{\min}} \leq 0 \leq \lambda$ always holds ($h_{\min} \in [0, 1)$), we let $c = \lambda + (1 - \lambda) h_{\min}$ and $0 \leq \lambda \leq 1$ yielding $h_{min} \leq c \leq 1$. At this point, we obtain $\dot{V}_\mu(\boldsymbol{\theta}) \leq -c \|\boldsymbol{v}\|^2 + \lambda C_4 \|\boldsymbol{v}\| + \lambda C_3 \mu$. By defining $V_\mu(\boldsymbol{\theta}) = \frac{1}{2} \|\boldsymbol{v}\|^2$ and Young's inequality, we obtain

$$\dot{V}_\mu(\boldsymbol{\theta}) \leq -c V_\mu(\boldsymbol{\theta}) + C \mu.$$

We complete our proof. $\qquad \square$

*Remark* C.6. Note that by relaxing inequality (7) to $\dot{V}_\mu(\boldsymbol{\theta}) \leq -(-\lambda + (1 - \lambda) h_{\min}) \|\boldsymbol{v}\|^2 + \lambda C_4 \|\boldsymbol{v}\| + \lambda C_3 \mu$, and let $-\lambda + (1 - \lambda) h_{\min} > 0$, we can deduce that $\lambda < \frac{h_{\min}}{1 + h_{\min}} < 0.5$. This suggests that the duty-cycle $\lambda$ should not be excessively large, meaning we require a greater number of Pareto-ascent phases to enable the policy to converge to the Pareto-stationary.

By Grönwall's inequality, we solve the inequality (6) as:

$$\dot{V}_\mu(\boldsymbol{\theta}(t)) + c V_\mu(\boldsymbol{\theta}(t)) \leq C \mu.$$

$$\Rightarrow e^{ct} \dot{V}_\mu(\boldsymbol{\theta}(t)) + c e^{ct} V_\mu(\boldsymbol{\theta}(t)) \leq C \mu e^{ct} \Rightarrow \frac{d}{dt} [e^{ct} V_\mu(\boldsymbol{\theta}(t))] \leq C \mu e^{ct}$$

$$\Rightarrow \int_0^t \frac{d}{ds} [e^{ct} V_\mu(\boldsymbol{\theta}(s))] ds \leq \int_0^t C \mu e^{cs} ds$$

$$\Rightarrow e^{ct} V_\mu(\boldsymbol{\theta}(t)) - V_\mu(\boldsymbol{\theta}(0)) \leq \frac{C \mu}{c} (e^{ct} - 1) \leq \frac{C \mu}{c} e^{ct}$$

$$\Rightarrow V_\mu(\boldsymbol{\theta}(t)) \leq e^{-ct} V_\mu(\boldsymbol{\theta}(0)) + \frac{C}{c} \mu. \tag{8}$$

Thus the value of $V_\mu(\boldsymbol{\theta})$ converges exponentially to an $O(\mu)$ neighborhood, i.e., $\boldsymbol{\theta}(0) \in \boldsymbol{\Theta}_s$ converges exponentially to a certain Pareto-stationary neighborhood by Pareto-tracking mechanism, denoted as $\lim_{t \to \infty} \sup \mathrm{dist}(\boldsymbol{\theta}(t), \boldsymbol{\Theta}_s) = \mathcal{O}(\mu)$.

### C.6. Euler Discretization and Step-Size Condition

For an arbitrary track $i \in \{1, ..., m\}$ we approximate the continuous dynamics by an explicit Eulerian discretization. The continuous dynamics is $\dot{\boldsymbol{\theta}} = F_i(\boldsymbol{\theta})$, the discrete iteration is

$$\boldsymbol{\theta}(t + 1) = \boldsymbol{\theta}(t) + \eta \dot{\boldsymbol{\theta}}(t),$$

where $\eta > 0$ is step-size (i.e. learning rate).

**Lemma C.7.** *If the step size satisfies* $0 \leq \eta \leq \eta_{\max}$, $\eta_{\max} = \min\{\frac{c}{2 L_V M_F^2}, \frac{1}{c}\}$, *we have*

$$\lim_{t \to \infty} V_\mu(\boldsymbol{\theta}(t)) = \mathcal{O}(\mu). \tag{9}$$

*Proof.* Using the gradient boundedness and Lipschitz continuity of $V_\mu(\boldsymbol{\theta})$, a first-order Taylor expansion is obtained:

$$V_\mu(\boldsymbol{\theta}(t+1)) \leq V_\mu(\boldsymbol{\theta}(t)) + \eta \nabla_{\boldsymbol{\theta}} V_\mu(\boldsymbol{\theta}(t))^\top F_i(\boldsymbol{\theta}(t)) + \tfrac{\eta^2}{2} L_V \|F_i(\boldsymbol{\theta}(t))\|^2,$$

where $L_V = \mathcal{O}(\max(H_{\max} G_{\max}, 1/\mu))$ denotes the Lipschitz constant of $\nabla_{\boldsymbol{\theta}} V_\mu(\boldsymbol{\theta})$. According to lemma C.5, we have

$$V_\mu(\boldsymbol{\theta}(t+1)) \leq V_\mu(\boldsymbol{\theta}(t)) + \eta(-cV_\mu(\boldsymbol{\theta}) + C\mu) + \tfrac{\eta^2}{2} L_V M_F^2$$

$$\Rightarrow V_\mu(\boldsymbol{\theta}(t+1)) \leq (1 - \eta c) V_\mu(\boldsymbol{\theta}(t)) + \eta C\mu + \tfrac{\eta^2}{2} L_V M_F^2,$$

where $M_F := \sup_{\boldsymbol{\theta} \in \boldsymbol{\Theta}} V\|F_i(\boldsymbol{\theta})\| \leq G_{\max}$ is a constant. In order to control the second-order term $\tfrac{\eta^2}{2} L_V M_F^2$, the step size should satisfy $0 < \eta \leq \min\{\frac{c}{2L_V M_F^2}, \frac{1}{c}\}$, then it can be guaranteed that

$$\Rightarrow V_\mu(\boldsymbol{\theta}(t+1)) \leq (1 - \eta c) V_\mu(\boldsymbol{\theta}(t)) + \eta C\mu,$$

$$\Rightarrow V_\mu(\boldsymbol{\theta}(t)) \leq (1 - \eta c)^t \left( V_\mu(\boldsymbol{\theta}(0)) - \frac{C}{c}\mu \right) + \frac{C}{c}\mu$$

$$\Rightarrow \lim_{t \to \infty} \sup V_\mu(\boldsymbol{\theta}(t)) = \frac{C}{c}\mu = \mathcal{O}(\mu),$$

and we complete our proof. □

*Remark* C.8. Lemma C.7 indicates that when using the GD algorithm to update the parameters $\boldsymbol{\theta}$ discretely, if the step-size satisfies $0 < \eta \leq \eta_{\max}$, then the value of $V_\mu(\boldsymbol{\theta}(0))$ will ultimately converge to the $\mathcal{O}(\mu)$-neighborhood.

## C.7. From Averaged System to Switching System

The real Pareto-tracking mechanism alternates periodically with period $\tau = \tau_{rev} + \tau_{asc}, \tau > 0$ (i.e. $\lambda = \frac{\tau_{rev}}{\tau}$). By standard averaging theory we have:

**Proposition C.9** (Switching vs Averaged System). *Suppose that the vector field for each phase satisfies Lipschitz continuity. Then for any period $\tau \geq 1$:*

- *On the timescale $t = \mathcal{O}(1)$, the distance between the switching system trajectory $\bar{\boldsymbol{\theta}}(t)$ and the average system trajectory $\boldsymbol{\theta}(t)$ has an upper bound:*
$$\|\bar{\boldsymbol{\theta}}(t) - \boldsymbol{\theta}(t)\| \leq \mathcal{O}(\tau), \forall t \in [0, \tau].$$

- *When $\tau$ is small, the exponential convergence property of the average system is approximately preserved for the switching system: there exist $C' > 0$ ($C' = \mathcal{O}(C)$) such that the Lyapunov function of the switching system satisfies the same form as (8), but with an additional periodic switching error term $\mathcal{O}(\tau)$:*

$$\dot{V}_\mu(\boldsymbol{\theta}(t)) \leq -cV_\mu(\boldsymbol{\theta}(t)) + C'(\mu + \tau).$$

$$\Rightarrow V_\mu(\boldsymbol{\theta}(t)) \leq e^{-ct} V_\mu(\boldsymbol{\theta}(0)) + \frac{C'}{c}(\mu + \tau). \tag{10}$$

*Proof.* According to the standard averaging theory we know that the period average of the vector field converges approximately to the average vector field, and the error is $\mathcal{O}(\tau)$ when the period $\tau$ is small enough (Llibre et al., 2014); the exponential stability is robust to small perturbations, therefore, combining the average system error $\mathcal{O}(\mu)$ with the periodic switching system error $\mathcal{O}(\tau)$ yields (10), and the proof is complete. □

*Remark* C.10. Proposition C.9 demonstrates that when the period $\tau$ is sufficiently small, the exponential convergence property of the switching system is approximately preserved, with a convergence rate of $c$. Eulerian discretisation also yields $\lim_{t \to \infty} \sup V_\mu(\boldsymbol{\theta}(t)) = \mathcal{O}(\mu + \tau)$.

## C.8. Main Theorem

According to the Lemmas C.2, C.4, C.5, C.7 and Proposition C.9, Theorem C.11 holds as follows:

**Theorem C.11** (Convergence of Pareto-Tracking Mechanism). *Suppose:*

- *Parameter space $\boldsymbol{\Theta}$ is convex and compact, and for all objective functions $\{J_i(\boldsymbol{\theta})\}$ are second-order continuous differentiable, $i \in \{1, \ldots, m\}$;*

- *Regularization parameter $\mu > 0$ is small enough (Lemma 1 holds);*

- *There exist a constant $h_{\min} \in (0, 1)$ such that directional positive curvature condition holds near Pareto-stationary set (Assumption (A1) holds);*

- *Duty-cycle $\lambda$ satisfies $0 \leq \lambda < \frac{h_{\min}}{1+h_{\min}} < 0.5$;*

- *Switching period $\tau$ is small;*

- *Step-size $\eta$ satisfies $0 < \eta \leq \eta_{\max} < 1$.*

*Then there exist constants $h_{min} \leq c \leq 1$ and $C > 0$ such that for the average system the Lyapunov function satisfies*

$$V_\mu(\boldsymbol{\theta}(t)) \leq (1 - \eta c)^t \left( V_\mu(\boldsymbol{\theta}(0)) - \frac{C}{c}\mu \right) + \frac{C}{c}\mu,$$

*and for the switching system the Lyapunov function satisfies*

$$V_\mu(\boldsymbol{\theta}(t)) \leq (1 - \eta c)^t \left( V_\mu(\boldsymbol{\theta}(0)) - \frac{C'}{c}(\mu + \tau) \right) + \frac{C'}{c}(\mu + \tau), \quad C' = \mathcal{O}(C),$$

*i.e., for any $\boldsymbol{\theta} \in \boldsymbol{\Theta}_s$ converges exponentially to certain neighborhood of the Pareto-stationary set in any arbitrary objective dimension by Pareto-tracking mechanism.*

*Remark* C.12. From Theorem C.11, the policies tracked by the MPFT framework remain within an $\mathcal{O}(\mu + \tau)$-neighborhood of the Pareto-stationary set in any arbitrary objective dimension, as corroborated by Figures 6 and 7 in Appendix G. The period $\tau$ depends on the number of training episodes per phase, denoted by $u$ and $v$. The duty cycle $0 \leq \lambda < \frac{h_{\min}}{(1+h_{\min})} < 0.5$ implies that $u < v$, and $\lambda$ is proportional to the convergence rate $c$, such that larger $\lambda$ leads to faster convergence. Hence, parameter settings should be adapted to the specific environment. In addition, the initial policy of the Pareto-tracking mechanism is the Pareto-vertex policy $\boldsymbol{\theta}^{i,*}$, and $\boldsymbol{\theta}^{i,*} \in \boldsymbol{\Theta}_s$ by definition. Therefore, during the Pareto front tracking process, the policy parameters stay within the vicinity of the Pareto-stationary set.

# D. Evaluation Metrics, and Time and Space Complexity

## D.1. HV, SP, and EU

The HV is given by $\mathbf{HV} = \Omega_m \left( \cup_{j=1}^{|\mathcal{N}|} \mathbb{V}_j \right)$, where $\Omega_m$ represents the volume measurement in $m$-dimensional Euclidean space (in two-dimensional space, $\Omega_2$ represents the area), and $\mathbb{V}_j$ is the space enclosed by the $j$-th solution in the set $\mathcal{N}$ and the reference point. A larger $\mathbf{HV}$ indicates a better approximation of Pareto front. The SP is given by $\mathbf{SP} = \frac{1}{|\mathcal{N}|-1} \sum_{i=1}^{m-1} \sum_{j=1}^{|\mathcal{N}|} \left( \tilde{\mathcal{N}}_j(i+1) - \tilde{\mathcal{N}}_j(i) \right)^2$, where $\mathcal{N}_j \in \mathbb{R}^m$ represents the mapping vector of the $j$-th policy in $\mathcal{F}$, $\mathcal{N}_j(i)$ is the value of objective $i$ in $\mathcal{N}_j$, and $\tilde{\mathcal{N}}_j(i)$ is the value after sorting the $i$-th objective in ascending order. A smaller $\mathbf{SP}$ indicates that the solutions in $\mathcal{N}$ are more densely distributed. The EU is given by $\mathbf{EU} = \mathbb{E}_{\boldsymbol{\omega} \sim \Omega} \left[ \max\{\boldsymbol{J}(\boldsymbol{\theta}_j)^\top \boldsymbol{\omega}\}_{j=1}^{|\mathcal{N}|} \right]$, where $\Omega$ is a specified preferences set. A higher expected utility is better.

## D.2. Calculation of env_steps for MPFT framework

$\mathbf{env\_steps} = \text{steps} \times \left( \sum_{i=1}^{m}(\Xi_i + \Psi_i) + \sum_{k=1}^{K}(\Xi_k + \Psi_k) \right)$, where $\Xi_i$ is the number of episodes in finding the approximate Pareto-vertex policies $\pi_{\boldsymbol{\theta}^{i,*}}$, $\Xi_k$ is the number of episodes in finding the approximate Pareto-interior policies $\pi_{\boldsymbol{\theta}^{k,*}}$, $\Psi_i$ is the number of episodes required to construct the Pareto-edge policy set $\mathcal{F}^i_{edge}$, and $\Psi_k$ is the number of episodes required to construct the Pareto-interior policy tracking set $\mathcal{F}^k_{interior}$ for all $i \in \{1, \ldots, m\}$ and all $k \in \{1, \ldots, K\}$. The steps is the number of timesteps in an episode of Algorithms 1.

### D.3. Calculation of env_steps for Evolutionary-based MORL Framework

**env_steps** $= n \times D \times (m_w + (G \times m_t))$, where $n$ is the number of policies selected for each iteration, $D$ is the number of interactions with the environment during one policy iteration (i.e., timesteps per actorbatch), $m_w$ is the number of warm-up iterations, $m_t$ is the number of iterations required for the population to complete one generation of evolution, and $G$ is the total number of generations of evolution of the population.

### D.4. Time and Space Complexity

Let $\Upsilon$ denotes an upper bound on the cost of training a single policy, and set the number of times a policy is trained equal to the number of times corresponding agent interacts with the environment. If parallel training is not considered, the time complexity of the Stage 1 is $\mathcal{O}(\text{steps} \times \sum_{i=1}^{m} \Xi_i)$. The time complexity of the Stage 2 is $\mathcal{O}(\text{steps} \times \sum_{i=1}^{m} \Psi_i)$. The time complexity of the Stage 3 is $\mathcal{O}\left(\text{steps} \times \sum_{k=1}^{K}(\Xi_k + \Psi_k)\right)$. The Stage 4 is negligible. Therefore, the time complexity of the MPFT is

$$T_{\text{MPFT}} = \mathcal{O}\left(\Upsilon \times \text{steps} \times \left(\sum_{i=1}^{m}(\Xi_i + \Psi_i) + \sum_{k=1}^{K}(\Xi_k + \Psi_k)\right)\right), \tag{11}$$

and the time complexity of the evolutionary framework based MP-MORL is

$$T_{\text{EVOL}} = \mathcal{O}\left(\Upsilon \times n \times D \times (m_w + (G \times m_t)) + P_{sel}\right) = \mathcal{O}\left(\Upsilon \times D \times (m_w + (G \times m_t)) + \mathcal{O}(P_{sel})\right), \tag{12}$$

where $P_{sel}$ denotes the time spent on policy selection, which varies from algorithm to algorithm, e.g., PGMORL is $P_{sel} = Gn^{m-2}$ (needs to solve a knapsack problem), PA2D-MORL is $P_{sel} = G$ (randomly select $n$ policies from the candidate policies), and C-MORL is $P_{sel} = m|\mathcal{F}|\log|\mathcal{F}|$ (select $n$ policies from $\mathcal{F}$ based on crowding distance). Note that, unlike evolutionary frameworks, MPFT does not require policy selection, and thus $P_{\text{sel}} = 0$. Consequently, when the same policy gradient algorithm (e.g., PPO) is used, we typically have $T_{\text{MPFT}} < T_{\text{EVOL}}$.

For the space complexity, the MPFT is $S_{\text{MPFT}} = \mathcal{O}(|\mathcal{F}| \times \Gamma)$, and the evolutionary framework based MP-MORL is $S_{\text{EVOL}} = \mathcal{O}((|\mathcal{F}| + |\mathcal{P}_{op}|) \times \Gamma)$, where $\mathcal{F}$ is the Pareto-approximation policy set, $\mathcal{P}_{op}$ is the policy population, and $\Gamma$ is the memory space occupied for saving a policy $\pi_{\boldsymbol{\theta}}$. Note that the values of $\Upsilon$ and $\Gamma$ will be different using different algorithms, but whatever algorithm is used $S_{\text{MPFT}}$ is significantly lower than $S_{\text{EVOL}}$. This is due to $|\mathcal{P}_{op}| \gg |\mathcal{F}|$.

## E. Comparision with Evolutionary Based MP-MORL

To clarify the conceptual differences between MPFT and existing evolutionary MP-MORL frameworks, we provide a concise comparison as shown in Table 4. PGMORL, PA2D-MORL, and C-MORL all rely on population-based or non-dominated policy set-based policy selection within an evolutionary loop, whereas MPFT eliminates population-based evolution entirely and instead employs a deterministic single-policy Pareto-tracking mechanism for each track. This fundamental shift leads to differences in initialization, evolution, sparse-region handling, computational behavior, and offline-RL compatibility.

*Table 4.* Comparison between evolutionary MP-MORL frameworks and MPFT.

| Component | PGMORL | PA2D-MORL | C-MORL | MPFT (ours) |
|---|---|---|---|---|
| Framework type | Evolutionary | Evolutionary | Evolutionary | **Policy tracking** |
| Initialization | $\mathcal{P}_{op} = \{\pi^{(1)}, \ldots, \pi^{(n)}\}$ | $\mathcal{P}_{op}$ | $\mathcal{P}_{op}$ | $\{\pi^{(1)}, \ldots, \pi^{(m)}\}$ |
| Policies selection | from $\mathcal{P}_{op}$ | from $\mathcal{P}_{op}$ and $\mathcal{F}$ | from $\mathcal{F}$ | **None** |
| Selection rule | Predictive model | Random | Crowding-distance | **None** |
| Policy updating method | Predictive model | Pareto-ascent direction | Interior point method | **Pareto-tracking mechanism** |
| Sparse-region handling | Interpolation | Sparse regions sampling | None | **New interior policy + tracking** |
| Complexity driver | $T_{\text{EVOL}}$ (12) | $T_{\text{EVOL}}$ (12) | $T_{\text{EVOL}}$ (12) | $T_{\text{MPFT}}$ (11) |
| Offline RL | × | × | × | ✓ |

# F. Experiment Detail

## F.1. Multi-Objective Task Environment

Our robotics control environment is built using MuJoCo 3.3.0 (Todorov et al., 2012) and the v5 version of the tasks in Gymnasium 1.1.1. According to the official documentation (Farama Foundation, 2025), the v5 version fixes a previous bug where the robot could receive rewards while in an unhealthy state, and it also increases the difficulty of robot control, posing a greater challenge for RL algorithms. The reward function for each environment is defined as $R_i$ for the $i$-th objective, with all rewards scaled to similar ranges. Each robotics control task has a fixed duration of 1000 timesteps (i.e., maximum timestep).

**HalfCheetah-2**: Action and observation space dimensionality: $\mathcal{A} \in \mathbb{A}^6$ and $\mathcal{S} \in \mathbb{R}^{17}$. The first objective is forward speed:

$$R_1 = \min(0.5 \times v_x, 2).$$

The second objective is energy efficiency:

$$R_2 = 2 - \sum_{i=1}^{6} a_i^2,$$

where $v_x$ is the speed in $x$ direction, and $a_i$ is the action of each actuator.

**Hopper-2**: Action and observation space dimensionality: $\mathcal{A} \in \mathbb{R}^3$ and $\mathcal{S} \in \mathbb{R}^{11}$. The first objective is speed:

$$R_1 = 2 \times v + R_{\text{alive}} - C_{\text{cost}}.$$

The second objective is jumping height:

$$R_2 = 20 \times \max(0, h - h_{\text{init}}) + R_{\text{alive}} - C_{\text{cost}},$$

where $v$ is the speed, $R_{\text{alive}} = 1$ is the alive bonus, $C_{\text{cost}} = 0.0002 \times \sum_{i=1}^{3} a_i^2$ is the control cost, $h$ is the current height, $h_{\text{init}}$ is the initial height, and $a_i$ is the action of each actuator.

**Swimmer-2**: Action and observation space dimensionality: $\mathcal{A} \in \mathbb{R}^2$ and $\mathcal{S} \in \mathbb{R}^8$. The first objective is speed:

$$R_1 = v.$$

The second objective is energy efficiency:

$$R_2 = 2 - \sum_{i=1}^{2} a_i^2,$$

where $v$ is the speed, and $a_i$ is the action of each actuator.

**Ant-2**: Action and observation space dimensionality: $\mathcal{A} \in \mathbb{R}^8$ and $\mathcal{S} \in \mathbb{R}^{105}$. The first objective is x-axis speed:

$$R_1 = 0.35 \times v_x + R_{\text{alive}} - C_{\text{cost}}.$$

The second objective is y-axis speed:

$$R_2 = 0.35 \times v_y + R_{\text{alive}} - C_{\text{cost}},$$

where $v_x$ is the x-axis speed, $v_y$ is the y-axis speed, $R_{\text{alive}} = 1$ is the alive bonus, $C_{\text{cost}} = \sum_{i=1}^{8} a_i^2$ is the control cost, and $a_i$ is the action of each actuator.

For Ant-2, we multiply the velocity reward by 0.35 to intentionally increase task difficulty. During initial training, large random exploratory actions cause the energy penalty ($C_{\text{cost}}$) to dominate, creating a strong local optimum corresponding to conservative, "standing still" behaviors. The 0.35 multiplier raises the threshold to offset this penalty, forcing the agent to explore highly coordinated gaits.

**Walker2d-2**: Action and observation space dimensionality: $\mathcal{A} \in \mathbb{R}^6$ and $\mathcal{S} \in \mathbb{R}^{17}$. The first objective is speed:

$$R_1 = v + R_{\text{alive}}.$$

The second objective is energy efficiency:

$$R_2 = 3 - \sum_{i=1}^{6} a_i^2 + R_{\text{alive}},$$

where $v$ is the speed, $R_{\text{alive}} = 1$ is the alive bonus, and $a_i$ is the action of each actuator.

**Hopper-3**: Action and observation space dimensionality: $\mathcal{A} \in \mathbb{R}^3$ and $\mathcal{S} \in \mathbb{R}^{11}$. The first objective is speed:

$$R_1 = 2 \times v + R_{\text{alive}}.$$

The second objective is jumping height:

$$R_2 = 20 \times \max(0, h - h_{\text{init}}) + R_{\text{alive}}.$$

The third objective is energy efficiency:

$$R_3 = \max(0, 3 - 20 \times \sum_{i=1}^{3} a_i^2) + R_{\text{alive}},$$

where $v$ is the speed, $R_{\text{alive}} = 1$ is the alive bonus, $h$ is the current height, $h_{\text{init}}$ is the initial height, and $a_i$ is the action of each actuator.

Additionally, the discrete 4-D environment **Lunar-Lander-4**, the discrete 6-D environment **Fruit-Tree-6**, and the continuous 9-D environment **Building-9** are built same as C-MORL (Liu et al., 2025b). For EU metric evaluation, we evenly generate an preference set $\Omega$ in a systematic manner with specified intervals $\Delta$. For all environment configuration details, see Table 5.

### F.2. Training Details

We run all our experiments on a workstation with an Intel i9-13900K CPU, NVIDIA GeForce RTX-4090, and 128G memory. For MOPPO, MOSAC, and MOTD7, their neural network structures are shown in the Pseudocodes 1–3, respectively. Each hidden layer comprises 256 units, activated by both `Tanh(·)` and `ReLU(·)`. For PGMORL, PA2D-MORL, and C-MORL we use the same neural network structure as MOPPO.

**Hyperparameter Settings:**

- **env_steps**: The total number of environment steps, with details provided in Appendix D.2.

- **MOPPO:** All hyperparameters are consistent across all benchmarks and multi-objective robotic control environments, as listed in Table 6.

- **MOSAC:** All hyperparameters are consistent across all multi-objective environments, as listed in Table 7.

- **MOTD7:** All hyperparameters are consistent across all multi-objective environments, as listed in Table 8.

- **Benchmarks:** Following (Xu et al., 2020), (Hu & Luo, 2024), and (Liu et al., 2025b) hyperparameters for PGMORL, PA2D-MORL and C-MORL are set to ensure convergence of all algorithms, as shown in Tables 9 and 10 across all $\leq$ 3-objective environments. The meaning of each parameter is described in Appendix D. C-MORL updates the policy using the interior point method (IPO). In Table 10, $G = m \times G'$, $G'$ denotes the total number of generations in population evolution, and $m$ is the number of objectives.

- **MPFT framework:** Episode settings for training (i.e., $\{\Xi_i, \Psi_i\}_{i=1}^{m}$, $\{\Xi_k, \Psi_k\}_{k=1}^{K}$, $u$, and $v$ in Algorithm 1) are summarized in Table 12 across all $\leq$ 3-objective environments, and steps settings are listed in Table 11. For all $> 3$-objective environments the hyperparameter configuration is as: for Fruit-Tree-6, $\Xi_{i=1} = 60$, $\{\Xi_i\}_{i=2}^{6} = 0$, $\Psi_{i=1} = 60$, $\{\Psi_i\}_{i=2}^{6} = 0$, $u = 1$, $v = 2$, steps $= 280$, and $K = 0$; for Lunar-Lander-4, $\Xi_{i=2} = 240$, $\{\Xi_i\}_{i\neq2} = 0$, $\Psi_{i=1} = 80$, $\{\Psi_i\}_{i\neq2} = 0$, $u = 1$, $v = 2$, steps $= 1000$, and $K = 0$; and for Building-9, $\Xi_{i=1} = 30$, $\{\Xi_i\}_{i=2}^{9} = 0$, $\Psi_{i=1} = 200$, $\{\Psi_i\}_{i=2}^{9} = 0$, $u = 1$, $v = 2$, steps $= 1000$, $K = 0$.

*Table 5.* Environment details of continuous control benchmarks.

| Environments | $m$ | Continuous | State Space | Action Space | Policy buffer sizes | Reference point | $\Delta$ |
|---|---|---|---|---|---|---|---|
| HalfCheetah-2 | 2 | Yes | $\mathcal{S} \subset \mathbb{R}^{17}$ | $\mathcal{A} \subset \mathbb{R}^{6}$ | 200 | $[0, 0]$ | 0.01 |
| Hopper-2 | 2 | Yes | $\mathcal{S} \subset \mathbb{R}^{11}$ | $\mathcal{A} \subset \mathbb{R}^{3}$ | 200 | $[0, 0]$ | 0.01 |
| Swimmer-2 | 2 | Yes | $\mathcal{S} \subset \mathbb{R}^{8}$ | $\mathcal{A} \subset \mathbb{R}^{2}$ | 200 | $[0, 0]$ | 0.01 |
| Ant-2 | 2 | Yes | $\mathcal{S} \subset \mathbb{R}^{105}$ | $\mathcal{A} \subset \mathbb{R}^{8}$ | 200 | $[0, 0]$ | 0.01 |
| Walker2d-2 | 2 | Yes | $\mathcal{S} \subset \mathbb{R}^{17}$ | $\mathcal{A} \subset \mathbb{R}^{6}$ | 200 | $[0, 0]$ | 0.01 |
| Hopper-3 | 3 | Yes | $\mathcal{S} \subset \mathbb{R}^{11}$ | $\mathcal{A} \subset \mathbb{R}^{3}$ | 300 | $[0, 0]$ | 0.1 |
| Lunar-Lander-4 | 4 | No | $\mathcal{S} \subset \mathbb{R}^{8}$ | $\mathcal{A} \subset \mathbb{R}^{4}$ | 300 | $[-101, -1001, -101, -101]$ | 0.1 |
| Fruit-Tree-6 | 6 | No | $\mathcal{S} \subset \mathbb{R}^{2}$ | $\mathcal{A} \subset \mathbb{A}^{2}$ | 300 | $[-1, -1, -1, -1, -1, -1]$ | 0.5 |
| Building-9 | 9 | Yes | $\mathcal{S} \subset \mathbb{R}^{29}$ | $\mathcal{A} \subset \mathbb{A}^{23}$ | 300 | $[0, 0, 0, 0, 0, 0, 0, 0, 0]$ | 0.5 |

*Table 6.* MOPPO hyperparameters.

| parameter name | value |
|---|---|
| learning rate | $3 \times 10^{-4}$ |
| discount ($\gamma$) | 0.99 |
| gae lambda | 0.95 |
| mini-batch size | 32 |
| ppo epoch | 10 |
| entropy coef | 0.01 |
| value loss coef | 0.5 |
| optimizer | Adam |

*Table 8.* MOTD7 hyperparameters.

| parameter name | value |
|---|---|
| learning rate | $3 \times 10^{-4}$ |
| discount ($\gamma$) | 0.99 |
| replay buffer size | $10^{6}$ |
| mini-batch size | 256 |
| target policy noise | $\mathcal{N}(0, 0.2^2)$ |
| target policy noise clipping $\epsilon$ | (-0.5, 0.5) |
| policy update frequency | 2 |
| Target update rate | 0.004 |
| optimizer | Adam |

*Table 9.* PA2D-MORL/PGMORL hyperparameters setting.

| Environment | $n$ | $m_w$ | $m_t$ | $D$ | $G$ | env_steps |
|---|---|---|---|---|---|---|
| Walker2d-2 | 8 | 80 | 20 | 4096 | 60 | $8 \times 4096 \times (80 + (60 \times 20))$ |
| Halfcheetah-2 | 8 | 80 | 20 | 4096 | 60 | $8 \times 4096 \times (80 + (60 \times 20))$ |
| Hopper-2 | 8 | 200 | 40 | 4096 | 60 | $8 \times 4096 \times (200 + (60 \times 40))$ |
| Ant-2 | 8 | 200 | 40 | 4096 | 60 | $8 \times 4096 \times (200 + (60 \times 40))$ |
| Swimmer-2 | 8 | 40 | 10 | 4096 | 60 | $8 \times 4096 \times (40 + (60 \times 10))$ |
| Hopper-3 | 15 | 200 | 40 | 4096 | 50 | $15 \times 4096 \times (200 + (50 \times 40))$ |

*Table 7.* MOSAC hyperparameters.

| parameter name | value |
| --- | --- |
| learning rate | $3 \times 10^{-4}$ |
| discount ($\gamma$) | 0.99 |
| replay buffer size | $10^6$ |
| mini-batch size | 256 |
| target smoothing coef ($\tau$) | 0.001 |
| target entropy ($\hat{\mathcal{H}}_i$) | $-|\mathcal{A}|$ |
| optimizer | Adam |

*Table 10.* C-MORL hyperparameters setting.

| Environment | $n$ | $m_w$ | $m_t$ | $D$ | $G$ | env_steps |
| --- | --- | --- | --- | --- | --- | --- |
| Walker2d-2 | 8 | 80 | 20 | 4096 | $2 \times 30$ | $8 \times 4096 \times (80+(60 \times 20))$ |
| Halfcheetah-2 | 8 | 80 | 20 | 4096 | $2 \times 30$ | $8 \times 4096 \times (80+(60 \times 20))$ |
| Hopper-2 | 8 | 200 | 40 | 4096 | $2 \times 30$ | $8 \times 4096 \times (200+(60 \times 40))$ |
| Ant-2 | 8 | 200 | 40 | 4096 | $2 \times 30$ | $8 \times 4096 \times (200+(60 \times 40))$ |
| Swimmer-2 | 8 | 40 | 10 | 4096 | $2 \times 30$ | $8 \times 4096 \times (40+(60 \times 10))$ |
| Hopper-3 | 12 | 200 | 40 | 4096 | $3 \times 20$ | $12 \times 4096 \times (200+(60 \times 40))$ |

*Table 11.* steps setting.

| Algorithm | Walker2d-2 | Halfcheetah-2 | Hopper-2 | Ant-2 | Swimmer-2 | Hopper-3 |
| --- | --- | --- | --- | --- | --- | --- |
| MPFT-MOPPO | 2048 | 2048 | 8192 | 4096 | 2048 | 8192 |
| MPFT-MOSAC | 2000 | 2000 | 4000 | 4000 | 2000 | 4000 |
| MPFT-MOTD7 | 2000 | 2000 | 2000 | 2000 | 2000 | 2000 |

# G. Additional Results

In this section, we provide additional experimental results across all robotics control environments, including visualizations of the Pareto-approximation fronts for all algorithms, tracking analyses of the MPFT framework, convergence analyses of the MPFT framework, hyperparameter sensitivity and robustness analyses of the MPFT framework. For all MPFT framework based MP-MORL algorithms, we focus on the top-1 sparse region (i.e., $K = 1$).

*Table 12.* Hyperparameters of MPFT framework based algorithms in $\leq$ 3-D environment ("$(1+2)$" indicates the numbers of consecutive Pareto-reverse and Pareto-ascent updates, i.e., $u=1$ and $v=2$).

| Environment | Algorithm (MPFT-) | $\Xi_{i=1}$ / $\Psi_{i=1}$ | $\Xi_{i=2}$ / $\Psi_{i=2}$ | $\Xi_{i=3}$ / $\Psi_{i=3}$ | $\{\Xi_k/\Psi_k\}_{k=1}^K$ |
|---|---|---|---|---|---|
| Walker2d-2 | MOPPO | 1500 / 300×(1+2) | 600 / 800×(1+2) | - / - | 800 / 800×(1+2) |
| | MOSAC | 1000 / 500×(1+2) | 400 / 1000×(1+2) | - / - | 500 / 600×(1+2) |
| | MOTD7 | 500 / 500×(1+2) | 100 / 500×(1+2) | - / - | 500 / 500×(1+2) |
| Halfcheetah-2 | MOPPO | 600 / 800×(0+2) | 600 / 800×(0+2) | - / - | 600 / 800×(0+2) |
| | MOSAC | 500 / 400×(0+2) | 500 / 400×(0+2) | - / - | 500 / 400×(0+2) |
| | MOTD7 | 100 / 300×(0+2) | 100 / 300×(0+2) | - / - | 100 / 500×(0+2) |
| Hopper-2 | MOPPO | 200 / 300×(1+2) | 200 / 300×(1+2) | - / - | 200 / 300×(1+2) |
| | MOSAC | 700 / 400×(1+2) | 700 / 400×(1+2) | - / - | 700 / 400×(1+2) |
| | MOTD7 | 800 / 800×(1+2) | 800 / 800×(1+2) | - / - | 800 / 800×(1+2) |
| Ant-2 | MOPPO | 1200 / 400×(0+2) | 1200 / 400×(0+2) | - / - | 1200 / 400×(0+2) |
| | MOSAC | 700 / 400×(1+2) | 700 / 400×(1+2) | - / - | 700 / 400×(1+2) |
| | MOTD7 | 800 / 800×(1+2) | 800 / 800×(1+2) | - / - | 800 / 800×(1+2) |
| Swimmer-2 | MOPPO | 400 / 150×(1+2) | 400 / 150×(1+2) | - / - | 400 / 200×(1+2) |
| | MOSAC | 300 / 200×(1+2) | 300 / 200×(1+2) | - / - | 300 / 200×(1+2) |
| | MOTD7 | 100 / 200×(0+2) | 100 / 200×(0+2) | - / - | 100 / 400×(0+2) |
| Hopper-3 | MOPPO | 500 / 400×(1+2) | 500 / 400×(1+2) | 500 / 400×(1+2) | 500 / 400×(1+2) |
| | MOSAC | 800 / 600×(1+2) | 800 / 600×(1+2) | 800 / 600×(1+2) | 800 / 600×(1+2) |
| | MOTD7 | 800 / 1200×(1+2) | 500 / 1500×(1+2) | 400 / 1400×(1+2) | 1000 / 1500×(1+2) |

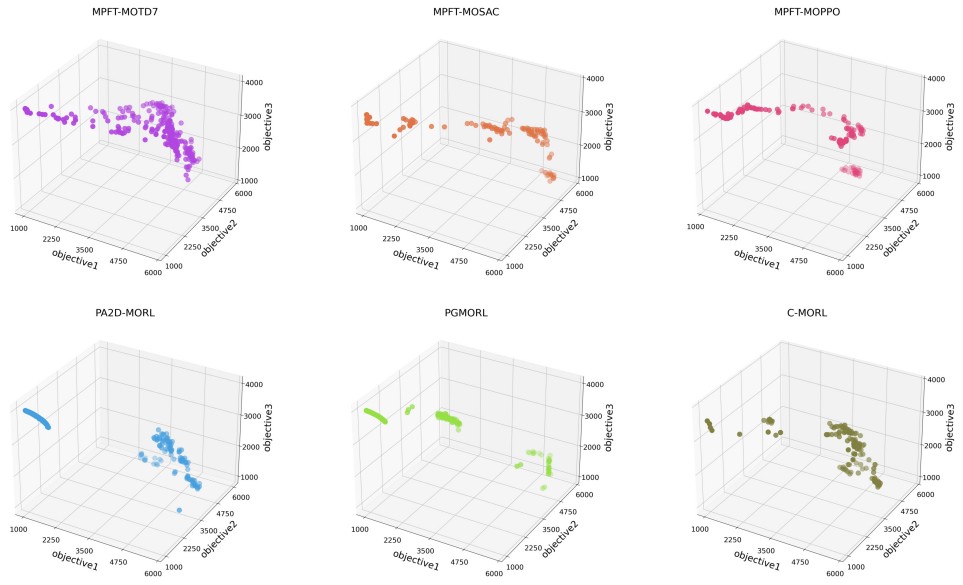

*Figure 3.* The Pareto front approximation comparison for the 3-objective Hopper-3 environment. We show the best seed for each algorithm. Ojective1: Forward reward; Objective2: Jumping reward; Objective3: Energy efficiency.

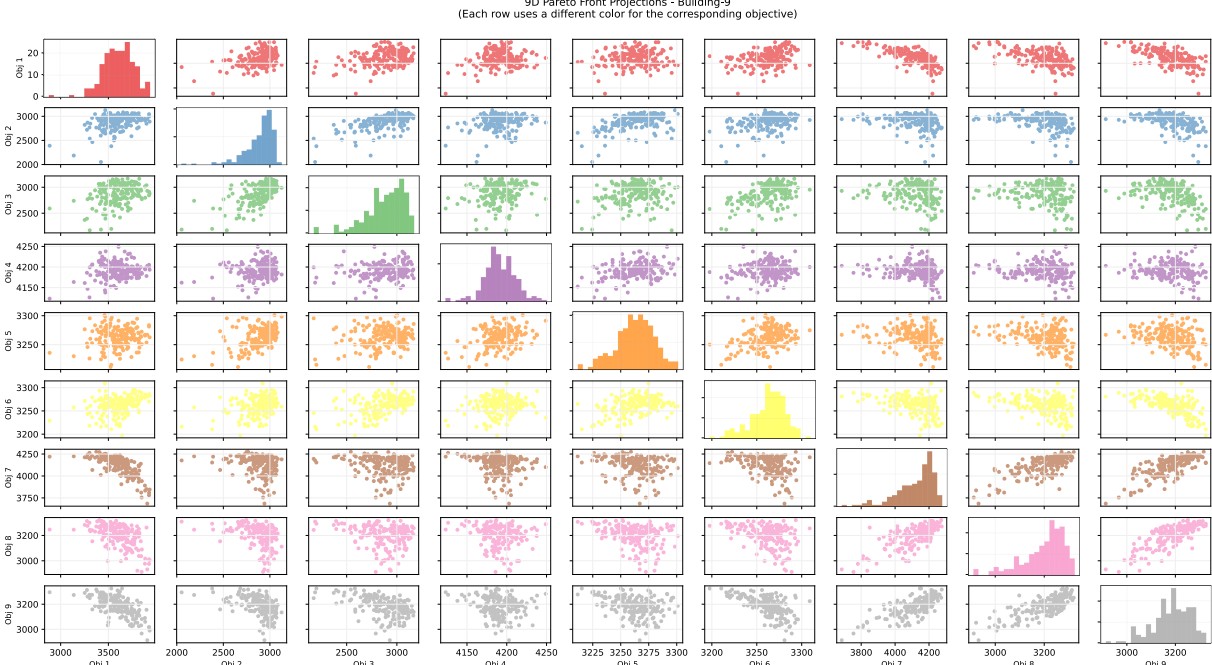

*Figure 4.* The Pareto front visualization of MPFT for Building-9. (HV: $8.33 \times 10^{31}$, SP: $1.74 \times 10^{3}$, EU: $3.53 \times 10^{31}$)

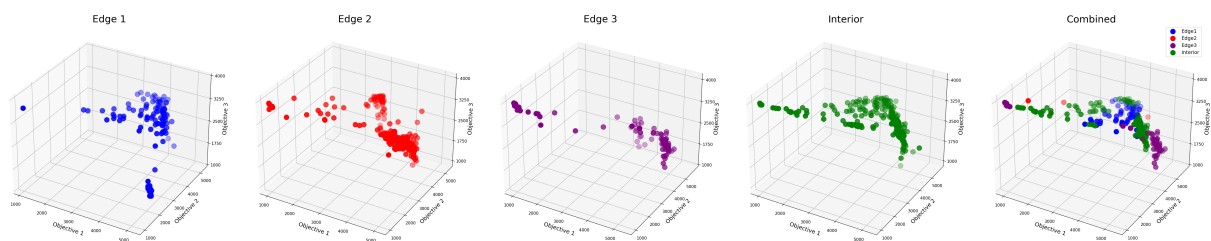

*Figure 5.* Pareto front tracking process of the MPFT-MOTD7 algorithm in the three-objective Hopper-3 environment. Ojective1: Forward reward; Objective2: Jumping reward; Objective3: Energy efficiency.

### G.1. Pareto-Approximation Fronts

We plot the Pareto fronts discovered by each algorithm for Hopper-3 environments in Figure 3. Additionally, we present a visualization of the Pareto front for Building-9 in Figure 4, demonstrating the Pareto-tracking mechanism's capability to track the Pareto front under high-dimensional objectives. The results demonstrate that the proposed MPFT framework can integrate both online and offline RL algorithms to obtain a Pareto-approximation policy set with SOTA performance. In particular, the MPFT, which incorporates the advanced offline RL algorithm TD7, achieves the highest HV and EU values and produces a densely tracked Pareto front.

Given that MPFT-MOTD7 achieves the best overall performance, the subsequent subsections focus on analyzing the MPFT framework based on this algorithm.

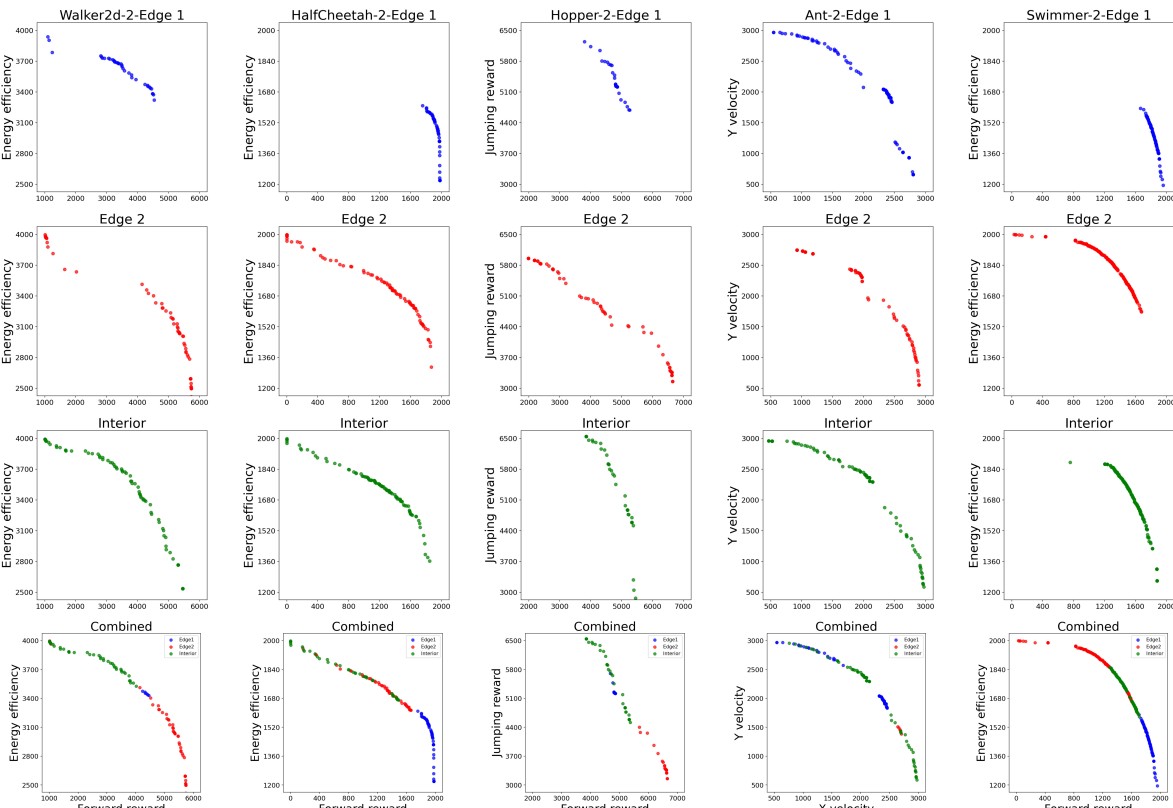

*Figure 6.* Pareto front tracking process for the MPFT-MOTD7 algorithm in all two-objective environments.

## G.2. MPFT Framework Tracking Analysis

In this subsection, we analyzed and visualized the tracked Pareto front for both Stage 2 and Stage 3. Figures 5 and 6 show the MPFT-MOTD7 algorithm tracking the Pareto-edge and Pareto-interior fronts in three-objective robotic control environment Hopper-3 and in all two-objective robotic control environments, respectively. The results indicate that the Pareto front tracked starting from the Pareto-vertex policies in Stage 2 is more densely distributed along the edge of the Pareto front, whereas the exploration of some Pareto-interior policies remains insufficient. Consequently, Stage 2 alone is insufficient to obtain a complete Pareto front, especially in challenging environments such as Hopper-2 and Hopper-3.

In Stage 3, the proposed objective weight adjustment method guides policy training toward the sparse regions of the tracked Pareto front, enabling the discovery of the Pareto-interior policies. These policies are then used as anchor points to track the Pareto front in the direction of each objective using the proposed Pareto-tracking mechanism. From the combined results in Figures 5 and 6, the sparse regions of the tracked Pareto front are visibly reduced after Stage 3. Moreover, our experimental results show that the proposed MPFT framework can independently trace a Pareto front starting from any Pareto-vertex policy. This is particularly advantageous in edge scenarios with limited hardware resources, as only a single Pareto front tracing is required rather than maintaining $m$ parallel tracks—an ability that evolutionary frameworks cannot readily achieve.

## G.3. Convergence Analysis

We do not compare the HV and SP learning curves of the MPFT framework based algorithms and the benchmarks in the same figure. This is because both HV and SP metrics can only be evaluated in Stage 4. But to verify the convergence of the MPFT framework, we present the HV learning curves for $\mathcal{F}_{edge}^i$ and $\mathcal{F}_{inter}^k$ to verify the convergence of the MPFT framework, which are traced in Stage2 and Stage3, respectively. The x-axis denotes the $\frac{\Psi_i}{u+v}$ or $\frac{\Psi_k}{u+v}$ episodes, the y-axis denotes the HV metric, and the shaded area represents the standard deviation calculated from five independent runs. The Hopper-3 curves are omitted because it is a three-objective environment and the Edge-3 figure cannot be properly aligned. From Figures 7, we can observe that each track, $\mathcal{F}_{edge}^i$ or $\mathcal{F}_{inter}^k$, eventually converges, indicating the eventual convergence

of the overall track set $\mathcal{F}$ traced by the MPFT framework. In addition, to ensure a fair comparison, the results of all evolutionary framework based benchmarks are reported after their convergence, and their HV learning curves are plotted in Figure 8. In this figure, the x-axis denotes $G$ in Tables 9 and 10, the y-axis denotes the HV metrics, and the shaded areas represent the standard deviations derived from five independent runs.

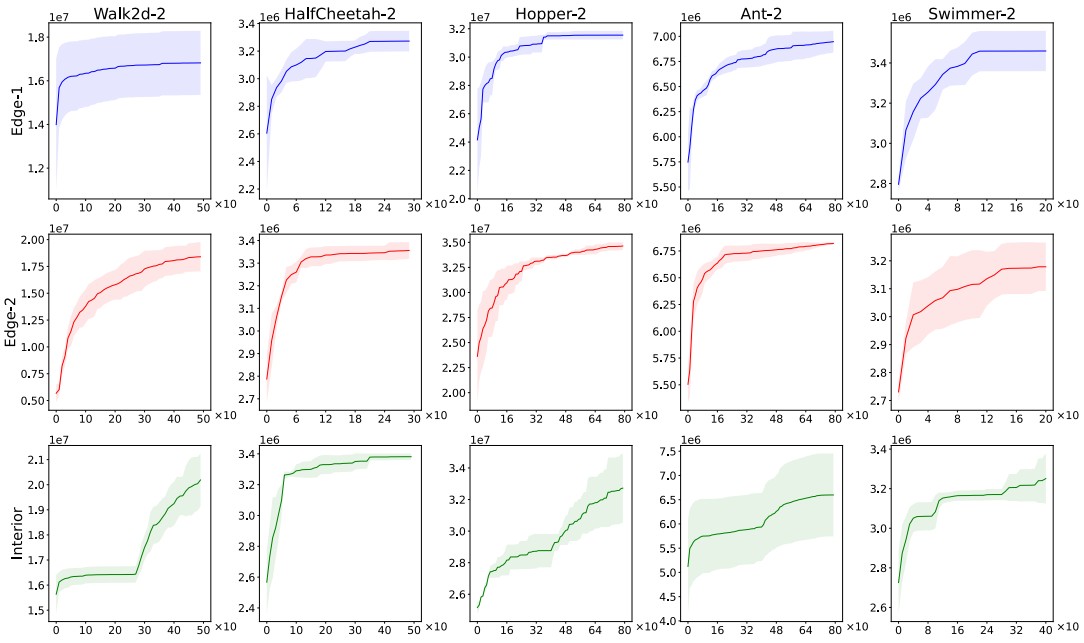

*Figure 7.* HV metric learning curves of the MPFT-MOTD7 algorithm on 2-objective robotic control environments.

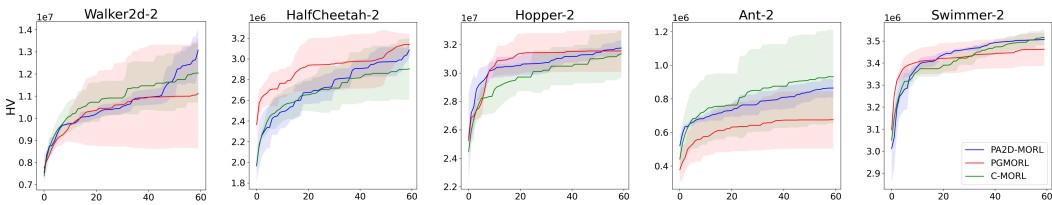

*Figure 8.* HV metric learning curves of all benchmarks on 2-objective robotic control environments.

### G.4. Sensitivity Analysis of Pareto Tracking Mechanism

We first analyze the sensitivity of the Pareto-tracking mechanism to the hyperparameters $u$ and $v$, in order to validate the conclusion $u < v$ derived in Appendix C. We fix $K = 1$ and keep all parameters consistent with Table 12, except for $u$ and $v$. Figure 9 reports the mean and standard deviation over five independent runs of MPFT-MOTD7 algorithm. We observe that the MPFT framework performs well when $u < v$, yielding stable HV and SP values, whereas the SP performance deteriorates when $u > v$. Specifically, the setting $u = 0, v = 2$ achieves the best performance in simpler environments (e.g., Swimmer-2 and HalfCheetah-2), while $u = 1, v = 2$ performs best in more complex environments (e.g., Ant-2). These results suggest that in simpler environments, a higher proportion of Pareto-ascent phase helps mitigate the excessive exploration induced by the Pareto-reverse phase, whereas in more complex environments, increasing the proportion of Pareto-reverse phase is beneficial for covering a broader Pareto front though its proportion should remain lower than that of

the Pareto-ascent phase.

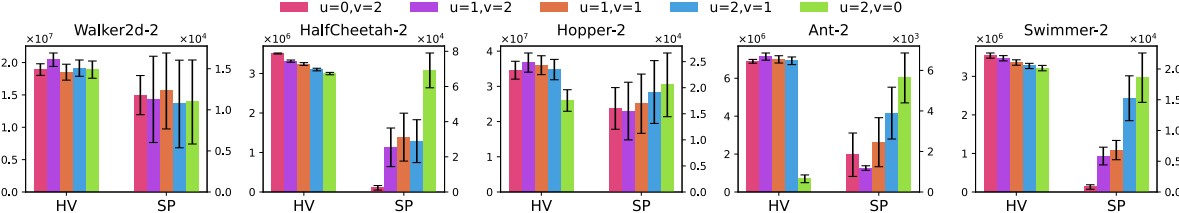

*Figure 9.* Performance of MPFT-MOTD7 under different $(u, v)$ settings on all 2-objective environments. Error bars denote $\pm 1$ standard deviation.

We further investigate the impact of the hyperparameter steps on the performance of the MPFT framework. Specifically, we fix $K = 1$ and keep all other settings identical to Table 12, except for steps. Figure 10 presents the mean and standard deviation over five independent runs of MPFT-MOTD7. The results show that increasing in steps has little effect on HV in relatively simple environments (e.g., Swimmer-2 and HalfCheetah-2), whereas more complex environments benefit from larger steps, particularly in terms of SP. However, once steps is sufficiently large for Pareto-tracking mechanism to converge, further increases steps provide diminishing returns on both HV and SP. Thus, selecting an appropriate steps is crucial for balancing efficiency and performance.

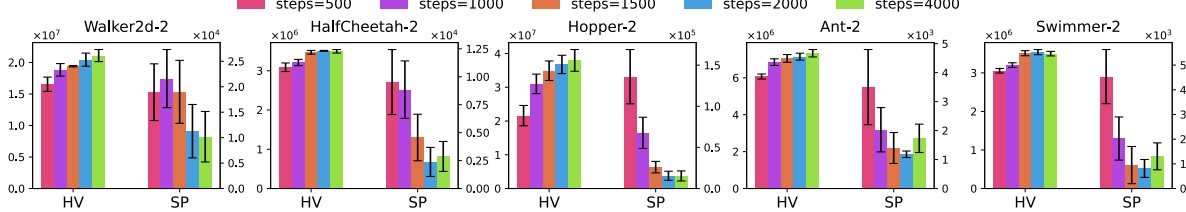

*Figure 10.* Performance of MPFT-MOTD7 under different steps settings on all 2-objective environments. Error bars denote $\pm 1$ standard deviation.

### G.5. Robustness Analysis of MPFT

In this subsection, we evaluate the robustness of MPFT with respect to four key factors: (i) the initialize parameters $\{\Xi_i, \Xi_k\}$, (ii) the tracking parameters $\{\Psi_i, \Psi_k\}$, (iii) the total interaction budget **env_steps**, and (iv) the policy-buffer size. We additionally compare the performance drop of SPFT relative to MPFT. All results are obtained through five independent runs.

*Table 13.* Evaluation of HV, SP, and EU for all 2-objective environments. 10% indicates that $\{\Xi_i\}$ and $\{\Xi_k\}$ is 10% of the corresponding $\{\Xi_i\}$ and $\{\Xi_k\}$ in Table 12.

| Environment | Metrics | 100% | 50% | 20% | 10% | Best Baseline |
|---|---|---|---|---|---|---|
| Walker2d-2 | HV $\uparrow (\times 10^7)$ | **2.044 $\pm$ 0.104** | **2.043 $\pm$ 0.528** | 1.853 $\pm$ 0.529 | 1.689 $\pm$ 0.243 | 1.308 $\pm$ 0.094 |
| | SP $\downarrow (\times 10^4)$ | 1.127 $\pm$ 0.525 | 1.012 $\pm$ 0.125 | 0.943 $\pm$ 0.066 | 0.901 $\pm$ 0.052 | **0.323 $\pm$ 0.132** |
| | EU $\uparrow (\times 10^3)$ | **4.443 $\pm$ 0.063** | **4.439 $\pm$ 0.077** | 3.966 $\pm$ 0.081 | 3.772 $\pm$ 0.048 | 3.474 $\pm$ 0.148 |
| HalfCheetah-2 | HV $\uparrow (\times 10^6)$ | **3.506 $\pm$ 0.013** | **3.510 $\pm$ 0.170** | 3.352 $\pm$ 0.030 | 3.339 $\pm$ 0.056 | 3.345 $\pm$ 0.037 |
| | SP $\downarrow (\times 10^4)$ | 0.238 $\pm$ 0.130 | 0.253 $\pm$ 0.111 | 0.240 $\pm$ 0.094 | 0.232 $\pm$ 0.078 | **0.024 $\pm$ 0.012** |
| | EU $\uparrow (\times 10^3)$ | **1.784 $\pm$ 0.010** | **1.786 $\pm$ 0.012** | **1.781 $\pm$ 0.015** | 1.773 $\pm$ 0.019 | 1.673 $\pm$ 0.155 |
| Hopper-2 | HV $\uparrow (\times 10^7)$ | **3.676 $\pm$ 0.272** | 3.636 $\pm$ 0.219 | 3.564 $\pm$ 0.215 | 3.396 $\pm$ 0.168 | 3.177 $\pm$ 0.052 |
| | SP $\downarrow (\times 10^4)$ | 1.552 $\pm$ 0.553 | 1.503 $\pm$ 0.965 | **1.498 $\pm$ 0.590** | 1.516 $\pm$ 0.518 | 2.281 $\pm$ 0.668 |
| | EU $\uparrow (\times 10^3)$ | **5.851 $\pm$ 0.148** | 5.740 $\pm$ 0.147 | 5.682 $\pm$ 0.145 | 5.550 $\pm$ 0.123 | 5.389 $\pm$ 0.045 |
| Ant-2 | HV $\uparrow (\times 10^6)$ | **7.044 $\pm$ 0.189** | 7.013 $\pm$ 0.183 | 6.816 $\pm$ 0.185 | 6.664 $\pm$ 0.232 | 0.865 $\pm$ 0.055 |
| | SP $\downarrow (\times 10^3)$ | 1.182 $\pm$ 0.113 | 1.363 $\pm$ 0.468 | 1.281 $\pm$ 0.227 | **1.141 $\pm$ 0.137** | 1.359 $\pm$ 0.347 |
| | EU $\uparrow (\times 10^3)$ | **2.467 $\pm$ 0.040** | **2.448 $\pm$ 0.033** | 2.427 $\pm$ 0.047 | 2.394 $\pm$ 0.039 | 0.883 $\pm$ 0.105 |
| Swimmer-2 | HV $\uparrow (\times 10^6)$ | **3.539 $\pm$ 0.067** | **3.505 $\pm$ 0.097** | 3.435 $\pm$ 0.109 | 3.406 $\pm$ 0.093 | 3.507 $\pm$ 0.004 |
| | SP $\downarrow (\times 10^3)$ | 0.814 $\pm$ 0.362 | 1.221 $\pm$ 0.471 | 1.054 $\pm$ 0.351 | 1.388 $\pm$ 0.537 | **0.188 $\pm$ 0.033** |
| | EU $\uparrow (\times 10^3)$ | **1.748 $\pm$ 0.017** | **1.746 $\pm$ 0.020** | 1.734 $\pm$ 0.012 | 1.711 $\pm$ 0.017 | 1.740 $\pm$ 0.001 |

**Impact of** $\{\Xi_i, \Xi_k\}$**.** Table 13 reports the effect of $\{\Xi_i, \Xi_k\}$ while keeping all other hyperparameters fixed. When the values are reduced to 50% of the default configuration in Table 12, performance degradation is negligible. When reduced to 10%, HV and EU decrease by at most 20%, while SP remains nearly unchanged. Despite this reduction, MPFT remains competitive relative to the best baselines. This behavior arises because $\{\Xi_i, \Xi_k\}$ directly influence the quality of Pareto-vertex and Pareto-interior policies; larger values allow MPFT to discover solutions further from the reference point, enabling more favorable initialization for Pareto front tracking.

*Table 14.* Evaluation of HV, SP, and EU for all 2-objective environments. 10% indicates that $\{\Psi_i\}$ and $\{\Psi_k\}$ is 10% of the corresponding $\{\Psi_i\}$ and $\{\Psi_k\}$ in Table 12.

| Environment | Metrics | 100% | 50% | 20% | 10% | Best Baseline |
|---|---|---|---|---|---|---|
| Walker2d-2 | HV $\uparrow$ ($\times 10^7$) | **2.043 ± 0.104** | 2.032 ± 0.092 | 1.880 ± 0.110 | 1.709 ± 0.129 | 1.308 ± 0.094 |
| | SP $\downarrow$ ($\times 10^4$) | 1.127 ± 0.525 | 1.652 ± 0.695 | 7.538 ± 2.017 | 19.27 ± 9.268 | **0.323 ± 0.132** |
| | EU $\uparrow$ ($\times 10^3$) | **4.443 ± 0.063** | 4.382 ± 0.011 | 4.207 ± 0.202 | 4.006 ± 0.191 | 3.474 ± 0.148 |
| HalfCheetah-2 | HV $\uparrow$ ($\times 10^6$) | **3.506 ± 0.013** | 3.462 ± 0.039 | 3.390 ± 0.087 | 3.293 ± 0.119 | 3.345 ± 0.037 |
| | SP $\downarrow$ ($\times 10^4$) | 0.238 ± 0.130 | 0.547 ± 0.366 | 3.296 ± 0.868 | 4.822 ± 2.591 | **0.024 ± 0.012** |
| | EU $\uparrow$ ($\times 10^3$) | **1.784 ± 0.010** | **1.784 ± 0.012** | **1.780 ± 0.006** | 1.772 ± 0.016 | 1.673 ± 0.155 |
| Hopper-2 | HV $\uparrow$ ($\times 10^7$) | **3.676 ± 0.272** | 3.539 ± 0.103 | 3.456 ± 0.333 | 3.375 ± 0.291 | 3.177 ± 0.052 |
| | SP $\downarrow$ ($\times 10^4$) | **1.552 ± 0.553** | 6.911 ± 2.934 | 12.862 ± 3.443 | 18.51 ± 6.317 | 2.281 ± 0.668 |
| | EU $\uparrow$ ($\times 10^3$) | **5.851 ± 0.148** | 5.691 ± 0.166 | 5.650 ± 0.119 | 5.501 ± 0.209 | 5.389 ± 0.045 |
| Ant-2 | HV $\uparrow$ ($\times 10^6$) | **7.044 ± 0.189** | 6.714 ± 0.283 | 5.681 ± 0.295 | 5.553 ± 0.467 | 0.865 ± 0.055 |
| | SP $\downarrow$ ($\times 10^3$) | **1.182 ± 0.113** | 5.492 ± 0.257 | 11.53 ± 3.210 | 13.08 ± 2.786 | 1.359 ± 0.347 |
| | EU $\uparrow$ ($\times 10^3$) | **2.467 ± 0.040** | 2.416 ± 0.034 | 2.283 ± 0.059 | 2.274 ± 0.070 | 0.883 ± 0.105 |
| Swimmer-2 | HV $\uparrow$ ($\times 10^6$) | **3.539 ± 0.067** | 3.498 ± 0.077 | 3.473 ± 0.075 | 3.348 ± 0.083 | 3.507 ± 0.004 |
| | SP $\downarrow$ ($\times 10^3$) | 0.814 ± 0.362 | 2.873 ± 1.658 | 8.102 ± 2.371 | 16.29 ± 4.741 | **0.188 ± 0.033** |
| | EU $\uparrow$ ($\times 10^3$) | **1.748 ± 0.017** | **1.746 ± 0.014** | **1.745 ± 0.014** | 1.739 ± 0.011 | 1.740 ± 0.001 |

**Impact of** $\{\Psi_i, \Psi_k\}$**.** Table 14 evaluates sensitivity to $\{\Psi_i, \Psi_k\}$. Reducing these values to 50% results in only mild performance drops; reducing them to 10% yields at most a 23% decrease in HV and EU, but noticeably affects SP. This is expected, as $\{\Psi_i, \Psi_k\}$ determine the density of discovered Pareto-approximate policies. Larger values allow MPFT to populate the Pareto set more densely, ensuring better front coverage. Even when reduced to 10%, MPFT maintains competitive HV and EU compared with the best baselines.

*Table 15.* Evaluation of HV, SP, and EU for all 2-objective environments. 10% indicates that **env_steps** is 10% of the corresponding **env_steps** in Table 1.

| Environment | Metrics | 100% | 50% | 20% | 10% | Best Baseline |
|---|---|---|---|---|---|---|
| Walker2d-2 | HV $\uparrow$ ($\times 10^7$) | **2.044 ± 0.104** | 2.031 ± 0.069 | 1.875 ± 0.039 | 1.701 ± 0.572 | 1.308 ± 0.094 |
| | SP $\downarrow$ ($\times 10^4$) | 1.127 ± 0.525 | 1.827 ± 0.042 | 8.332 ± 3.942 | 24.11 ± 8.091 | **0.323 ± 0.132** |
| | EU $\uparrow$ ($\times 10^3$) | **4.443 ± 0.063** | 4.382 ± 0.013 | 4.119 ± 0.018 | 4.002 ± 0.074 | 3.474 ± 0.148 |
| HalfCheetah-2 | HV $\uparrow$ ($\times 10^6$) | **3.506 ± 0.013** | 3.456 ± 0.030 | 3.339 ± 0.033 | 3.284 ± 0.062 | 3.305 ± 0.037 |
| | SP $\downarrow$ ($\times 10^4$) | 0.238 ± 0.130 | 0.552 ± 0.351 | 3.953 ± 1.865 | 5.422 ± 2.778 | **0.024 ± 0.012** |
| | EU $\uparrow$ ($\times 10^3$) | **1.784 ± 0.010** | 1.779 ± 0.027 | 1.774 ± 0.008 | 1.768 ± 0.012 | 1.673 ± 0.155 |
| Hopper-2 | HV $\uparrow$ ($\times 10^7$) | **3.676 ± 0.272** | 3.528 ± 0.249 | 3.404 ± 0.105 | 3.342 ± 0.268 | 3.177 ± 0.052 |
| | SP $\downarrow$ ($\times 10^4$) | **1.552 ± 0.553** | 4.399 ± 2.926 | 15.967 ± 3.853 | 19.97 ± 6.833 | 2.281 ± 0.668 |
| | EU $\uparrow$ ($\times 10^3$) | **5.851 ± 0.148** | 5.640 ± 0.163 | 5.625 ± 0.146 | 5.484 ± 0.196 | 5.389 ± 0.045 |
| Ant-2 | HV $\uparrow$ ($\times 10^6$) | **7.044 ± 0.189** | 6.677 ± 0.258 | 5.595 ± 0.248 | 5.520 ± 0.113 | 0.865 ± 0.055 |
| | SP $\downarrow$ ($\times 10^3$) | **1.182 ± 0.113** | 5.536 ± 0.768 | 12.43 ± 4.992 | 17.41 ± 4.340 | 1.359 ± 0.347 |
| | EU $\uparrow$ ($\times 10^3$) | **2.467 ± 0.040** | 2.410 ± 0.121 | 2.261 ± 0.038 | 2.249 ± 0.034 | 0.883 ± 0.105 |
| Swimmer-2 | HV $\uparrow$ ($\times 10^6$) | **3.539 ± 0.067** | 3.480 ± 0.103 | 3.340 ± 0.212 | 3.250 ± 0.236 | 3.507 ± 0.004 |
| | SP $\downarrow$ ($\times 10^3$) | 0.814 ± 0.362 | 3.119 ± 0.658 | 10.62 ± 4.582 | 16.42 ± 2.384 | **0.188 ± 0.033** |
| | EU $\uparrow$ ($\times 10^3$) | **1.748 ± 0.017** | **1.743 ± 0.018** | 1.728 ± 0.031 | 1.717 ± 0.039 | 1.740 ± 0.001 |

**Impact of** **env_steps.** To study interaction efficiency, we adjust $\{\Xi_i, \Xi_k\}$, $\{\Psi_i, \Psi_k\}$, and steps to reduce **env_steps** while keeping all other settings fixed. Table 15 shows that using only 50% of the default interaction budget results in slight performance degradation; using 10% leads to at most a 23% drop in HV and EU, while SP decreases more significantly. Since **env_steps** affects (i) the convergence of Pareto tracking (via steps), (ii) the quality of discovered Pareto-vertex/interior policies (via $\{\Xi_i, \Xi_k\}$), and (iii) the number of discovered Pareto-approximate policies (via $\{\Psi_i, \Psi_k\}$), a larger budget naturally leads to denser and higher-quality Pareto fronts. Importantly, in practical applications, evaluation typically focuses on EU rather than HV and SP; under this criterion, even at 10% of the default interaction budget, MPFT remains competitive

relative to the best baselines.

*Table 16.* Evaluation of HV, SP, and EU for all 2-objective environments under varying policy buffer sizes, alongside a comparison with the best-performing baseline. 100 indicates the policy buffer sizes (i.e., 100 policies model are stored after trianing).

| Environment | Metrics | 200 | 100 | 50 | 20 | Best Baseline |
|---|---|---|---|---|---|---|
| Walker2d-2 | HV ↑ ($\times 10^7$) | **2.044 ± 0.104** | **2.044 ± 0.104** | 2.029 ± 0.071 | 2.016 ± 0.377 | 1.308 ± 0.094 |
| | SP ↓ ($\times 10^4$) | 1.127 ± 0.525 | 1.127 ± 0.525 | 3.048 ± 0.960 | 14.88 ± 5.143 | **0.323 ± 0.132** |
| | EU ↑ ($\times 10^3$) | **4.443 ± 0.063** | **4.443 ± 0.063** | **4.407 ± 0.101** | 4.392 ± 0.074 | 3.474 ± 0.148 |
| HalfCheetah-2 | HV ↑ ($\times 10^6$) | **3.506 ± 0.013** | **3.488 ± 0.029** | 3.446 ± 0.074 | 3.424 ± 0.062 | 3.087 ± 0.173 |
| | SP ↓ ($\times 10^4$) | 0.238 ± 0.130 | 0.314 ± 0.139 | 0.772 ± 0.145 | 2.126 ± 0.105 | **0.024 ± 0.012** |
| | EU ↑ ($\times 10^3$) | **1.784 ± 0.010** | **1.783 ± 0.016** | **1.780 ± 0.009** | 1.777 ± 0.007 | 1.673 ± 0.155 |
| Hopper-2 | HV ↑ ($\times 10^7$) | **3.676 ± 0.272** | **3.651 ± 0.220** | **3.604 ± 0.105** | 3.559 ± 0.230 | 3.177 ± 0.052 |
| | SP ↓ ($\times 10^4$) | **1.552 ± 0.553** | 6.128 ± 2.977 | 10.26 ± 4.998 | 20.28 ± 6.761 | 2.281 ± 0.668 |
| | EU ↑ ($\times 10^3$) | **5.851 ± 0.148** | 5.752 ± 0.147 | 5.731 ± 0.134 | 5.644 ± 0.208 | 5.389 ± 0.045 |
| Ant-2 | HV ↑ ($\times 10^6$) | **7.044 ± 0.189** | 7.024 ± 0.261 | 7.016 ± 0.274 | 6.833 ± 0.242 | 0.865 ± 0.055 |
| | SP ↓ ($\times 10^3$) | **1.182 ± 0.113** | 5.835 ± 0.571 | 12.21 ± 1.884 | 18.04 ± 1.781 | 1.359 ± 0.347 |
| | EU ↑ ($\times 10^3$) | **2.467 ± 0.040** | **2.467 ± 0.039** | **2.464 ± 0.040** | 2.458 ± 0.035 | 0.883 ± 0.105 |
| Swimmer-2 | HV ↑ ($\times 10^6$) | **3.539 ± 0.067** | 3.498 ± 0.096 | 3.422 ± 0.177 | 3.308 ± 0.192 | 3.507 ± 0.004 |
| | SP ↓ ($\times 10^3$) | 0.814 ± 0.362 | 5.596 ± 1.569 | 9.665 ± 3.338 | 15.61 ± 3.253 | **0.188 ± 0.033** |
| | EU ↑ ($\times 10^3$) | **1.748 ± 0.017** | 1.738 ± 0.029 | 1.736 ± 0.001 | 1.725 ± 0.032 | 1.740 ± 0.001 |

**Impact of Policy Buffer Size.** Table 16 compares MPFT with varying policy-buffer sizes against the best baseline. Even when the buffer is substantially reduced, MPFT maintains consistently strong performance while using significantly fewer stored policies. In Walker2d-2, the results remain unchanged across buffer sizes, indicating that the Pareto-optimal set in this environment does not exceed the policy buffer capacities.

*Table 17.* Evaluation of HV, SP, and EU for all 2-objective environments, comparison with the SPFT and best-performing baseline.

| Environment | Metrics | MPFT | SPFT | Best Baseline |
|---|---|---|---|---|
| Walker2d-2 | HV ↑ ($\times 10^7$) | **2.044 ± 0.104** | **1.682 ± 0.180** | 1.308 ± 0.094 |
| | SP ↓ ($\times 10^4$) | 1.127 ± 0.525 | 6.156 ± 3.868 | **0.323 ± 0.132** |
| | EU ↑ ($\times 10^3$) | **4.443 ± 0.063** | **4.054 ± 0.028** | 3.474 ± 0.148 |
| HalfCheetah-2 | HV ↑ ($\times 10^6$) | **3.506 ± 0.013** | **3.310 ± 0.133** | 3.087 ± 0.173 |
| | SP ↓ ($\times 10^4$) | 0.238 ± 0.130 | 0.673 ± 0.338 | **0.024 ± 0.012** |
| | EU ↑ ($\times 10^3$) | **1.784 ± 0.010** | **1.755 ± 0.005** | 1.673 ± 0.155 |
| Hopper-2 | HV ↑ ($\times 10^7$) | **3.676 ± 0.272** | **3.252 ± 0.170** | 3.177 ± 0.052 |
| | SP ↓ ($\times 10^4$) | **1.552 ± 0.553** | 2.707 ± 1.610 | 2.281 ± 0.668 |
| | EU ↑ ($\times 10^3$) | **5.851 ± 0.148** | **5.420 ± 0.158** | 5.389 ± 0.045 |
| Ant-2 | HV ↑ ($\times 10^6$) | **7.044 ± 0.189** | **6.815 ± 0.146** | 0.865 ± 0.055 |
| | SP ↓ ($\times 10^3$) | **1.182 ± 0.113** | 5.413 ± 1.460 | 1.359 ± 0.347 |
| | EU ↑ ($\times 10^3$) | **2.467 ± 0.040** | **2.339 ± 0.185** | 0.883 ± 0.105 |
| Swimmer-2 | HV ↑ ($\times 10^6$) | **3.539 ± 0.067** | 3.460 ± 0.123 | 3.507 ± 0.004 |
| | SP ↓ ($\times 10^3$) | 0.814 ± 0.362 | 0.690 ± 0.305 | **0.188 ± 0.033** |
| | EU ↑ ($\times 10^3$) | **1.748 ± 0.017** | 1.739 ± 0.021 | **1.740 ± 0.001** |

**Comparison Between SPFT and MPFT.** Table 17 reports the performance difference between SPFT and MPFT. Despite using only a *single* policy for Pareto tracking, SPFT still achieves HV and EU performance that surpasses the best baseline in most environments, confirming the strong effectiveness of MPFT's Pareto-tracking mechanism.

**Overall Summary.** When $u < v$, appropriate steps is chosen to ensure convergence of the Pareto-tracking mechanism, and $\{\Xi_i, \Psi_i\}_{i=1}^m$ and $\{\Xi_k, \Psi_k\}_{k=1}^K$ are large, MPFT yields stable and competitive results. Moreover, under constrained computational or memory resources, MPFT can still produce competitive Pareto fronts by reducing **env_steps**, shrinking policy-buffer sizes, or using SPFT.

### G.6. Evaluation on Offline Datasets

To verify MPFT's adaptability to pure offline settings, we evaluated MOTD7 and MOSAC against MORvS and CQL on the D4MORL MO-Hopper dataset (the most challenging environment in the D4MORL benchmark). We operated under

a 1,500,000-step budget across 5 runs by directly sampling from the dataset, which replaces the Data Storage process in Algorithms 4 and 5. As shown in Table 18, MPFT demonstrates competitive performance. Notably, performance is better on the Expert dataset than the Amateur dataset, validating the MPFT framework's capability to leverage high-quality offline data.

*Table 18.* Evaluation of HV for D4MORL MO-Hopper dataset, comparison with the MORvS and CQL.

| Dataset | Behavioral policy | MORvS (Zhu et al., 2023) | CQL (Kumar et al., 2020) | MOSAC | MOTD7 |
|---|---|---|---|---|---|
| Expert (HV $\times 10^7$ ) | 2.09 | 1.98 | 1.56 | 1.87 | 2.04 |
| Amateur (HV $\times 10^7$) | 1.97 | 1.79 | 1.02 | 1.58 | 1.73 |

## H. Discussion and Limitations of MPFT

With the novel design of Pareto-tracking mechanism, and sparse regions filling method, the proposed MPFT framework offers a systematic and efficient way to approximate the Pareto front without maintaining a large policy population. Despite its effectiveness in reducing agent-environment interactions, we observe that in some environments the tracked Pareto front still shows sparse regions, indicating underexplored areas. To address this limitation, we plan to develop more advanced sparse regions filling strategies and provide stronger global convergence guarantees. Additionally, while MPFT integrates both online and offline RL, its performance depends on the underlying algorithms: PPO benefits from a simpler network architecture, enabling faster training, but suffers from low sample efficiency and limited performance, whereas SAC and TD7 achieve higher efficiency and performance at the expense of greater training resources. In the future, we plan to explore more lightweight and sample-efficient algorithms within MPFT to further enhance its practicality, such as the meta-policy, mixture-of-experts architectures, and multi-head networks can be adopted in the network architecture.

In addition, a smaller limitation of the MPFT, common to deep learning methods, is the potential destruction of representation due to Pareto-reverse gradients, which push the network to intentionally unlearn a policy by sacrificing an objective. However, architectures with supplementary state and action representations, such as the SALE component in MOTD7, mitigate this issue.

From a theoretical perspective, our convergence analysis in Appendix C relies on a key technical assumption (Assumption A1). Such assumptions are common in optimization theory to establish local convergence rates and typically hold in locally convex regions. However, given the inherently non-convex landscape of deep MORL objectives—with potential saddle points and flat regions—a universal guarantee of this assumption cannot be established, despite strong empirical performance across diverse tasks; relaxing this assumption under weaker conditions remains an important direction for future research.

---

**Pseudocode 1. MOPPO Network Structure**

**Variables:**
```
hidden_dim = 256
```
**State Value Network $V(s; \phi)$:**
```
L0 = Linear(state_dim, hidden_dim)
L1 = Linear(hidden_dim, hidden_dim)
L2 = Linear(hidden_dim, m)
```
$V(s; \phi)$ **Forward Pass:**
```
input = state;  x = Tanh(L0(input))
x = Tanh(L1(x)); value = L2(x)
```
**Policy Network $\pi_\theta$:**
```
L0 = Linear(state_dim, hidden_dim)
L1 = Linear(hidden_dim, hidden_dim)
L2 = Linear(hidden_dim, action_dim)
Log_Std = Parameter(zeros(action_dim))
```
$\pi_\theta$ **Forward Pass:** `input = state`
```
x = Tanh(L0(input))
x = Tanh(L1(x))
mean = Tanh(L2(x))
std = exp(Log_Std).clamp(1e-6,20)
action ∼ Normal(mean, std)
```

---

**Pseudocode 2. MOSAC Network Structure**

```
Variables: hidden_dim = 256
```
**State Value Network $V(s; \phi)$:**
```
▷ MOSAC uses two state value networks each with the same network and forward pass.
L0 = Linear(state_dim, hidden_dim); L1 = Linear(hidden_dim, hidden_dim)
L2 = Linear(hidden_dim, m)
```
**$V(s; \phi)$ Forward Pass:**
```
input = state; x = ReLU(L0(input))
x = ReLU(L1(x)); value = L2(x)
```
**State-Action Value Network $Q(s, a; \varphi)$:**
```
▷ MOSAC uses two state-action value networks each with the same network and forward
pass.
L0 = Linear(state_dim + action_dim, hidden_dim)
L1 = Linear(hidden_dim, hidden_dim); L2 = Linear(hidden_dim, m)
```
**$Q(s, a; \varphi)$ Forward Pass:**
```
input = concatenate([state, action]); x = ReLU(L0(input))
x = ReLU(L1(x)); value = L2(x)
```
**Policy Network $\pi_{\theta}$:** ```L0 = Linear(state_dim, hidden_dim)```
```
L1 = Linear(hidden_dim, hidden_dim); Mean = Linear(hidden_dim, action_dim)
Log_Std = Linear(hidden_dim, action_dim)
```
**$\pi_{\theta}$ Forward Pass:**
```
input = state; x = ReLU(L0(input)); x = ReLU(L1(x))
mean = Mean(x)
std = exp(Log_Std(x)).clamp(1e-6,20)
action ~ Normal(mean, std)
```

---

**Pseudocode 3. MOTD7 Network Structure**

```
Variables: zs_dim = 256, hidden_dim = 256
```
**State-Action Value Network $Q^{(z_s, z_{sa})}(s, a; \varphi)$:**
```
▷ MOTD7 uses two state value networks each with the same network and forward pass.
L0 = Linear(state_dim + action_dim, hidden_dim)
L1 = Linear(zs_dim * 2 + hidden_dim, hidden_dim)
L2 = Linear(hidden_dim, hidden_dim)
L3 = Linear(hidden_dim, m)
```
**$Q^{(z_s, z_{sa})}(s, a; \varphi)$ Forward Pass:**
```
input = concatenate([state, action])
x = AvgL1Norm(L0(input)); x = concatenate([zsa, zs, x])
x = ReLU(L1(x)); x = ReLU(L2(x)); value = L3(x)
```
**Policy Network $\pi_{\theta}$:**
```
L0 = Linear(state_dim, hidden_dim)
L1 = Linear(zs_dim + hidden_dim, hidden_dim)
L2 = Linear(hidden_dim, hidden_dim)
L3 = Linear(hidden_dim, action_dim)
```
**$\pi_{\theta}$ Forward Pass:**
```
input = state; x = AvgL1Norm(L0(input)); x = concatenate([zs, x])
x = ReLU(L1(x)),x = ReLU(L2(x))
action = tanh(L3(x))
```
**State Encoder Network $f$:**
```
L1 = Linear(state_dim, hidden_dim)
L2 = Linear(hidden_dim, hidden_dim)
L3 = Linear(hidden_dim, zs_dim)
```
**$f$ Forward Pass:**
```
input = state; x = ReLU(L1(input))
x = ReLU(L2(x)); zs = AvgL1Norm(L3(x))
```
**State-Action Encoder Network $g$:**
```
L1 = Linear(action_dim + zs_dim, hidden_dim); L2 = Linear(hidden_dim, hidden_dim)
L3 = Linear(hidden_dim, zs_dim)
```
**$g$ Forward Pass:**
```
input = concatenate([action, zs])
x = ReLU(L1(input)); x = ReLU(L2(x)); zsa = L3(x)
```

