# OpenReview forum: "Population-Free Pareto Tracking for Sample-Efficient Multi-Policy MORL"
_ICML.cc/2026/Conference — ICML 2026 regular_

### Official Review · Reviewer_gxQ3 · 2026-03-08

**Soundness:** 3
**Presentation:** 3
**Significance:** 3
**Originality:** 4
**Overall Recommendation:** 5
**Confidence:** 5

**Summary:**

This paper proposes a general purpose multi-policy pareto front tracking algorithm that can efficiently find policies to approximate the pareto front. The solution is specifically applicable for multi-policy MORL. Along with this, they propose MO-TD7, a multi-policy version of the TD7 algorithm and integrate it with MPFT. A specific detail about their algorithm is that it can integrate with any single-policy MORL algorithm (MOPPO, MOSAC, etc) and make it multi-policy.  Their contributions can be summarized as a multi-policy pareto front tracking algorithm, which has 4 steps. MPFT finds vertex points (points which maximise a single objective) and then find interior points. To do so, they solve an optimization problem by suppressing one objective and finding a policy that can maximise the others. Then they fill the sparse regions in the pareto front tracked so far. The final step is to combine all solutions.

Their solution is generalizable and works with both offline MORL and online MORL. The results show that MPFT outperforms SOTA.

**Compliance With Llm Reviewing Policy:**

Affirmed.

**Final Justification:**

After the rebuttal and clarifications, I am raising my score to 5 (accept). The authors have provided experiments with offline datasets which have nullified the weakness I raised that the authors mention offline compatibility but never demonstrate. The comparison with cmorl is also fair and I agree with the explanations from the authors. I congratulate the authors for their commendable paper and look forward to their code release!

**Key Questions For Authors:**

Please address the weaknesses

**Limitations:**

Yes

**Strengths And Weaknesses:**

**Strengths**

1. I am impressed with the level of work done by the authors. They have thoroughly analyzed every possible detail of their proposed algorithm and gone out of their way to provide an implementation for TD7.
2. The framework is clearly generalizable and extensible. This provides a pathway to integrate other Single-policy MORL algorithms to a multi-policy framework. Single-policy algorithms have clear limitations of collapse and suboptimal behavior and this can solve it.
3. This solves a major sample-efficiency problem faced by existing MP-MORL algorithms.
4. It is surprising that MPFT works with FruitTree, when the assumption is that the pareto front is continuous in the parameter space, while these environments have a highly discrete and discontinuous pareto front.

**Weaknesses**

1. While the authors claim it works with offline datasets, they do not show any experiments. The authors simply state that the D4MORL and offline-MOO datasets were constructed using the PGMORL algorithm. Immediately after that, they make a blanket claim: "They are not proper for evaluating the performance of the MOSAC and the MOTD7". I am not sure why? This makes it seem like they overclaim their contribution of working with offline datasets without results to prove it.

2. I am not fully convinced with the env_steps calculation for the baselines. From what I read about C-MORL, it seems like an extremely sample efficient algorithm itself and they evaluate on a maximum of 2.5 million environment interactions to get their results. I don't believe it will perform so poorly with 10x to 100x more interactions. Could the authors clarify if they have manually counted the environment interactions by incrementing a counter whenever env.step() was called, or only through their formula? (C-MORL also caps the episode length for hopper, cheetah and ant at 500, while gymnasium uses 1000.)

3. It is also not clear why the chosen environments are more difficult. They look like the standard versions in MO-gymnasium (they also use v5 versions of mujoco), could the authors clarify why they are more difficult?

4. No code has been provided in the supplementary materials or any qualitative analysis of the rollouts after training. I would like to see the rollout videos to compare the differences between the baselines and proposed MPFT.

A smaller weakness of the approach that I expect any deep learning method to have is the destruction of representation due to the pareto reverse gradients. It is possible that the entire network representation is destroyed because we are pushing the network to unlearn the policy it learned by intentionally sacrificing the policy it learned for that vertex. Also, training vertex policies to optimize for one specific objective might cause degenerate policies where individual objectives might mean nothing. However, I do not flag these as critical weaknesses as there is no such environment to my knowledge that breaks due to the assumptions made by the authors. These could be mentioned in the limitations section.

---

Overall, I do not find any theoretical or methodological weakness in the proposed contribution. The major concerns lies in the claim of being compatible with offline datasets with no substantiation (missing experiments and blanket claim that D4MORL is not compatible). If the authors address the 3 key weaknesses raised, I am inclined to raise my score.

**Minor errors (Should be corrected for camera-ready)**

1. The running title is still "Submission and Formatting Instructions for ICML 2026", this should be changed
2. Figure 2 b is half-cheetach, it should be half-cheetah
3. Label objective axes with the real names rather than Objective 1, 2 ...


---

After the rebuttal and clarifications, I am raising my score to 5 (accept). The authors have provided experiments with offline datasets which have nullified the weakness I raised that the authors mention offline compatibility but never demonstrate. The comparison with cmorl is also fair and I agree with the explanations from the authors. I congratulate the authors for their commendable paper and look forward to their code release!

---

> ### Author Rebuttal · Authors · 2026-03-30
>
> We sincerely thank you for recognizing the value and contributions of our work.
>
> **Response to Weakness gxQ3.1:**
>
> We apologize for the clarity issue. We initially excluded D4MORL because PGMORL trajectories often contain local optima, hindering high-quality Pareto-vertex/interior policies discovery. However, we agree that demonstrating offline capability is crucial. Due to the time constraints of the discussion phase, we focused our new experiments on MO-Hopper—the most challenging dataset in D4MORL—to verify MPFT's adaptability to offline settings (1.5M steps, 5 runs). By directly sampling from the dataset (replacing our Data Storage process in Algorithms 4 and 5), MPFT performs competitively even in pure offline scenarios.
> | Dataset | Behavioral policy | MORvS [1] | CQL [2] | MOSAC | MOTD7 |
> | :---: | :---: | :---: | :---: | :---: | :---: |
> | Expert (HV $\times10^{7}$ ) | 2.09 | 1.98 | 1.56 | 1.87 | 2.04 |
> | Amateur (HV $\times10^{7}$) | 1.97 | 1.79 | 1.02 | 1.58 | 1.73 |
>
> Notably, our method performs better on the Expert dataset than on the Amateur dataset, which further validates our initial concerns regarding the dataset quality.
>
> [1] Zhu B, et al. Scaling Pareto-Efficient Decision Making via Offline Multi-Objective RL, ICLR 2023.
>
> [2] Kumar A, et al. Conservative q-learning for offline reinforcement learning, NeurIPS, 2020.
>
> **Response to Weakness gxQ3.2:**
>
> Thanks for your comment. Our theoretical steps match actual env.step() counts in code. We set a larger step budget primarily to ensure fair comparisons with PGMORL and PA2D-MORL. We fully acknowledge C-MORL's exceptional sample efficiency; however, merely increasing its training steps does not yield significant gains in our setting. Because C-MORL's PPO-based architecture relies on clipping for stability rather than Q-value guidance, its exploration tends to plateau at local optima within our more challenging environments (see Weakness 3). To provide a direct comparison, we ran 5 Ant experiments using C-MORL's exact original configuration (v4, episode length 500, identical rewards) under a 1.5M step budget:
> | Method | HV $\uparrow$ ($\times 10^5$) | SP $\downarrow$ ($\times 10^3$) | EU $\uparrow$ ($\times 10^2$) |
> | :---: | :---: | :---: | :---: |
> | C-MORL | 3.13 | 1.67 | 4.29 |
> | Ours | 12.3 | 4.91 | 8.18 |
>
> These results demonstrate that MPFT maintains stronger performance even under identical, lighter task configurations.
>
> **Response to Weakness gxQ3.3:**
>
> Thanks for your comment. Our environments are harder because: 1) We use Mujoco v5, whereas C-MORL uses v4 (based on code file: https://github.com/RuohLiuq/C-MORL/tree/main/scripts. They are using version 1.1.0 of MO-gymnasium, which only supports v4.), and PGMORL/PA2D-MORL use v2; 2) We amplified objective conflicts via reward shaping. In Ant, our rewards are $0.35 \times \\{v_x, v_y\\} - C_{cost} + R_{alive}$, whereas C-MORL ignores $C_{cost}$ and the 0.35 speed multiplier; 3) We increased maximum episode length from 500 to 1000. This drastically increases the probability of falling in later stages, making it much harder for suboptimal policies to survive and accumulate high rewards.
>
> **Response to Weakness gxQ3.4:**
>
> Thanks for your comment. Due to ICML policies and desk-reject risks (anonymous links strictly permit only images/tables), we cannot provide videos now. However, we firmly commit to open-sourcing code (MPFT, all baselines, all environments, and rollout video) upon acceptance (if possible). Qualitatively, Ant videos show MPFT learns a highly optimized gait: moving quickly on three legs while raising one to minimize ground contact and energy. In contrast, baselines typically crawl slowly on all four legs, even losing balance before 1000 steps. This validates our Response gxQ3.2: without Q-value guidance, PPO easily gets trapped in suboptimal local minima (e.g., slow crawling to minimize control costs).
>
> **Response to Weakness gxQ3.5:**
>
> Thanks for your comment. We agree with your assessment. MOTD7 performs best primarily due to its SALE component. By providing supplementary state and action representation, SALE mitigates the "destruction of representation" during Pareto-reverse updates, making MOTD7 highly adaptable to MPFT. We will also discuss this representation destruction as a broader deep learning limitation in our revised "Limitations" section. Ablation (Ant, 1.5M steps, 5 runs):
>
> | Method | HV $\uparrow$ ($\times 10^6$) | SP $\downarrow$ ($\times 10^4$) | EU $\uparrow$ ($\times 10^3$) |
> | :---: | :---: | :---: | :---: |
> | Ours | 5.17 | 1.76 | 2.23 |
> | -SALE | 3.09 | 5.21 | 1.57 |
>
> **Response to Minor errors gxQ3.1-3:**
>
> Thanks for your comment. We have updated the running title, correct the "Halfcheetach" typo, and properly label the real objective names on the axes.
>
> *We sincerely hope that these explanations and additional results address your concerns. We look forward to hearing from you.*

---

> > ### Author Rebuttal · Reviewer_gxQ3 · 2026-03-31
> >
> > Thank you for the rebuttal. The authors have done a great job in addressing my questions, I have a few follow up questions and comments before I can raise the score
> >
> > 1. Thank you for the offline evaluation results. By 1.5M steps, do you mean there were 1.5M interactions/transitions present in the offline dataset or did you perform 1.5M online interactions after learning from the offline dataset? Also, why do you say Hopper is the most challenging? From what I know, humanoid is the most challenging because it is extremely difficult to get the correct gait motion.
> >
> > 2. I see that C-MORL uses a gamma=0.99 for its evaluation. Have you used gamma=1.0 for your evaluations of MPFT or 0.99? That will change the results drastically. The simple fix is to use 0.99 for MPFT with your saved weights since it is not possible to rerun the C-MORL experiments in this short duration. https://github.com/RuohLiuq/C-MORL/blob/main/scripts/hopper-2d.py Line 36 shows their gamma setting.
> >
> > 3. C-MORL does not exlude the control cost since it is baked into the reward functions for v4 and v5 (https://github.com/Farama-Foundation/MO-Gymnasium/blob/main/mo_gymnasium/envs/mujoco/ant_v5.py). Is the 0.35 a deliberate choice? If so, what does it entail. I went through your appendix and it does not seem the reward functions make the task harder (Please correct me if I am wrong)
> >
> > 4. Have you applied some kind of action processing in the environment after the baselines provide their actions?
> >
> > W4-W5 have been fully addressed, thank you.
> >
> > I commend the work put by the authors in the rebuttal. I am probing slightly more because I am concerned about the evaluation protocol. If the authors could confirm the specific experimental settings, environment choices made, I will be happy to raise my score.

---

> > > ### Author Response · Authors · 2026-04-01
> > >
> > > Dear Reviewer gxQ3:
> > >
> > > Thank you very much for your thoughtful follow-up questions and for carefully reviewing our previous rebuttal. Your sharp observations have been incredibly helpful in refining our work. Here are our detailed responses to your new questions:
> > >
> > > **Response to Q1:**
> > >
> > > Thank you for your attention to the experimental details. Regarding the 1.5M steps, this refers to the number of training steps on the offline dataset. For the online evaluation, we did not impose an additional online interaction limit. Instead, we evaluated all policies saved in the policy buffer during training.
> > >
> > > Regarding the most challenging environment, you are correct. In the broader context of RL (especially single-objective RL), Humanoid is indeed widely recognized as the most challenging task due to its extremely high degrees of freedom and complex dynamics. When we stated that "Hopper is the most challenging environment," we specifically meant within the testing environments included in the D4MORL offline dataset benchmark [1]. The original D4MORL paper does not provide an offline dataset or evaluation for Humanoid, making Hopper the most difficult task within this benchmark. We apologize for the confusion caused by our lack of precision and will explicitly add the qualifier "in the D4MORL benchmark" in our revised version.
> > >
> > > [1] Zhu B, et al. Scaling Pareto-Efficient Decision Making via Offline Multi-Objective RL, ICLR 2023.
> > >
> > > **Response to Q2:**
> > >
> > > Thank you very much for your insightful reminder. Upon careful inspection, we identified an implementation oversight in the evaluation code for the new Ant-v4 environment, which led to inflated performance values. Following your suggestion, we re-evaluated MPFT using our saved weights, strictly applying the $gamma = 0.99$ decay. The corrected experimental results are reported in the table below
> > > | Method | HV $\uparrow$ ($\times 10^5$) | SP $\downarrow$ ($\times 10^3$) | EU $\uparrow$ ($\times 10^2$) |
> > > | :---: | :---: | :---: | :---: |
> > > | C-MORL | 3.13 | 1.67 | 4.29 |
> > > | Ours | 3.45 | 1.10 | 4.46 |
> > >
> > > As shown, after adopting a completely fair setting, Our results have returned to the same numerical range as those of C-MORL. More importantly, even under the exact same setting as C-MORL, MPFT consistently demonstrates superior performance. Thank you again for helping us identify this evaluation code error, which ensured the rigor of our comparison. We sincerely apologize for our previous oversight.
> > >
> > > **Response to Q3:**
> > >
> > > Thank you for the correction. We carefully reviewed the MO-Gymnasium v4/v5 source code, and you are correct that the control cost is already baked into the reward functions. As the C-MORL paper does not explicitly mention this, it led to our previous misunderstanding, for which we sincerely apologize. Regarding multiplying the velocity reward by $0.35$, this was a deliberate design choice to increase the task's difficulty. We will add the following detailed motivation in our revised appendix:
> > >
> > > Specifically, during initial training, the agent inevitably performs large random exploratory actions, causing the energy penalty ($C_{cost}$) to dominate the total reward. In our setting, the dominance of the energy penalty creates a strong local optimum corresponding to low-movement behaviors. As a result, the agent tends to learn a conservative "standing still" policy. By multiplying the velocity reward by $0.35$, we substantially raise the threshold to offset this energy penalty through simple forward movement. This forces the agent to explore highly coordinated, energy-efficient gaits to achieve larger returns.
> > >
> > > We observed that baselines may get trapped in this "high penalty, low velocity" local optimum due to PPO's limitations. PPO’s value function only evaluates the current policy rather than providing action-wise comparisons, making it difficult to identify better actions beyond the current behavior, unless pure random exploration coincidentally yields a complete high-return trajectory. Even then, PPO's clip() operation aggressively restricts large updates, inadvertently discarding the high-value signals necessary to escape.
> > >
> > > In contrast, Q-learning-based methods (e.g., TD7, SAC) learn an explicit action-value function, which enables action ranking. Through TD learning and off-policy replay, high-return trajectories—although rare—can be propagated backward to guide policy improvement. This difference fundamentally improves the ability to escape local optima induced by short-term penalties, even if initial exploration incurs high energy penalties.
> > >
> > > The above analysis encapsulates our perspective on how this specific reward scaling increases the environment's difficulty. We believe these clarifications fully address your concerns.
> > >
> > > ------
> > >
> > > Thank you for your strong support and raising the score to Accept! We will release our code upon publication. We deeply appreciate your time and invaluable guidance.

---

### Official Review · Reviewer_orbi · 2026-03-09

**Soundness:** 2
**Presentation:** 3
**Significance:** 2
**Originality:** 3
**Overall Recommendation:** 4
**Confidence:** 3

**Summary:**

This work proposed a new framework for the Multi objective Reinforcement Learning problem using Multi Policy Pareto Front Tracking mechanism. In summary, the idea is starting from multiple initial policies, which lay on the approximated Pareto frontier, crawling over the frontier using Pareto-reverse and Pareto-ascent direction.

The word "Population-Free" might be confused due to the fact that this work do not use the evolutionary algorithm. Therefore, it is not need for the population pools.

**Compliance With Llm Reviewing Policy:**

Affirmed.

**Final Justification:**

After the rebuttal phased, I decide to raise the score to 4.

**Key Questions For Authors:**

- Instead of starting from the vertex and fill the sparse space later, could I random some policies, optimize them in one direction to push them closer to the frontiers and using them to track the Pareto front? Will that have better results?
- In Appendix G (Figure 5 and Figure 6), I notice that for most of the time, the interior section is spread over until one of the vertex. Could I assume that the closer the starting policies of the tracking to the elbow/utopia points on the frontiers, the better the tracking?
- Do you have any assumption for the large gap between the baselines and the MPFT method on the Ant-2 datasets? For other datasets, the MPFT method is still outperform, but the gap is not that large.
- Could you explain more about the SPFT settings? I think it is not mentioned on the Appendix F.2. I am confused this settings with the policies in illustration in the Figure 1.

**Limitations:**

No discussion for potential negative societal impact

**Strengths And Weaknesses:**

### Strength
The paper provides the analysis for the exponential convergence when tracing the frontier from the starting policy. The paper is well-struct and easy to follow.
### Weakness
The quality of the front is depended on the nature of the frontier and the initial policies, which is not analyzed in the papers. In general, I assume that this method works on the convex frontiers with good enough initialization, but for non-convex or complex frontiers and high dimension spaces, the efficiency is questionable.

The evaluation for time complexity is quite weird. The time complexity of this work, as well as other baseline methods, are estimated with the hyper-parameters as input. Although in the experimental results, the authors show that the proposed methods achieves better results with lower number of steps, I think this should not be the standard evaluation for time complexity. Maybe settings for the same budgets of actual running time for comparison is more convincing.

---

> ### Author Rebuttal · Authors · 2026-03-30
>
> Thank you for your comments. We used "Population-Free" to contrast our framework with evolutionary MP-MORL methods that require policy populations. We will clarify this in the revised Introduction.
>
> **Response to Weakness orbi.1:**
>
> MPFT performs "distributed exploration" by launching multiple tracking trajectories from different extreme policies. If one trajectory gets trapped in a non-convex or disconnected region, others can still cover the remaining frontier. Coupled with sparse region filling (Stage 3) that patches gaps, this enables MPFT to accurately and efficiently find the Pareto front in complex spaces.
>
> Empirically, we highlight that even a single tracking trajectory is highly effective. As shown in Table 3, our single-trajectory variant SPFT efficiently achieves superior performance in B-9 (high-dimensional) and FT-6. As Reviewer gxQ3 explicitly acknowledged, FT-6 features a highly discrete and non-continuous Pareto front, strongly validating our efficiency in complex spaces. Finally, Table 13 shows that degrading initial policies drops HV/EU by <20% with minor SP impact. Thus, MPFT achieves efficient tracking even under poor initialization and complex, non-convex frontiers.
>
> **Response to Weakness orbi.2:**
>
> We use $env\\_steps$ to evaluate sample efficiency following standard RL metrics (time complexity analysis is in Appendix D.4). We agree wall-clock time is also important. We conducted 5 runs on Walker2d under the same 5-hour budget on identical hardware. As shown below, MPFT maintains SOTA performance in HV and EU.
> | Method | HV $\uparrow$ ($\times 10^7$) | SP $\downarrow$ ($\times 10^4$) | EU $\uparrow$ ($\times 10^3$) |
> | :--- | :--- | :--- | :--- |
> | Ours | 1.91 | 4.32 | 4.23 |
> | PGMORL | 0.99 | 2.21 | 3.04 |
> | PA2D-MORL | 1.12 | 0.98 | 3.27 |
> | C-MORL | 1.16 | 1.06 | 3.29 |
>
> While our SP is relatively inferior here due to incomplete convergence under the strict time limit, MPFT significantly outperforms all baselines in EU—a far more critical metric for deployments.
>
> **Response to Question orbi.1:**
>
> To test this, we conducted 5 runs in Walker2d. In Stage 3, instead of updating toward sparse boundaries, we initialized random policy and optimized it along a fixed weight direction [0.5, 0.5] to serve as anchor.
> | Method | HV $\uparrow$ ($\times 10^7$) | SP $\downarrow$ ($\times 10^4$) | EU $\uparrow$ ($\times 10^3$) |
> | :--- | :--- | :--- | :--- |
> | Ours | 2.04 | 1.13 | 4.44 |
> | Fixed | 1.96 | 1.75 | 4.17 |
>
> The results indicate that simply pushing random policy along fixed weights makes it difficult to land precisely in the "sparse region" of the frontier. The policy often converges to dense areas already covered by existing tracks. Starting new tracks from this redundant point leads to high policy overlap (reflected in the degraded SP).
>
> **Response to Question orbi.2:**
>
> This phenomenon occurs because our tracking budget $\Psi$ is set significantly higher than the initial exploration budget $\Xi$ to observe asymptotic convergence (the large $\Psi$ is redundant, per Table 14). It will not occur when $\Psi$ is small. We agree with your hypothesis: the closer the starting policies are to the "elbows'' of the frontier, the better the tracking quality. This ability to find high-quality starting points depends largely on the optimization power of the underlying RL algorithm. This is why MOTD7 outperforms MOSAC/MOPPO in our framework. MPFT’s decoupled architecture allows it to seamlessly integrate more advanced RL algorithms (like TD7) to further push the boundaries of tracking quality. This is also one of our key contributions.
>
> **Response to Question orbi.3:**
>
> The gap on Ant-2 is driven by high-dimensional complexity ($\mathcal{S} \in \mathbb{R}^{105}, \mathcal{A} \in \mathbb{R}^{8}$) and reward shaping (scaling velocity reward by 0.35, Appendix F.1) to increase difficulty. MOPPO often hit local minima, prioritizing energy saving over velocity. Conversely, Q-value-guided offline architectures in MPFT (MOTD7/MOSAC) provide superior exploration, enabling the agent to escape local optima and maintain speed while conserving energy. Due to space limits, please refer to our response to Reviewer gxQ3 (W.2-3) for further analysis.
>
> **Response to Question orbi.4:**
>
> SPFT is an MPFT variant that disables multi-policy tracking and sparse region filling to highlight efficiency in high-dimensional tasks. As detailed on the bottom of page 25 (Appendix F.2), for FT-6, we set training episodes for all objectives except the first to zero ($\\{\Xi_{i},\Psi_{i}\\}_{i=2}^{6}=0$). Thus, only one trajectory is executed, corresponding to "Policy 1 tracking" in Fig. 1.
>
> **Response to Limitations orbi.1**
>
> We have added a discussion on the broader societal implications of deploying multi-objective RL algorithms to the "Impact Statements" section.
>
> *We thank you again for your evaluation. We sincerely hope that these explanations and additional results address your concerns and look forward to your feedback.*

---

> > ### Author Rebuttal · Reviewer_orbi · 2026-04-01
> >
> > Thank you for the clarification. I have some follow-up questions.
> >
> > - For the experiments with the random policy: if the initial solutions are close to each other and share similar directions, the outcomes are likely to converge to nearby regions. This makes your results understandable. However, this also suggests that the effectiveness of the framework may strongly depend on the problem’s initialization—specifically, how well Step 2 is initialized. In case Step 1 could not initialize the edges policy on all disconnected regions of the frontiers, I am highly doubt on the full approximation of the frontier.
> > - Since there appears to be an issue with the C-MORL setup, could this affect all of the reported C-MORL results in Table 1?

---

> > > ### Author Response · Authors · 2026-04-02
> > >
> > > Dear Reviewer orbi:
> > >
> > > Thank you for your follow-up questions. Here are our detailed responses to your new questions:
> > >
> > > **Response to Q1:**
> > >
> > > We apologize for any presentation confusion and appreciate the opportunity to clarify “initialize the edge policies on all disconnected regions of the frontiers“.
> > >
> > > First, finding "edge policies on all disconnected regions" is not Stage 1's objective. Importantly, MPFT does not rely on Stage 1 to cover all disconnected regions; such regions are explicitly handled by Stage 3.
> > > Stage 1 is designed to identify single-objective extremes (which we term Pareto-vertex policies). These policies represent the outer boundaries of the feasible objective space, not the geometric edges/vertices of every local disconnected region. Discovering policies within these disconnected regions is precisely the primary objective of Stage 3.
> > >
> > > Here is how MPFT densely approximates Pareto fronts:
> > >
> > > 1) Exposing gaps (Stages 1 \& 2): Unlike arbitrary random initialization, Stage 1 optimizes for single-objective extremes. Even on severely disconnected frontiers, robust single-objective RL reliably anchors searches at the objective space boundaries, ensuring outermost regions are not missed. Next, Stage 2 tracks from these Pareto-vertex policies. If the frontier is disconnected, continuous tracking naturally halts, leaving a noticeable "sparse gap." Thus, we utilize Stage 2 to precisely locate potential disconnected gaps.
> > >
> > > 2) Bridging disconnected regions via Stage 3 (Sparse Region Filling): When the sparse regions is detected post-Stage 2, our framework does not stop. Stage 3 identifies these regions (Algorithm 2 in Appendix B.2) and re-initializes random policies. Leveraging the weight adjustment method designed in Section 4.4, it performs targeted searches to find policies that fill these gaps (which we term Pareto-interior policies). These are highly likely to fall exactly within the "disconnected regions" you noted. Once discovered, they serve as new anchors to restart Pareto-tracking, bridging gaps to approximate the complete front.
> > >
> > > Thus, Stage 3 explicitly resolves concerns about missing disconnected regions. This context also explains our additional experiment during the discussion phase, where we initialized a random policy with fixed weights $[0.5, 0.5]$ to find the Pareto-interior policies in Stage 3. This evaluated the difficulty of finding Pareto-interior policies (Stage 3), not Pareto-vertex policies (Stage 1). The results demonstrate that without our specific weight adjustment method, it is difficult to hit policies within the sparse/disconnected regions. Once our method successfully locates a Pareto-interior policy within a disconnected region, the subsequent Pareto-tracking mechanism can effectively trace that sparse region. This directly addresses the concern that disconnected regions may be missed due to imperfect initialization.
> > >
> > > ***Regarding concerns that effectiveness "strongly depends" on Stage 1 initialization***:  Our robustness analysis in Appendix G.5 (Table 13) shows that even with poor initial solutions, MPFT's HV and EU degradation are bounded (drop by no more than $20\\%$), without harming the SP metric.
> > >
> > > Rigorously speaking, while MPFT's 4-stage design efficiently tracks and densely approximates the front, we acknowledge it cannot mathematically "guarantee" a "full approximation" of every possible disconnected region in highly complex fronts. ***However, to the best of our knowledge, no existing deep MORL method provides such guarantees, making this an inherent challenge rather than a limitation specific to MPFT.***
> > >
> > > **Response to Q2:**
> > >
> > > We thoroughly rechecked our code and confirm the evaluation oversight in the newly added Ant-v4 experiment did not affect Table 1 results. Original experiments used a separate, fully validated pipeline with correct settings.
> > >
> > > The large performance gap between MPFT and baselines in Ant-v5 might seem surprising. Besides the increased task difficulty, this gap may also be influenced by hyperparameter selection. To ensure a fair comparison, we aligned C-MORL's hyperparameters with PGMORL and PA2D-MORL. While we adopt unified hyperparameters for fairness, we acknowledge that different methods may have different sensitivities. Due to time limits, we could not conduct exhaustive hyperparameter tuning for every baseline. Overall, Table 1 presents a fair comparison under unified settings, robustly validating MPFT's superior performance.
> > >
> > > Thank you again for your question. We believe these clarifications fully address your concerns.
> > >
> > > ------
> > >
> > > We sincerely thank you for reconsidering our work and increasing your score. Your comments have been helping us improve the quality of this paper. Thank you again!

---

### Official Review · Reviewer_ThWv · 2026-03-12

**Soundness:** 3
**Presentation:** 3
**Significance:** 3
**Originality:** 3
**Overall Recommendation:** 4
**Confidence:** 4

**Summary:**

This paper proposes a Pareto optimization-based method to enhance learning efficiency of multi-objective learning. Unlike existing approaches, our method eliminates the need to maintain policy populations for multi-objective learning. The proposed framework is straightforward to implement: it begins with independent Pareto optimization for each objective, followed by a policy search within a joint policy space defined by a series of policy vertices. Furthermore, the method is highly compatible with existing reinforcement learning (RL) algorithms. Experimental results on several OpenAI Gym environments demonstrate that the proposed method achieves superior performance across three standard Pareto optimization metrics.

**Compliance With Llm Reviewing Policy:**

Affirmed.

**Key Questions For Authors:**

See Strengths and Weakensses

**Limitations:**

Yes

**Strengths And Weaknesses:**

### Strengths

1. This paper is easy to follow
2. The proposed method shows a good compatibility to existing RL methods
3. The ablation study gives a good analysis on the critical factors to the proposed method

### Weaknesses

1. The symbol system is somewhat complicated.

---

> ### Author Rebuttal · Authors · 2026-03-30
>
> We sincerely thank you for recognizing the value and contributions of our work.
>
> **Response to Weakness ThWv.1:**
>
> Thanks for your comments. We have made every effort to simplify the symbol notations while ensuring mathematical rigor. Please understand that to clearly and meticulously articulate the details of our proposed ideas and methods, it was unavoidable to introduce somewhat complex-looking symbols with various superscripts and subscripts. This level of formalization is indispensable for rigorously defining our framework and theoretical analysis.
>
> To alleviate potential reading difficulties caused by these notations, we have paired our mathematical descriptions with intuitive structural aids. Specifically, we provided the overall MPFT framework overview in Figure 1 to illustrate the high-level process, a comprehensive Key Notations table in Appendix A for quick reference, and the step-by-step execution logic in Algorithm 1.
>
> We welcome you to point out any symbols or definitions that remain unclear, and we will gladly refine the text to further improve readability. Please let us know if you have any other questions; we would be more than happy to address them.

---

> > ### Author Rebuttal · Reviewer_ThWv · 2026-04-08
> >
> > I thank the authors with their responses, I keep my score and I hope the authors can simplify their writing on the formulation system in the later revision, e.g., the "Training episodes" why we need to use two different notations to show that

---

> > > ### Author Response · Authors · 2026-04-08
> > >
> > > **Dear Reviewer ThWv:**
> > >
> > > Thank you for maintaining your positive score and for your continued support of our work!
> > >
> > > Regarding the notations for training episodes, we clarify why using two distinct symbols ($\Xi$ and $\Psi$) is necessary for our formulation system.
> > >
> > > 1) $\Xi$ specifically denotes the training episodes required for policy initialization (e.g., searching for Pareto-vertex or Pareto-interior policies in Stage 1/3). The value of $\Xi$ directly determines the performance of the policy initialization.
> > >
> > > 2) $\Psi$ denotes the training episodes dedicated to Pareto front tracking (Stage 2/4). The value of $\Psi$ directly determines the performance of the Pareto tracking stage.
> > >
> > > It is essential to distinguish them because they govern completely different optimization processes with distinct objectives and computational budgets. Using a single notation would obscure the underlying structure of our MPFT framework.
> > >
> > > However, we fully agree that this could cause initial confusion. To address your valid concern without losing mathematical precision, we have kept the notations distinct but added explicit, detailed descriptions in the revised manuscript right where these symbols are introduced, clearly differentiating their specific roles.
> > >
> > > Thank you again for your valuable feedback, which helps make our paper much clearer!

---

### Official Review · Reviewer_ijgF · 2026-03-13

**Soundness:** 3
**Presentation:** 3
**Significance:** 3
**Originality:** 3
**Overall Recommendation:** 5
**Confidence:** 4

**Summary:**

This paper investigates an interesting problem in multi-objective RL (MORL). In MORL, a common approach is to combine multiple rewards using a scalarization function and learn a single policy. This setting is typically referred to as single-policy MORL (SP-MORL). However, a single policy may not be sufficient in practice because user preferences are often difficult to specify precisely in advance. Thus, the learned policy may not align well with different possible user preferences. To address this issue, multi-policy MORL methods aim to learn a diverse set of policies that approximate the Pareto frontier. However, existing multi-policy approaches rely on evolutionary methoes that maintain a huge number of policies. These approaches can be computationally expensive and often require extensive environment interactions, which can be very costly in real world applications. To address this, the authors propose MPFT, which tracks the Pareto frontier directly and operates in four stages: (1) identifying extreme single-objective policies, (2) tracking the Pareto frontier from these vertices using alternating Pareto-reverse and Pareto-ascent updates, (3) identifying and filling sparse regions, and (4) aggregating all tracked policies to form the final Pareto-approximation set.

**Compliance With Llm Reviewing Policy:**

Affirmed.

**Final Justification:**

The authors have addressed my concerns in a satisfactory manner, and I have adjusted my score from weak accept to accept. I believe this is a solid paper, and the authors have done a good job in proposing a new MORL method and thoroughly evaluating it across several domains.

**Key Questions For Authors:**

See weaknesses and below for additional questions:
1. The experimental evaluation mainly considers domains with 2-3 objectives. How MPFT scale as the number of objectives increases (e.g., in settings with 100 objectives)?
2. As discussed above, how sensitive is the algorithm to the hyperparameters controlling the balance between reverse and ascent updates? Is there any systematic way for selecting these parameters?
3. In cases where the initial single-objective policies are suboptimal due to exploration challenges, how robust is the Pareto tracking process?

**Limitations:**

Yes

**Strengths And Weaknesses:**

Strengths:
1. The paper is very well written and easy to follow. Overall, it was a pleasant read.

2. The paper addresses an important yet relatively underexplored problem. Improving sample efficiency in multi-policy MORL while avoiding evolutionary populations is a meaningful and interesting research direction.

3. The proposed four-stage framework is well motivated and intuitive.

4. The introduction of the Pareto-reverse direction to escape vertex regions, combined with the Pareto-ascent direction for progressing along the frontier, is an elegant idea.

5. The experimental section is very good and thorough. The authors evaluate their method on a diverse set of complex domains and compare against relevant baselines. Additionally, the framework is extended beyond TD7 to algorithms such as PPO and SAC, which further demonstrates its flexibility.

Weaknesses: Overall, I found the paper interesting and I appreciate the elegance of the proposed approach and its thorough empirical evaluation. However, I have few concerns.:
1. The tracking method relies on extreme single-objective solutions. In domains with difficult exploration or where agents frequently converge to suboptimal rewards or local optima, the obtained extreme policies may themselves be suboptimal. In such cases, the Pareto tracking could become trapped near these suboptimal vertices. It would be helpful to discuss how robust the method is to imperfect vertex policies and how to mitigate this issue.
2. The effectiveness of MPFT appears to depend on the balance between Pareto-reverse steps and Pareto-ascent steps. This means that the method may require careful tuning of additional hyperparameters. Since RL algos are already sensitive to hyperparameter, this could introduce additional complexity and potentially affect stability.

---

> ### Author Rebuttal · Authors · 2026-03-30
>
> We sincerely thank you for recognizing the value and contributions of our work.
>
> **Response to Weakness ijgF.1 & Question ijgF.3 (Suboptimal vertex policies and robustness)**:
>
> Thanks for your comments. In multi-objective optimization tasks, the optimization difficulty varies significantly across objectives. For instance, in the Walker2d-2 environment, optimizing energy efficiency is much simpler than optimizing speed, which is why we set $\Xi_{2}$ smaller than $\Xi_{1}$ in Table 12. Consequently, even if a small budget $\Xi_{1}$ yields a suboptimal vertex-1 policy, we can often still obtain a high-quality vertex-2 policy. It is practically rare for all discovered vertex policies to fail simultaneously. Even if they do, our sparse region filling (Stage 3) acts as a safeguard to mitigate this issue, as it identifies gaps in the objective space and adaptively launches new tracking to ensure a dense and uniform approximation of the Pareto front. This proves MPFT is robust to initial policy quality and frontier nature.
>
> Furthermore, we have empirically validated this robustness in Appendix G.5 (Table 13). By reducing $\Xi_{i}$ and $\Xi_{k}$ to 10\%, we intentionally degraded the initial quality of the Pareto-vertex and interior policies. The results across five 2-objective environments show that HV and EU drop by no more than 20\%, while SP remains stable. By observing the training logs, we found that when tracking begins from a suboptimal Pareto-vertex, the policy does not immediately explore along the front. Instead, it continues to optimize all objectives simultaneously (i.e., all objective values increase). Only after reaching the vicinity of a high-quality Pareto-approximate policy does it begin tracking the frontier, at which point the number of policies in the non-dominant policy set begins to increase.
>
> **Response to Weakness ijgF.2 & Question ijgF.2 (Hyperparameter sensitivity for Pareto-reverse episodes $u$ and Pareto-ascent episodes $v$):**
>
> Thanks for your comments. The algorithm's sensitivity to $u$ and $v$ is fundamentally tied to the self-correcting behavior described above, which provides a systematic rule for selecting them: $v$ needs to be greater than $u$. By definition, the Pareto-ascent direction strictly increases all objective values before reaching Pareto-stationarity. When $v > u$ (the Pareto-ascent proportion exceeds Pareto-reverse), a suboptimal policy leverages the ascent direction to continuously climb until hitting the true frontier, after which tracking commences.
>
> Conversely, when $v \le u$, the objective values of a suboptimal vertex fluctuate wildly. This instability occurs because the heavier weighting of the Pareto-reverse direction destroys the policy's stability before it can securely anchor near a high-quality vertex. As long as $v > u$, the framework naturally overcomes suboptimal initializations and yields stable performance. Our theoretical analysis in Appendix C supports this point. Empirically, Fig. 9 also demonstrated that as long as $u < v$, the framework yields stable HV and SP. For simpler environments, $u=0, v=2$ is enough, while more complex environments benefit from $u=1, v=2$ to encourage broader exploration.
>
> **Response to Question ijgF.1 (Scaling to 100 objectives):**
>
> Thanks for your question. To the best of our knowledge, in currently recognized MORL benchmarks, the highest number of objectives evaluated is 9 in the Building-9, and our proposed MPFT handles this effectively as shown in Table 3. Theoretically, our method can be easily equipped for dimensions higher than 9 (or even up to 100 objectives). This is because MPFT tracks policies directly without maintaining a policy population, ensuring its time complexity scales linearly, $\mathcal{O}(m)$. This makes it inherently more scalable to massive objective spaces than evolutionary frameworks. Furthermore, we have theoretically proven that the convergence of our Pareto-tracking mechanism toward the Pareto-stationary set is independent of the number of objectives. Ultimately, the empirical difficulty in MORL scales primarily with the severity of objective conflicts rather than the sheer number of dimensions.
>
> *We sincerely hope that these explanations address your concerns. We look forward to hearing from you.*

---

> > ### Author Rebuttal · Reviewer_ijgF · 2026-04-03
> >
> > Thanks for your reply. The authors have adequately addressed my comments, so I will increase my scores accordingly.

---

> > > ### Author Response · Authors · 2026-04-08
> > >
> > > **Dear Reviewer ijgF:**
> > >
> > > Thank you very much for reading our rebuttal and for increasing your score. We are delighted that our responses have adequately addressed your comments. We appreciate your time, effort, and constructive feedback throughout the review process, which has helped us improve the quality of our paper.
> > >
> > > Thank you again for your support!

---

### Decision · Program_Chairs · 2026-04-30

**Decision:**

Accept (regular)

**Comment:**

The paper proposes a novel non-evolutionary multi-policy pareto-front tracking algorithm for Multi-Objective RL (MORL).
The proposed approach is evaluated on continuous-control tasks with promising results.

The reviewers agree that the proposed method is valuable, its advantages over existing methods clearly explained and convincingly supported by well-designed and thorough experiments.

The discussion phase played a fundamental role in clarifying some important aspects of the work and solving some small but non-negligible issues. For this reason, I urge the authors to incorporate the reviewer's suggestions and the proposed fixes in the final version. The most important ones are:

- Adding the offline learning results to the paper. Consider using the extra page for this.
- Correcting the discrepancies in the comparison with C-MORL that emerged in the discussion with Reviewer gxQ3
- Releasing the code

Notice that new experiments on Ant-v4 were useful to dismiss some of the reviewers' concerns, but should not be included in the paper since Ant-v4 is deprecated by gymnasium. If time allows, consider adding low-budget experiments on the entire control suite, but I do not consider this addition necessary for publication.